# Reward-oriented Causal Representation Learning

**Zirui Yan**[*]
Rensselaer Polytechnic Institute
yanz11@rpi.edu

**Emre Acartürk**[*]
Rensselaer Polytechnic Institute
acarte@rpi.edu

**Ali Tajer**
Rensselaer Polytechnic Institute
tajer@ecse.rpi.edu

## Abstract

Causal representation learning (CRL) is the process of disentangling the *latent* low-dimensional causally-related generating factors underlying high-dimensional observable data. Extensive recent studies have characterized CRL identifiability and *perfect* recovery of the latent variables and their attendant causal graph. This paper introduces the notion of *reward-oriented* CRL, the purpose of which is to move away from perfectly learning the latent representation and instead learning it to the extent needed for optimizing a desired downstream task (reward). In reward-oriented CRL, perfectly learning the latent representation can be excessive; instead, it must be learned at the *coarsest* level sufficient for optimizing the desired task. Reward-oriented CRL is formalized as the optimization of a desired function of the observable data over the space of all possible interventions and focuses on linear causal and transformation models. To sequentially identify the optimal subset of interventions, an adaptive exploration algorithm is designed that learns the latent causal graph and the variables needed to identify the best intervention. It is shown that for an $n$-dimensional latent space and a $d$-dimensional observation space, over a horizon $T$ the algorithm's regret scales as $\tilde{O}(d^{\frac{1}{3}}n^{\frac{1}{3}}u^{\frac{2}{3}}T^{\frac{2}{3}} + u\sqrt{T})$, where $u$ measures total uncertainty in the graph estimates. Furthermore, an almost-matching lower bound is shown to scale as $\Omega(d^{\frac{1}{3}}n^{\frac{1}{3}}p^{\frac{2}{3}}T^{\frac{2}{3}} + p\sqrt{T})$, in which $u$ is replaced by $p$ that counts the number of causal paths in the graph.

## 1 Introduction

Consider a data-generating process in which *latent* low-dimensional causally-related variables are mapped to *observational* high-dimensional data through an *unknown* transformation. Causal representation learning (CRL) is the process of using observational data to learn the latent, unobserved generating factors, i.e., the latent variables and the latent causal graph that specifies their causal interactions. CRL is considered a significant step toward understanding the world by learning appropriate representations that support causal interventions, reasoning, and planning [1].

**CRL literature.** There exists rich recent literature on CRL *identifiability* analysis – the objective of which is establishing conditions under which the latent space can be recovered uniquely – across various models for the latent causal model (e.g., linear, parametric, and non-parametric) and the unknown transformation (e.g., linear, parametric, and non-parametric). Some representative studies include [2–13]. Aiming to establish possibility/impossibility results, the existing literature is primarily focused on the asymptotic setting of access to an infinite number of observations, with limited studies on finite-sample guarantees [4]. Specifically, we refer to [2–5] for the most closely related works in terms of setting and methodology.

---

[*]Equal contribution.

39th Conference on Neural Information Processing Systems (NeurIPS 2025).

Learning a unique, causal representation underlying the observed data is hypothesized to enable more improved and robust reasoning for *downstream* tasks [1]. This is often encountered in technological, social, and biological domains where the observed data often lack straightforward interpretations and are generated by an unobserved data-generating mechanism with interpretable semantics.

**Reward-oriented CRL.** To understand the recovery limits in CRL, the existing literature has so far decoupled the CRL from the downstream objectives. This decoupling means that CRL may expend extra effort learning fine-grained details that do not contribute to the downstream objective, or conversely, lack sufficient accuracy to be useful. To address this gap, we introduce the notion of *reward-oriented CRL*, which directly integrates the downstream goal into the representation learning pipeline. In this paper, we consider rewards that are functions of the latent variables. Since the latent variables are not directly observable, efficient optimization of the reward may include, as a sub-task, recovering these latent representations. In this case, one would only need to learn the latent variables and graph to the coarsest level to optimize the downstream objectives. To formalize the objectives, we define a *utility* function that maps the latent causal system to a downstream utility. This utility will be subsequently optimized over the space of possible interventions in the causal system. For example, consider a robotic arm with causally related joint variables. The arm's movements are monitored from camera images, and our downstream objective is to use these images to optimize the arm's movements for a specific task. In such a task, achieving high placement accuracy does not require perfect recovery of all latent variables; instead, it suffices to capture the critical joint relationships or to bound estimation errors below a level that leaves final task performance unaffected. Sending commands to the arm to adjust its actions is the intervention model on the underlying causal system.

Table 1: Summary of results for reward-oriented CRL.

| Intervention | Latent variables | Graph recovery | Regret bounds |
|---|---|---|---|
| soft | up to scaling & mixing | transitive closure | Upper bound: $\tilde{O}(d^{\frac{1}{3}}n^{\frac{1}{3}}u_{\mathrm{S}}^{\frac{2}{3}}T^{\frac{2}{3}} + u_{\mathrm{S}}\sqrt{T})$
Lower bound: $\Omega(d^{\frac{1}{3}}n^{\frac{1}{3}}p^{\frac{2}{3}}T^{\frac{2}{3}} + p\sqrt{T})$ |
| hard | up to scaling | perfect | Upper bound: $\tilde{O}(d^{\frac{1}{2}}n^{\frac{1}{3}}u_{\mathrm{H}}^{\frac{2}{3}}T^{\frac{2}{3}} + u_{\mathrm{H}}\sqrt{T})$ |

**Connection to causal bandits.** Optimizing utility in the latent space faces uncertainties in several ways, e.g., causal model, transformation, and probability models. Inevitably, an algorithm optimizing utility needs to explore the system to resolve these uncertainties before committing to a decision. A data-adaptive exploration of interventions is intimately related to the literature on causal bandit (CB), albeit with two significant differences. First, in CB, the learner directly interacts with the causal system and observes the causal variables, a premise that does not hold in reward-oriented CRL. Second, in the CB literature, it is often assumed that the causal graph's topology is known, which is not the case in reward-oriented CRL. Some representative studies on various causal models (e.g., linear and non-linear) and intervention models ($do$, stochastic hard, and stochastic soft) include [14–26].

Due to these two significant differences with CB, designing reward-oriented algorithms will differ significantly from the CB algorithms by including a process that can perform CRL. This process has to perform an accurate-enough CRL that facilitates identifying the best intervention. Hence, the CRL process we need to design differs from the existing ones that aim for *perfect* latent recovery.

We focus on reward-oriented CRL with (i) linear structural equation models over an $n$-dimensional latent causal system, (ii) linear transformations mapping to $d$-dimensional observations, and (iii) linear utility functions. Table 1 summarizes the main guarantees on *latent space recovery* and *regret bounds* until time $T$, where $u$ measures total uncertainty and $p$ counts the number of causal paths. We defer the discussion of $do$ intervention to Appendix I. Our key observations include the following.

- **Finite-sample identifiability.** We provide finite-sample identifiability for linear SEMs, accommodating both stochastic soft and hard interventions. This is a critical step, since during exploration, we have access to only a finite number of samples.

- **Almost matching regret bounds.** We establish upper and lower regret bounds for reward-oriented CRL. These bounds are specified in terms of the topology of the causal graph. Under soft interventions, these bounds match in their dependence on graph parameters and the time horizon $T$.

- **Refined bounds for causal bandits.** The reward-oriented CRL framework subsumes CB by setting the latent to observation transformation to the identity function. We show that our algorithm also improves the state-of-the-art regret bounds for the relevant CB settings.

**Notations.** For $n \in \mathbb{N}$, we define $[n] \triangleq \{1, \ldots, n\}$. Vectors are represented by lowercase bold letters, and element $i$ of vector $\mathbf{v}$ is denoted by $\mathbf{v}[i]$. Matrices are represented by uppercase bold letters, and we denote row $i$ and element $(i, j)$ of matrix $\mathbf{A}$ by $[\mathbf{A}]_i$ and by $\mathbf{A}[i, j]$, respectively. Moore-Penrose pseudoinverse of a matrix $\mathbf{A}$ is denoted by $\mathbf{A}^\dagger$. For any matrix $\mathbf{A}$ we denote the rank, column and null spaces of $\mathbf{A}$ by $\mathsf{rank}(\mathbf{A})$, $\mathsf{col}(\mathbf{A})$ and $\mathsf{null}(\mathbf{A})$, respectively. $\mathbb{1}$ denotes the indicator function. Sets and events are denoted by calligraphic letters. The cardinality of the set $\mathcal{A}$ is denoted by $|\mathcal{A}|$. For a vector $\mathbf{x}$ and positive semidefinite matrix $\mathbf{A}$, we define $\|\mathbf{x}\|_\mathbf{A} = \sqrt{\mathbf{x}^\top \mathbf{A} \mathbf{x}}$ as the weighted $\ell_2$ norm. The $\ell_2$-norms of vector $\mathbf{x} \in \mathbb{R}^d$ and matrix $\mathbf{A}$ are denoted by $\|\mathbf{x}\|_2$ and $\|\mathbf{A}\|_2$, respectively. Finally, $\tilde{\mathcal{O}}$ is an order notation that ignores constant and poly-logarithmic factors.

## 2 Reward-oriented CRL Framework

**Data-generating process.** Consider a data-generating process that transforms high-level, low-dimensional latent variables into low-level, high-dimensional observable data. Formally, consider a causal system represented by an *unknown* directed acyclic graph (DAG) $\mathcal{G}$ with $n$ nodes generating causally-related latent random variables $Z \triangleq [Z[1], \ldots, Z[n]]^\top$. These latent variables are mapped to a higher-dimensional observed data $X \in \mathbb{R}^d$ by an *unknown* linear transformation $\mathbf{G} \in \mathbb{R}^{d \times n}$ according to $X = \mathbf{G} \cdot Z$, where $d \geq n$ and $\mathbf{G}$ is full column rank. The set of parents and ancestors of node $i \in [n]$ are denoted by $\mathsf{pa}(i)$ and $\mathsf{an}(i)$, respectively. We denote the probability density function (pdf) of $Z$ by $p$ and denote the conditional pdf of $Z[i]$ given its parent variables by $p_i(z[i] \mid z[\mathsf{pa}(i)])$. We call a permutation $\pi = (\pi_1, \ldots, \pi_n)$ of $[n]$ a *valid causal order* if for all $i, j \in [n]$, the membership $i \in \mathsf{pa}(j)$ implies $\pi_i < \pi_j$. We denote the maximum in-degree of $\mathcal{G}$ by $d_\mathcal{G}$ and the length of its longest causal path by $L$. The latent causal variables are assumed to be related through a linear structural equation model (SEM) specified by $Z = \mathbf{B} \cdot Z + \varepsilon$, where $\mathbf{B} \in \mathbb{R}^{n \times n}$ is the edge weight matrix and $\varepsilon \in \mathbb{R}^n$ accounts for the exogenous noise, whose expected value is denoted by $\nu \triangleq \mathbb{E}[\varepsilon]$. The weight matrix $\mathbf{B}$ directly models the causal relations in the sense that the element $\mathbf{B}[i, j]$ is non-zero if and only if $j \in \mathsf{pa}(i)$.

**Interventions.** We consider two types of intervention mechanisms: stochastic *hard* and *soft*, the distinction of which is how they impact the marginal distributions of the latent variables.

- *Soft intervention:* As the least restrictive form of intervention, when applied to node $i \in [n]$, a soft intervention changes the conditional distribution $p_i(z[i] \mid z[\mathsf{pa}(i)])$ to a distinct one. Equivalently, this alters the *observational* linear causal mechanism to an alternative one as follows:

$$\text{observational:} \quad Z[i] = [\mathbf{B}]_i \cdot Z + \varepsilon[i], \qquad \text{interventional:} \quad Z[i] = [\mathbf{B}^*]_i \cdot Z + \varepsilon^*[i], \quad (1)$$

  where $[\mathbf{B}^*]_i$ denotes the vector of post-intervention edge weights, and $\varepsilon^*[i]$ denotes the post-intervention noise with mean $\nu^*[i] \triangleq \mathbb{E}[\varepsilon^*[i]]$. Consequently, we define the interventional weight matrix $\mathbf{B}^*$ as the composition of rows $\{[\mathbf{B}^*]_i : i \in [n]\}$ and the mean vector as $\nu^*$.
- *Hard intervention:* As a special case of soft interventions, a *hard* intervention on node $i$ removes its ancestral statistical dependence and changes $p_i(z[i] \mid z[\mathsf{pa}(i)])$ to a *marginal* distribution that is only a function of $z[i]$. This mechanism is equivalent to setting $\mathbf{B}^* = 0$.

For simplicity of notations, throughout the analysis, we assume only one intervention mechanism per node. However, this can be readily generalized to an arbitrary number as discussed in Appendix H. We allow multiple nodes to be intervened on simultaneously and denote the space of possible interventions by $\mathcal{A} \triangleq 2^{[n]}$. We denote the probability measure induced by the set of interventions $\mathbf{a} \in \mathcal{A}$ and the associated expectation by $\mathbb{P}_\mathbf{a}$ and $\mathbb{E}_\mathbf{a}$, respectively.

**Reward-oriented CRL.** In reward-oriented CRL, the objective is to identify the set of interventions in $\mathcal{A}$ that maximizes an expected reward defined as a function of $Z$. In this paper, we focus on linear reward functions specified by $U(Z) \triangleq \theta^\top Z + \varepsilon_U$, where $\theta \in \mathbb{R}^n$ is an *unknown* reward parameter and $\varepsilon_U$ represents a utility noise term with mean 0. We denote the expected value of the utility $U$ under intervention $\mathbf{a} \in \mathcal{A}$ by

$$\mu_\mathbf{a} \triangleq \mathbb{E}_\mathbf{a}[U(Z)] . \tag{2}$$

A learner's objective is to identify the optimal intervention $\mathbf{a}^*$, denoted by

$$\mathbf{a}^* \triangleq \arg\max_{\mathbf{a} \in \mathcal{A}} \mu_\mathbf{a} . \tag{3}$$

All aspects of probability distributions, i.e., the topology of $\mathcal{G}$, pre- and post-intervention matrices, and noise distributions, are unknown. To identify $\mathbf{a}^*$, the learner performs a sequence of interventions

and receives feedback consisting of the observed data $X$ and reward $U(Z)$. The objective is to identify $\mathbf{a}^*$ with the fewest number of interventions, which are selected adaptively based on the data and the collected rewards. The sequence of interventions over time is denoted by $\{\mathbf{a}_t \in \mathcal{A} : t \in \mathbb{N}\}$. Accordingly, we denote the set of the latent variables, data, and reward collected up to $t$ by

$$\mathbf{Z}_t \triangleq [Z_1, \ldots, Z_t], \quad \mathbf{X}_t \triangleq [X_1, \ldots, X_t], \quad \text{and} \quad \mathbf{U}_t \triangleq [U(Z_1), \ldots U(Z_t)]. \tag{4}$$

To assess the efficiency of the learner in identifying $\mathbf{a}^*$, we define the cumulative utility regret that it incurs relative to an oracle with access to the best intervention $\mathbf{a}^*$ as

$$\mathcal{R}_T \triangleq \sum_{t=1}^{T} \left( \mathbb{E}_{\mathbf{a}^*}[U(Z)] - \mathbb{E}_{\mathbf{a}_t}[U(Z)] \right) = T\mu_{\mathbf{a}^*} - \sum_{t=1}^{T} \mu_{\mathbf{a}_t}. \tag{5}$$

We remark that using such a regret-based approach to identify $\mathbf{a}^*$ can be naturally posed as a multi-armed bandit problem in which each of the $2^n$ possible interventions can be represented by one arm. Applying a vanilla bandit algorithm results in a regret that scales exponentially in $n$, rendering a regret that, even for a moderate-sized latent structure, can be highly inefficient. The objective in this paper is to design algorithms that can break the exponential dependence on $n$ by properly leveraging the intricate causal structures among the latent variables.

**Identifiability metrics.** Circumventing the exponential dependence of the regret on dimension $n$, and efficiently identifying $\mathbf{a}^*$ hinge on properly recovering the latent variables $Z$ and the underlying causal graph $\mathcal{G}$, both unobserved. These recoveries are the core objective of CRL. Recent studies provide extensive *identifiability* guarantees for CRL – which specify conditions under which $Z$ and $\mathcal{G}$ can be uniquely recovered (up to some uncertainty) from the observed data $X$. These guarantees are *asymptotic*, assuming access to an *infinite* number of samples of $X$. In the reward-oriented CRL, however, the decisions at each time $t$ must be based solely on $t$ samples. To formalize the types of identifiability guarantees needed in our framework, we first specify the known infinite-sample identifiability guarantees that apply to the setting of this paper (linear SEMs and transformation), followed by their finite-sample counterparts. For this purpose, given $\mathbf{X}_t$, we define $\hat{\mathcal{G}}_t$, $\hat{\mathbf{Z}}_t$, and $\mathbf{H}_t$ as estimates of $\mathcal{G}$, $\mathbf{Z}_t$, and $\mathbf{G}^\dagger$, respectively. We also denote the transitive closure of $\mathcal{G}$ by $\mathcal{G}_{\mathrm{tc}}$.

**Theorem 1.** *Under linear SEM, linear transformation, and one intervention per node, CRL is endowed with the following* infinite-sample *identifiability guarantees.*

  *(i)* **Hard intervention** ([2, Theorem 2]): *It is possible to perfectly recover $\mathcal{G}$ and recover $Z$ up to scaling, i.e., $\hat{\mathcal{G}}_\infty = \mathcal{G}$ and $\hat{\mathbf{Z}}_\infty \triangleq \mathbf{H}_\infty \mathbf{X}_\infty = \mathbf{C}_{\mathrm{H}} \mathbf{Z}_\infty$ for some diagonal matrix $\mathbf{C}_{\mathrm{H}} \in \mathbb{R}^{n \times n}$.*

  *(ii)* **Soft intervention:** *It is possible to perfectly recover a transitive closure of $\mathcal{G}$ [2, Theorem 1] and recover $Z$ up to scaling and mixing with parents, i.e., $\hat{\mathcal{G}}_\infty = \mathcal{G}_{\mathrm{tc}}$ and $\hat{\mathbf{Z}}_\infty = \mathbf{H}_\infty \mathbf{X}_\infty = \mathbf{C}_{\mathrm{S}} \mathbf{Z}_\infty$ for some matrix $\mathbf{C}_{\mathrm{S}} \in \mathbb{R}^{n \times n}$ such that for any given $j$, $\mathbf{C}_{\mathrm{S}}[i,j] = 0$ for all $i \notin \{j\} \cup \mathsf{pa}(j)$.*

See Appendix E.1 for details. Next, we specify finite-sample counterparts of these identifiability statements in a probably approximately correct (PAC) sense similar to [4].

**Definition 1** (($\epsilon, \delta$)-PAC recovery). *For a given $t \in \mathbb{N}$, the finite-sample estimates $\hat{\mathcal{G}}_t$, $\hat{\mathbf{Z}}_t$, and $\mathbf{H}_t$ are said to achieve ($\epsilon, \delta$)–PAC recovery if the following statements hold with probability at least $1 - \delta$*

  *(i)* **Hard intervention:** *$\hat{\mathcal{G}}_t = \mathcal{G}$ and $\hat{\mathbf{Z}}_t = \mathbf{H}_t \mathbf{X}_t = (\mathbf{C}_{\mathrm{H}} + \mathbf{E}_t) \cdot \mathbf{Z}_t$, where $\mathbf{C}_{\mathrm{H}}$ is a full-rank diagonal matrix, and we have $\|\mathbf{E}_t\|_2 \leq \epsilon$.*

  *(ii)* **Soft intervention:** *$\hat{\mathcal{G}}_t = \mathcal{G}_{\mathrm{tc}}$ and $\hat{\mathbf{Z}}_t = \mathbf{H}_t \mathbf{X}_t = (\mathbf{C}_{\mathrm{S}} + \mathbf{E}_t) \cdot \mathbf{Z}_t$, where $\mathbf{C}_{\mathrm{S}}$ is a full rank matrix, for any given $j$, $\mathbf{C}[i,j] = 0$ a for all $i \notin \{j\} \cup \mathsf{pa}(j)$, and $\|\mathbf{E}_t\|_2 \leq \epsilon$.*

**Assumptions.** Next, we outline the assumptions adopted for the reward-oriented CRL framework. First, we adopt the following CRL assumption (see, e.g., [2, Assumption 1(b)] and [3, Assumption 1]), which ensures that the effects of an intervention can always be traced from the changes in the precision matrix (inverse covariance) of latent variables $Z$.

**Assumption 1.** *An intervention on any non-root node $i$ changes row $i$ of the weight matrix, i.e.,*

$$\text{if} \quad \mathsf{pa}(i) \neq \emptyset \quad \text{then} \quad [\mathbf{B}]_i \neq [\mathbf{B}^*]_i. \tag{6}$$

This assumption automatically holds for hard and *do* interventions and is mild for soft interventions, requiring that the effect of an intervention is not limited to the exogenous noise distributions. Similar assumptions are common in the CRL literature and can effectively be interpreted as requiring that the statistics we use be non-degenerate. Next, we provide SEM-related assumptions that are standard in the causal bandit literature [21].

**Assumption 2** (Weight matrices). *Matrices* $\mathbf{B}$ *and* $\mathbf{B}^*$ *and the utility parameter* $\theta$ *are unknown but have finite entries with known ranges. We denote the range of these entries by* $m_B \in \mathbb{R}^+$, *i.e.,* $|\mathbf{B}[i,j]| \leq m_B$, $|\mathbf{B}^*[i,j]| \leq m_B$, *and* $|\theta[i]| \leq m_B$ *for all* $i,j \in [n]$.

**Assumption 3** (Noise model). *We assume that the statistical models for the noise* $\nu$ *and* $\nu^*$ *are unknown and bounded. We define the* $m_\varepsilon \in \mathbb{R}^+$ *to specify the range of noise terms, i.e.,* $|\varepsilon_t[i]| \leq m_\varepsilon$ *for all* $i \in [n]$ *and* $t \in [T]$. *Finally, we assume the utility noise* $\varepsilon_U$ *is* 1-*sub-Gaussian.*

For simplicity in the presentation and without loss of generality, we set $m_B = m_\varepsilon = 1$. An immediate conclusion of Assumptions 2 and 3 is that $Z$ is bounded, i.e., $\|Z\| \leq m$ for some $m \in \mathbb{R}_+$.

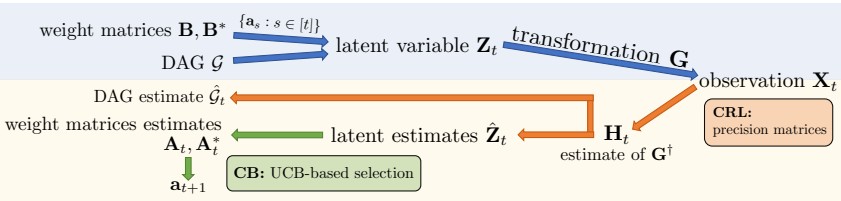

Figure 1: Schematic Pipeline of RO-CRL

# 3 Reward-oriented CRL Algorithm

Now we introduce the **R**eward-**o**riented **C**ausal **R**epresentation **L**earning algorithm (RO-CRL) for hard and soft intervention. The pseudocode is provided in Algorithm 1 in Appendix A.

## 3.1 Algorithmic Overview & Key Properties

Identifying the best intervention $\mathbf{a}^*$ defined in (3) requires estimating the expected utility under $2^n$ distinct statistical models associated with $2^n$ possible intervention combinations $\mathbf{a} \in \mathcal{A}$. The key to breaking the exponential dependence of the number of estimation routines we have to perform on $n$ is leveraging the intricate connection among all the $2^n$ statistical models. Specifically, all the models inherit their randomness from two sources of noise models, $\nu$ and $\nu^*$. The algebraic forms of the statistical models are shaped by the weight matrices $\mathbf{B}$ and $\mathbf{B}^*$ and the utility parameters $\theta$. Hence, learning $2^n$ distributions can be reduced to estimating two noise models and estimating the entries of $\mathbf{B}$, $\mathbf{B}^*$, $\theta$. Estimating these parameters, however, depends on access to the causal graph $\mathcal{G}$ (informative about the sparsity structures of $\mathbf{B}$ and $\mathbf{B}^*$) and the latent variables $\mathbf{Z}_t$, both of which are not directly observable.

To address this, RO-CRL properly explores various intervention combinations to form reliable enough estimates of $\mathcal{G}$ and $\mathbf{Z}_t$ from observations $\mathbf{X}_t$. Achieving the right level of exploration is critical to avoid collecting excessive data $\mathbf{X}_t$, which will compromise the regret. Attaining the right exploration, therefore, depends on determining what the *coarsest* possible estimates of $\mathcal{G}$ and $\mathbf{Z}_t$ are that ensure reliable identification of the optimal intervention $\mathbf{a}^*$. Hence, RO-CRL balances a trade-off that involves an adaptive exploration schedule that focuses on recovering $\mathcal{G}$ and $\mathbf{Z}_t$ to the extent needed to find $\mathbf{a}^*$. At each time $t$, RO-CRL constructs estimates for the graph $\hat{\mathcal{G}}_t$ and inverse transform $\mathbf{H}_t$ using differences between sample precision matrices with samples from the observational and interventional data. Since CRL only recovers variables up to scaling factors, we compute robust estimates of appropriately scaled $\mathbf{B}$ and $\mathbf{B}^*$. After sufficient exploration of interventions, they are subsequently selected according to the upper confidence bound (UCB) principle. The overall pipeline of this process is illustrated in Figure 1.

**Precision matrix differences.** The main statistic we use to estimate the latent variables and graph is the differences in precision matrices under various interventions. Denote the precision matrices of $Z$ and $X$ under distribution $\mathbb{P}_\mathbf{a}$ by $\Theta_\mathbf{a}^Z$ and $\Theta_\mathbf{a}$, respectively. Accordingly, define $\mathbf{R}_i^Z$ and $\mathbf{R}_i$ as the associated precision differences between observational ($\mathbf{a} = \emptyset$) and single-node interventional ($\mathbf{a} = \{i\}$) data, i.e., $\mathbf{R}_i^Z \triangleq \Theta_{\{i\}}^Z - \Theta_\emptyset^Z$ and $\mathbf{R}_i \triangleq \Theta_{\{i\}} - \Theta_\emptyset$. To specify the empirical counterparts of these precision differences, we denote the number of times that $\mathbf{a} \in \mathcal{A}$ is selected until time $t$ by

$$N_{\mathbf{a},t} \triangleq \sum_{s \in [t]} \mathbb{1}\{\mathbf{a}_s = \mathbf{a}\}. \tag{7}$$

Hence, the empirical sample mean and covariance of $X$ for $\mathbf{a} \in \mathcal{A}$ are given by

$$\mu_{\mathbf{a},t} \triangleq \frac{1}{N_{\mathbf{a},t}} \sum_{s \in [t]} \mathbb{1}\{\mathbf{a}_s = \mathbf{a}\} X_s , \quad \Sigma_{\mathbf{a},t} \triangleq \frac{1}{N_{\mathbf{a},t}} \sum_{s \in [t]} \mathbb{1}\{\mathbf{a}_s = \mathbf{a}\} X_s X_s^\top - \mu_{\mathbf{a},t} \cdot \mu_{\mathbf{a},t}^\top . \quad (8)$$

Accordingly, the empirical precision and precision differences are denoted by $\Theta_{\mathbf{a},t} \triangleq (\Sigma_{\mathbf{a},t})^\dagger$ and $\mathbf{R}_{i,t} \triangleq \Theta_{\{i\},t} - \Theta_{\emptyset,t}$, respectively. We use the following properties of $\mathbf{R}_{i,t}$ to estimate $\mathcal{G}$ and $Z$.

**Lemma 1.** *The non-zero rows of latent precision difference $\mathbf{R}_i^Z$ describe the latent graph:*

$$\left\| \left[\mathbf{R}_i^Z\right]_j \right\|_2 \neq 0 \implies j \in \{i\} \cup \mathsf{pa}(i) . \quad (9)$$

*Furthermore, the observable precision difference $\mathbf{R}_i = (\mathbf{G}^\dagger)^\top \mathbf{R}_i^Z \mathbf{G}^\dagger$ is subspace-constrained:*

$$\mathsf{col}(\mathbf{R}_i) \subseteq \mathsf{span}\{[\mathbf{G}^\dagger]_j \ : \ j \in \{i\} \cup \mathsf{pa}(i)\} . \quad (10)$$

## 3.2 Key Processes in RO-CRL

RO-CRL consists of two main stages. It begins with a *forced exploration phase*, in which we apply node-level atomic interventions on all nodes a fixed number of times. The purpose of forced exploration is to establish initial estimates for the relevant statistics. This is followed by an *adaptive exploration stage*, in which we dynamically select a sequence of interventions adaptively to the observations and utilities. This stage itself consists of several inference processes, whose collective objective is to specify a rule for translating observations into an intervention selection.

**Stage 1 – Forced exploration.** To construct initial estimates of $\mathbf{R}_i$, the algorithm explores the observational model along with each of the *single-node* atomic intervention models for $T_0$ times. To formalize this, the set of interventions in this forced exploration phase is denoted by $\mathcal{A}_0 \triangleq \{\emptyset, \{1\}, \{2\}, \ldots, \{n\}\}$. The constant $T_0$ is chosen such that the initial $\mathbf{R}_i$ are sufficiently accurate to produce a reliable enough graph estimate for the latent causal graph $\mathcal{G}$.

**Stage 2 – Adaptive exploration.** After forced exploration, at every $t \geq (N+1)T_0$, we perform three inference and decision procedures to identify the next intervention $\mathbf{a}_{t+1}$.

1. **CRL:** First, we recover the latent variables and graph. This is initiated by first finding the estimate $\mathbf{H}_t$ for the inverse of $\mathbf{G}$, followed by using that to find estimates $\hat{\mathcal{G}}_t$.

2. **UCB-based intervention selection:** Subsequently, we introduce a UCB-based decision rule that leverages the above estimates in conjunction with the reward $\mathbf{U}_t$ to specify $\mathbf{a}_{t+1}$.

3. **Under-sampling rule:** Finally, we implement a process that ensures that single-node interventions are explored sufficiently over time. When an intervention is deemed under-sampled, it will be prioritized for sampling over the intervention. This process ensures that the estimates $\mathbf{H}_t$ and $\hat{\mathcal{G}}_t$ are updated incrementally, preventing total estimation error in $\mathbf{Z}_t$ from growing linearly with $t$.

## 3.3 Inference & Decision Rules

The quality of the inference and decision rules in Stage 2 is critical for identifying the sequence of interventions. In this subsection, we specify these rules.

**1 – CRL rules.** The CRL rules are implemented in two stages: *baseline recovery* and *refined recovery*. The *baseline recovery* process serves two key roles: (i) it estimates the causal graph $\mathcal{G}$ and latent variables $\mathbf{Z}_t$ under soft interventions, and (ii) it acts as an intermediate step for estimating $\mathcal{G}$ and $\mathbf{Z}_t$ under hard interventions. The *refined recovery* stage is applied only when hard interventions are present, with the specific goal of improving estimates produced by the baseline process. As expected, hard interventions yield stronger recovery guarantees, with their advantage stemming from this additional refinement step.

**1.a – Baseline recovery steps.** The baseline recovery of the latent space is based on the shared properties of soft and hard interventions, such as Lemma 1, and it has three inference routines.

(i) **Inverse transform $\mathbf{H}_t$.** We construct the baseline estimate $\mathbf{H}_t \in \mathbb{R}^{n \times d}$ row by row by assigning

$$[\mathbf{H}_t]_i \leftarrow \text{ the principal eigenvector of } \mathbf{R}_{i,t} , \quad (11)$$

where *principal eigenvector* denotes the eigenvector associated with the largest absolute eigenvalue of a symmetric matrix. The intuition here is that, due to Lemma 1, with high probability, the following property holds with a low error level.

$$\left([\mathbf{H}_t]_i \cdot \mathbf{G}\right)[j] \approx 0 , \qquad \forall j \notin \{i\} \cup \mathsf{pa}(i) . \quad (12)$$

(ii) **Variables $\hat{\mathbf{Z}}_t$.** The estimated latent variables are subsequently estimated via

$$\hat{\mathbf{Z}}_t = \mathbf{H}_t \mathbf{X}_t = (\mathbf{H}_t \mathbf{G}) \cdot \mathbf{Z}_t \,, \tag{13}$$

and are approximately equal to the true latent variables $\mathbf{Z}_t$ up to mixing with parent variables.

(iii) **Graph $\hat{\mathcal{G}}$.** To form a graph estimate, we first compute the *estimated* latent precision differences

$$\hat{\mathbf{R}}_{i,t}^Z \triangleq (\mathbf{H}_t^\dagger)^\top \cdot \mathbf{R}_{i,t} \cdot \mathbf{H}_t^\dagger \,, \qquad \forall i \in [n] \,, \tag{14}$$

and then assign the non-zero rows of these matrices as edges according to:

$$i \to j \in \hat{\mathcal{G}}_t \iff i \neq j \quad \text{and} \quad \|[\hat{\mathbf{R}}_{i,t}]_j\|_2 > \gamma \,. \tag{15}$$

Next, we find the transitive closure of $\hat{\mathcal{G}}_t$ as the estimated graph $\hat{\mathcal{G}}_t$. This procedure directly mirrors the graph property in Lemma 1, and recovers the transitive closure of $\mathcal{G}$ with high probability. We denote the parent set of node $i$ in $\hat{\mathcal{G}}_t$ by $\mathsf{pa}_t(i)$. In this rule, $\gamma > 0$ is a threshold for determining if a row of the estimated latent precision differences is zero. To ensure that this threshold can differentiate between true edges and non-edges, we need to select it carefully. We show that choosing it in instance-dependent interval $\gamma \in (0, \gamma^*)$ is necessary, where

$$\gamma^* \triangleq \min\{\|[\hat{\mathbf{R}}_{i,\infty}^Z]_j\|_2 \ : \ \|[\hat{\mathbf{R}}_{i,\infty}^Z]_j\|_2 \neq 0\} \,. \tag{16}$$

**1.b – Refined recovery under hard interventions.** When using hard interventions, the intervention target becomes independent of its non-descendants in the latent space. We impose this independence condition on the estimated latent variables by constructing minimum mean-squared-error (MMSE) estimates of a node's non-descendants in the estimated graph, which is equal to the transitive closure of the true graph with high probability. Specifically, we do the following.

(i) **Refined inverse transform $\mathbf{H}_t$.** Using the sample covariance matrices defined in (8), we first compute the estimated latent sample covariance matrices via

$$\hat{\Sigma}_{\mathbf{a},t}^Z = \mathbf{H}_t \cdot \Sigma_{\mathbf{a},t} \cdot \mathbf{H}_t^\top \,, \qquad \forall \mathbf{a} \in \mathcal{A}_0 \,. \tag{17}$$

We collect the MMSE estimates of node $i$ on $\mathsf{pa}_t(i)$ in environment $\{i\}$ in matrix $\mathbf{\Xi}_t \in \mathbb{R}^{n \times n}$ as

$$\mathbf{\Xi}_t[i, \mathsf{pa}_t(i)] \triangleq \hat{\Sigma}_{\{i\},t}^Z[i, \mathsf{pa}_t(i)] \cdot \left(\hat{\Sigma}_{\{i\},t}^Z[\mathsf{pa}_t(i), \mathsf{pa}_t(i)]\right)^{-1} \,. \tag{18}$$

Using these MMSE estimates, we update the estimate $\mathbf{H}_t$ as

$$\mathbf{H}_t \leftarrow (\mathbf{I} - \mathbf{\Xi}_t) \cdot \mathbf{H}_t \,. \tag{19}$$

(ii) **Variables refinement.** We re-apply the relation $\hat{\mathbf{Z}}_t = \mathbf{H}_t \mathbf{X}_t$. We show that this procedure refines the estimates $\hat{\mathbf{Z}}_t$ to be approximately equal to the true variables $\mathbf{Z}_t$ up to scaling.

(iii) **Graph refinement.** We refine the graph estimate $\hat{\mathcal{G}}_t$ by applying (15) using the updated $\mathbf{H}_t$. Since the variables are now recovered up to scaling by Lemma 1, this step returns the latent graph $\mathcal{G}$ exactly with high probability. Therefore, we *do not* take the transitive closure of $\hat{\mathcal{G}}_t$.

**2 – UCB-based selection rule.** After estimating the latent variables and the graph, we specify our selection rule. Formalizing this rule involves characterizing confidence ellipsoids of the relevant parameters under different interventions. To this end, we first estimate the graph parameters and subsequently use the confidence ellipsoids of these estimates to construct the confidence ellipsoid of the interventions, enabling a full description of the UCB-based selection of interventions.

**2.a – Parameter estimation.** We first estimate the SEM parameters $\mathbf{B}, \mathbf{B}^*$, and utility parameters $\theta$ using the estimated variables $\hat{\mathbf{Z}}_t$ and the estimated graph $\hat{\mathcal{G}}_t$. To keep the notation compact, we present the following algorithm and regret analysis as if the noise means after recovery are known, i.e., $\hat{\nu}_{\mathbf{a}}$ for $\mathbf{a} \in \mathcal{A}$ are known. This can be readily relaxed to accommodate unknown mean setting by using the same reparameterization technique as in [21], which involves adding a dummy node 1 to the graph estimates $\hat{\mathcal{G}}_t$, along with the associated modifications to $\hat{\mathbf{Z}}_t$ and $\mathsf{pa}_t(i)$.

As we use finite samples $\mathbf{X}_t$ to estimate $\mathbf{Z}_t$, there will inevitably be an estimation error. Therefore, at each time $t \in \mathbb{N}$, we design the weighted ridge regression estimators $\mathbf{A}_t$ and $\mathbf{A}_t^*$ at time $t$ for $\mathbf{B}$ and $\mathbf{B}^*$, respectively. To estimate the observational weights $[\mathbf{A}]_i$, we only use the samples in which node $i$ was not intervened. Conversely, to estimate the interventional weights $[\mathbf{A}^*]_i$, we use the samples in

which node $i$ was intervened. We encode these selection rules via *diagonal* weight matrices $\mathbf{W}_{i,t}$ and $\mathbf{W}_{i,t}^*$, which we will specify later. The $i$-th rows of the estimates $\mathbf{A}_t$ and $\mathbf{A}_t^*$ are specified as follows.

$$[\mathbf{A}_t]_i \triangleq [\mathbf{V}_{i,t}]^{-1}[\hat{\mathbf{Z}}_t]_{\mathsf{pa}_t(i)}^\top \mathbf{W}_{i,t}([\hat{\mathbf{Z}}_t]_i - \hat{\nu}[i]), \text{ and} [\mathbf{A}_t^*]_i \triangleq [\mathbf{V}_{i,t}^*]^{-1}[\hat{\mathbf{Z}}_t]_{\mathsf{pa}_t(i)}^\top \mathbf{W}_{i,t}^*([\hat{\mathbf{Z}}_t]_i - \hat{\nu}^*[i]) \quad (20)$$

where we have defined *weighted and doubly weighted Gram matrices* as

$$\mathbf{V}_{i,t} \triangleq [\mathbf{Z}_t]_{\mathsf{pa}_t(i)}^\top \mathbf{W}_{i,t}[\mathbf{Z}_t]_{\mathsf{pa}_t(i)} + \mathbf{I}_n , \quad \text{and} \quad \tilde{\mathbf{V}}_{i,t,s+1} \triangleq \mathbf{Z}_{i,t,s}^\top \mathbf{W}_{i,t}^2[:s,:s]\mathbf{Z}_{i,t,s} + \mathbf{I}_n , \quad (21)$$

$$\mathbf{V}_{i,t}^* \triangleq [\mathbf{Z}_t]_{\mathsf{pa}_t(i)}^\top \mathbf{W}_{i,t}^*[\mathbf{Z}_t]_{\mathsf{pa}_t(i)} + \mathbf{I}_n , \quad \text{and} \quad \tilde{\mathbf{V}}_{i,t,s+1}^* \triangleq \mathbf{Z}_{i,t,s}^\top \mathbf{W}_{i,t}^{*2}[:s,:s]\mathbf{Z}_{i,t,s} + \mathbf{I}_n , \quad (22)$$

where $\mathbf{Z}_{i,t,s} \triangleq \mathbf{Z}_t[\mathsf{pa}_t(i),:s]$ and $\mathbf{I}_n$ is the identity matrix. An important consideration in the above estimates is the design of the weight matrices. At each time $t$, we construct *diagonal matrices* $\{\mathbf{W}_{i,t}, \mathbf{W}_{i,t}^* : i \in [n]\}$ to softly filter out the outlier samples that are likely to have higher estimation error. The diagonal elements for $s \in [t]$ are defined as

$$\mathbf{W}_{i,t}[s,s] \triangleq \mathbb{1}\{i \notin \mathbf{a}_s\}\frac{1}{\zeta_t} \min\left\{1 , \|\hat{\mathbf{Z}}_t[\mathsf{pa}_t(i), s]\|_{[\tilde{\mathbf{V}}_{i,t,s}]^{-1}}^{-1}\right\} , \quad (23)$$

$$\text{and} \quad \mathbf{W}_{i,t}^*[s,s] \triangleq \mathbb{1}\{i \in \mathbf{a}_s\}\frac{1}{\zeta_t} \min\left\{1 , \|\hat{\mathbf{Z}}_t[\mathsf{pa}_t(i), s]\|_{[\tilde{\mathbf{V}}_{i,t,s}^*]^{-1}}^{-1}\right\} . \quad (24)$$

The designs of the diagonal weights are inversely proportional to $\zeta_t$, which is a bound on the cumulative estimation error. This design uses smaller weights when $\zeta_t$ is higher. The weights are also inversely proportional to a weighted $\ell_2$ norm, often referred to as the *weighted exploration bonus*. A higher exploration bonus means lower confidence in the sample and hence, lower weights.

Similarly, with $\mathbf{Z}_{\theta,t,s} \triangleq \mathbf{Z}_t[:,:s]$, we define the estimate for $\theta$ at time $t \in \mathbb{N}$ as follows.

$$\theta_t \triangleq [\mathbf{V}_{\theta,t}]^{-1}\hat{\mathbf{Z}}_t^\top \mathbf{W}_{\theta,t}\mathbf{U}_t , \qquad \mathbf{W}_{\theta,t}[s,s] \triangleq \frac{1}{\zeta_t} \min\left\{1, \|\hat{\mathbf{Z}}_t[:,s]\|_{[\tilde{\mathbf{V}}_{\theta,t,s}]^{-1}}^{-1}\right\} , \quad (25)$$

$$\mathbf{V}_{\theta,t} \triangleq \hat{\mathbf{Z}}_t^\top \mathbf{W}_{\theta,t}\hat{\mathbf{Z}}_t + \mathbf{I}_n , \qquad \tilde{\mathbf{V}}_{\theta,t,s} \triangleq \hat{\mathbf{Z}}_{\theta,t,s}^\top \mathbf{W}_{\theta,t}^2[:s,:s]\hat{\mathbf{Z}}_{\theta,t,s} + \mathbf{I}_n . \quad (26)$$

**2.b – Confidence ellipsoids and decision rule.** After estimating the SEM and utility parameters, we use a UCB-based rule for sequential selection of interventions. The UCB under intervention $\mathbf{a} \in \mathcal{A}$ is defined as the maximum value of expected utility when the weights are in the confidence ellipsoids of $\mathbf{a}$, denoted by $\mathcal{C}_{\mathbf{a},t}$. In order to construct the confidence ellipsoid for $\mathbf{a}$, we first form the following confidence ellipsoids for the estimated *parameters*, i.e., $\mathbf{A}_t$, $\mathbf{A}_t^*$ and $\theta_t$:

$$\mathcal{C}_{i,t} \triangleq \left\{\xi : \left\|\xi - [\mathbf{A}_t]_i\right\|_{\mathbf{V}_{i,t}[\tilde{\mathbf{V}}_{i,t,t}]^{-1}\mathbf{V}_{i,t}} \leq \beta_{i,t}(\delta_t)\right\} , \quad (27)$$

$$\mathcal{C}_{i,t}^* \triangleq \left\{\xi : \left\|\xi - [\mathbf{A}_t^*]_i\right\|_{\mathbf{V}_{i,t}^*[\tilde{\mathbf{V}}_{i,t,t}^*]^{-1}\mathbf{V}_{i,t}^*} \leq \beta_{i,t}(\delta_t)\right\} , \quad (28)$$

$$\text{and} \quad \mathcal{C}_{\theta,t} \triangleq \left\{\xi : \left\|\xi - \theta_t\right\|_{\mathbf{V}_{\theta,t}[\tilde{\mathbf{V}}_{\theta,t,t}]^{-1}\mathbf{V}_{\theta,t}} \leq \beta_t(\delta_t)\right\} , \quad (29)$$

where $\{\beta_{i,t}(\delta_t) \in \mathbb{R}_+, t \in \mathbb{N}, i \in [n]\}$ and $\{\beta_t(\delta_t) \in \mathbb{R}_+, t \in \mathbb{N}\}$ are sequences of confidence radii that control the size of confidence ellipsoids and $\delta_t$ is the tolerance of wrong estimates at time $t$. Accordingly, we define the relevant confidence ellipsoid for node $i$ under intervention $\mathbf{a} \in \mathcal{A}$ as

$$\mathcal{C}_{i,\mathbf{a},t} \triangleq \mathbb{1}\{i \in \mathbf{a}\}\,\mathcal{C}_{i,t}^* + \mathbb{1}\{i \notin \mathbf{a}\}\,\mathcal{C}_{i,t} , \quad \text{and} \quad \mathcal{C}_{\mathbf{a},t} \triangleq \{\cup_{i \in [n]}\mathcal{C}_{i,\mathbf{a},t}\} \cup \mathcal{C}_{\theta,t} . \quad (30)$$

Based on these, at time $t$, our algorithm selects the intervention that maximizes the UCB. Let $L_t$ be the length of the longest causal path of $\hat{\mathcal{G}}_t$. Due to the linear structure in SEMs, it is defined as

$$\text{UCB}_{\mathbf{a},t} \triangleq \max_{\{\tilde{\mathbf{A}},\tilde{\theta}\} \in \mathcal{C}_{\mathbf{a},t}} \left\langle \tilde{\theta} , \sum_{\ell=0}^{L_t} \tilde{\mathbf{A}}^\ell \cdot \nu_{\mathbf{a}}\right\rangle \; \forall \mathbf{a} \in \mathcal{A} , \quad \text{and} \quad \mathbf{a}_{t+1} = \arg\max_{\mathbf{a} \in \mathcal{A}} \text{UCB}_{\mathbf{a},t} . \quad (31)$$

**3 – Under-sampling rule.** RO-CRL iteratively update various estimates. The performance of the UCB step is contingent on the performance of the CRL step producing increasingly accurate estimates $\hat{\mathcal{G}}_t$ and $\mathbf{H}_t$. To ensure such accuracy, we impose a rule requiring that single-node interventions be sufficiently explored over time. Such interventions are needed to construct the necessary statistics for the CRL step. To formalize this, we define the set of under-explored interventions as

$$\mathcal{A}_t^{\mathsf{UE}} \triangleq \{\mathbf{a} \in \mathcal{A}_0 \mid N_{\mathbf{a},t} < f_t(\hat{\mathcal{G}}_t)\} , \quad (32)$$

where the function $f_t(\hat{\mathcal{G}}_t)$ is a non-decreasing term that controls the adaptive exploration to collect observations $\mathbf{X}_t$ when necessary. If $\mathcal{A}_t^{\mathsf{UE}} \neq \emptyset$, the algorithm is forced to random sample from $\mathcal{A}_t^{\mathsf{UE}}$.

# 4 Regret Analysis for RO-CRL

In this section, we present the node-level instance-dependent regret results for RO-CRL. We also present algorithm-independent regret lower bounds to complement RO-CRL's achievable regret.

As an intermediate step toward regret analysis, we first provide PAC identifiability guarantees for recovering $\mathcal{G}$ and $\mathbf{Z}_t$. These results depend on the least frequently selected single-node intervention up to time $t$, denoted by $s_t \triangleq \min_{\mathbf{a} \in \mathcal{A}_0} N_{\mathbf{a},t}$.

**Theorem 2** (Sample complexity). *For any instant $t$ of RO-CRL that satisfies $s_t \geq N(\epsilon, \delta)$, where*

$$N(\epsilon, \delta) \triangleq C^2 \max\{\epsilon^{-2}, \epsilon_{\max}^{-2}\} \left(d + \log(1/\delta)\right), \tag{33}$$

*under Assumption 1, the estimate $\mathbf{H}_t$ constructed in (11) (or (19)) and estimate $\hat{\mathcal{G}}_t$ constructed in (15) ensure $(\epsilon, \delta)$–PAC recovery of $\mathbf{Z}_t$ and $\mathcal{G}$ under soft (or hard) interventions specified in Definition 1. This implies that with probability at least $1 - \delta$, the error term $\mathbf{E}_t$ specified in Definition 1 under both hard and soft interventions satisfies*

$$\|\mathbf{E}_t\|_2^2 \leq C^2 \left(d + \log(1/\delta)\right)/s_t. \tag{34}$$

The regret bounds have delicate differences under hard and soft interventions captured by the constants $u \in \{u_\mathrm{S}, u_\mathrm{H}\}$. Let $\mathrm{m} \in \{\mathrm{S}, \mathrm{H}\}$ indicate the intervention type, and $\mathcal{H}_\mathrm{m}(i) \in \{\mathcal{H}_\mathrm{S}(i), \mathcal{H}_\mathrm{H}(i)\}$ such that $\mathcal{H}_\mathrm{H}(i) = \mathsf{pa}(i)$ and $\mathcal{H}_\mathrm{S}(i) = \mathsf{an}(i)$. Then $u$ is defined as

$$u_{\mathrm{m},i} = \begin{cases} 0 & \text{if } i \text{ is a root node} \\ \sum_{j \in \mathcal{H}_\mathrm{m}(i)} u_{\mathrm{m},j} + \sqrt{|\mathcal{H}_\mathrm{m}(i)|} & \text{otherwise} \end{cases}, \quad \text{and} \quad u_\mathrm{m} = \sum_{i=1}^{n} u_{\mathrm{m},i} + \sqrt{n}. \tag{35}$$

By setting $\delta_t = \frac{6\delta}{\pi^2 t^2}$ for confidence radii and controlling $T_0$ to satisfy $T_0 \geq N(\epsilon_{\max}, \delta_{nT_0})$, we ensure that with probability at least $1 - 4\delta$, the choice of $f_t$ defined in (32), $\zeta_t$ for weights design and $\beta$'s for confidence radii will be equal to or in the order of

$$f_t(\hat{\mathcal{G}}_t) = \max\{d^{\frac{1}{3}} n^{-\frac{2}{3}} u^{\frac{2}{3}} t^{\frac{2}{3}}, N(\epsilon_{\max}, \delta_t)\}, \qquad \zeta_t = \mathcal{O}\left(t\sqrt{\left(d + \log(1/\delta_t)\right)/f_t(\hat{\mathcal{G}}_t)}\right), \tag{36}$$

$$\beta_{i,t}(\delta_t) = \begin{cases} \tilde{\mathcal{O}}(\sqrt{|\mathsf{pa}(i)|}) & \text{Hard intervention} \\ \tilde{\mathcal{O}}(\sqrt{|\mathsf{an}(i)|}) & \text{Soft intervention} \end{cases}, \qquad \beta_t(\delta_t) = \tilde{\mathcal{O}}(\sqrt{n}), \tag{37}$$

where under soft intervention $u = u_\mathrm{S}$ and under soft intervention $u = u_\mathrm{H}$.

**Theorem 3** (Regret upper bound). *Under Assumptions 1–3, with probability at least $1 - 4\delta$, the average cumulative regret of RO-CRL is upper bounded by*

$$\mathcal{R}_T \leq \tilde{\mathcal{O}}\left(d^{\frac{1}{3}} n^{\frac{1}{3}} u^{\frac{2}{3}} T^{\frac{2}{3}} + u\sqrt{T}\right), \tag{38}$$

When the noise means $\hat{\nu}_\mathbf{a}$ for $\mathbf{a} \in \mathcal{A}$ are unknown, the impact on the regret upper bound order is reflected in the parameter $u$, where the value will be set to $1$ if $i$ is a root node instead of $0$.

Next, we establish a lower bound on the regret in the unknown mean setting. As no finite-sample lower bound for CRL is known in the literature, we provide the lower bound by fixing the estimation error as in Theorem 2. As the reward depends on how noise terms cumulatively contribute to the utility, the lower bound depends on the number of paths in the $\mathcal{G}$. Denote $m_{i,j}$ as the number of causal paths from node $i$ to node $j$ in $\mathcal{G}_\mathrm{tc}$, and denote the number of paths $p$ as $p \triangleq n + 1 + \sum_{i=1}^{n} \sum_{j=1}^{n} m_{i,j}$.

**Theorem 4** (Lower bound under soft intervention). *For any CRL-based algorithm that satisfies Theorem 2, there exist instances of a causal model on $\mathcal{G}_\mathrm{tc}$ and estimation error $\mathbf{E}_t$ such that the expected cumulative regret of any algorithm is at least*

$$\mathcal{R}_T \geq \Omega\left(d^{\frac{1}{3}} n^{\frac{1}{3}} p^{\frac{2}{3}} T^{\frac{2}{3}} + p\sqrt{T}\right). \tag{39}$$

**Remark 1** (Tightness or the bounds). *When comparing the upper bound in Theorem 3 and the lower bound in Theorem 4, we observe that both bounds show similar behavior with respect to graph-dependent parameters and the time horizon $T$. The only discrepancy comes from $u_\mathrm{S}$ and $p$, where $u_\mathrm{S}$ has an extra factor to count the dimension of graph connectivity in transitive closure.*

**Remark 2** (Causal bandits). *Our reward-oriented CRL reduces to the standard causal bandit setting if we set $\mathbf{G} = \mathbf{I}_n$ and the utility function to $U(Z) = Z[n]$. By these choices, our regret bounds immediately provide regret bounds for causal bandits. Specifically, our upper and lower bounds simplify to $\tilde{\mathcal{O}}(u_\mathrm{H}\sqrt{T})$ and $\Omega(p\sqrt{T})$, respectively, where $p$ is defined on $\mathcal{G}$ instead of $\mathcal{G}_\mathrm{tc}$. These bounds match at $\tilde{\Theta}(d_\mathcal{G}^L \sqrt{T})$ for graph-independent regret bounds, eliminating the gap $d$ in previous results [24].*

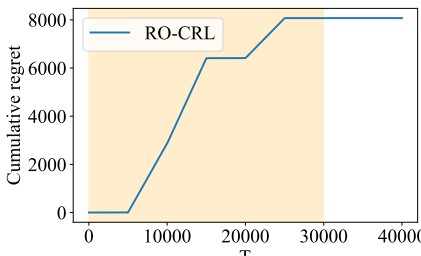 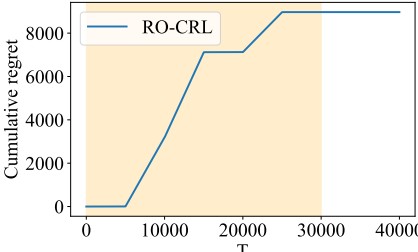

Figure 2: Results of regret of RO-CRL . *Left:* soft intervention. *Right:* hard intervention. Yellow shading denotes the exploration phase.

# 5  Experiments

In this section, we evaluate the empirical performance of RO-CRL. We report the regret of RO-CRL under both soft and hard interventions. Additional experiments, including CRL recovery, scaling behavior, comparison to baselines, and assumption violations, are deferred to Appendix C.[2]

**Latent graph.**  We generate a random acyclic graph on $n$ nodes by enforcing a strictly lower-triangular weight matrix. To ensure that the UCB in RO-CRL is computable for any randomly generated graph, we force all weights and noises to be positive (See Appendix B). Specifically, observational weights are drawn from $[0.25, 1]$ and interventional weights are set $0.1$ times the corresponding observational weights for soft interventions and to $0$ for hard interventions. Noise terms are sampled i.i.d. from the uniform distribution $U[0, 1]$. We set $n = 5$ for these experiments and repeat the experiments $50$ times.

Figure 2 illustrates the cumulative regret of RO-CRL under two types of interventions: soft interventions (left) and hard interventions (right). In both cases, we observe an initial phase of forced exploration (highlighted in yellow), during which the algorithm collects sufficient interventional data to estimate the underlying causal structure and variables. After this phase, the regret curves begin to flatten, indicating that RO-CRL effectively identifies near-optimal interventions and achieves sublinear regret.

# 6  Conclusion

In this paper, we have introduced the framework for formalizing and analyzing reward-oriented causal representation learning, the objective of which is to optimize a downstream task over the space of possible interventions in the causal system. The key difference between reward-oriented and conventional CRL is that CRL's objective of perfectly recovering the latent graph and variables can be excessive for optimizing a downstream task. Specifically, for reward-oriented CRL, one needs to learn the latent graph and variables at the coarsest level that enable the identification of optimal interventions. To resolve uncertainties associated with the latent causal graph and the latent probability distributions, we have adopted a sequential framework to explore different interventions and identify the optimal one with as few time instances as possible. We have designed an adaptive exploration algorithm that learns only the coarsest representation necessary to optimize the downstream reward. We have provided finite-sample latent recovery guarantees for the causal graph and its variables, and regret bounds for different types of interventions. In the standard causal bandit setting, these bounds simplify and match, even improve upon previous results.

## Acknowledgments and Disclosure of Funding

This work was supported in part by the Rensselaer-IBM Future of Computing Research Collaboration (FCRC).

---

[2]The codebase for the experiments can be found at `https://github.com/ZiruiYan/RO-CRL`.

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

# Reward-oriented Causal Representation Learning
# Supplementary Materials

## Table of Contents

# A  Pseudocode

---

**Algorithm 1** Reward-oriented CRL (RO-CRL)

---

 1: **Forced exploration.** Sample $T_0$ times for each intervention $\mathbf{a} \in \mathcal{A}_0$.
 2: **for** $t = (n+1)T_0, \ldots$ **do**
 3:      $\triangleright$ Latent recovery
 4:      Update the inverse transform estimate $\mathbf{H}_t$ via (11)
 5:      Estimate $\hat{\mathbf{Z}}_t$ according to $\hat{\mathbf{Z}}_t = \mathbf{H}_t \mathbf{X}_t$
 6:      Update the graph estimate $\hat{\mathcal{G}}_t$ via (15)
 7:      **if** hard interventions **then**
 8:          Update the inverse transform estimate again using (19)
 9:          Update $\hat{\mathbf{Z}}_t$ according to $\hat{\mathbf{Z}}_t = \mathbf{H}_t \mathbf{X}_t$
10:          Update the graph estimate again $\hat{\mathcal{G}}_t$ via (15)
11:      $\triangleright$ Under-sampling rule
12:      **if** $\mathcal{A}_t^{UE} \neq \emptyset$ **then**
13:          **Pull** $\mathbf{a}_t$ random sample from $\mathcal{A}_t^{UE}$
14:      **else**
15:          $\triangleright$ Parameter estimation
16:          Set weight matrix $\mathbf{W}_{i,s}$, $\mathbf{W}_{i,s}^*$ and $\mathbf{W}_{\theta,s}$ for $s \in [t]$ according to (23)–(25)
17:          Update $\mathbf{A}_t$, $\mathbf{A}_t^*$ and $\theta_t$ according to (20) and (26)
18:          Set $\mathbf{A}_t^* = \mathbf{0}$ under hard intervention
19:          $\triangleright$ UCB selection
20:          Compute $\mathrm{UCB}_{\mathbf{a},t}$ according to (31) for $\mathbf{a} \in \mathcal{A}$
21:          **Pull** $\mathbf{a}_{t+1} = \arg\max_{\mathbf{a} \in \mathcal{A}} \mathrm{UCB}_{\mathbf{a},t}$
22:      Observe $X_t$ and $U(Z_t)$

---

# B  Computational complexity of RO-CRL

The computational cost of RO-CRL per step can be broken down into two parts:

1. **CRL:** CRL routine depends on matrix inversions ($\mathcal{O}(d^3)$), which can be expensive for large $d$. However, since the transformation is linear, we can detect the supporting subspace and project the samples to it, effectively reducing the observation dimension $d$ to $n$, and yielding an overall per-step complexity of $\mathcal{O}(n^4)$.

2. **UCB-based selection:** The computational bottleneck is the intervention-selection step: the UCB is intractable on general causal graphs [26, Section 3.3]. To make the algorithm practical, [26] proposes techniques that efficiently compute an upper bound on the UCB, which we can incorporate into our approach. However, the upper bound is loose, and its robustness to imperfect latent recovery is unknown.

In the experiments, we use the following modification to ensure efficiency in the intervention selection step for RO-CRL.

**Non-negative edge weights and noise (shifted system):** We investigate the setting where all weights and noises are positive. A general system can be linearly transformed into such a non-negative system. A key property of such systems is that all variables are monotone in their parents. To calculate the UCB, we can therefore maximize nodes sequentially in causal order and use the closed form in linear bandits. The resulting complexity is $\mathcal{O}\big( \sum_{i \in [n]} |\mathsf{pa}_t(i)|^3 + n^3 \big)$ for UCB selection in (31), where $n$ is the number of latent dimensions.

# C  Additional Experiments

**Latent Recovery**    Figure 3 shows the variations of the graph recovery rate versus the sample size $s_t$. As expected, it is observed that the recovery improves with more samples $s_t$. Furthermore, the hard interventions, the stronger special case, consistently yield higher recovery rates than soft

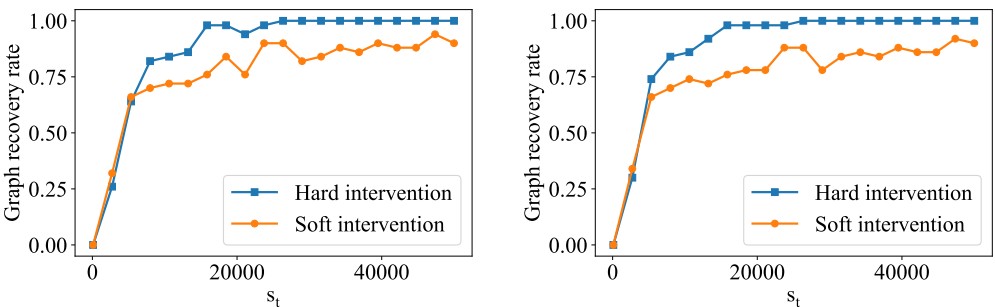

Figure 3: Results of graph recovery on different value of $s_t$ for soft intervention ($\mathcal{G}_{\mathrm{tc}}$) and hard intervention ($\mathcal{G}$). *Left:* $d = 10$. *Right:* $d = 75$.

interventions. Figures 4 and 5 show the norm of the error term $\mathbf{E}_t$ and the average estimation error on latent variables $Z$ for varying $s_t$. We can see both terms decay, which conforms to the theoretical decay $\sqrt{1/s_t}$ rate established in Theorem 2.

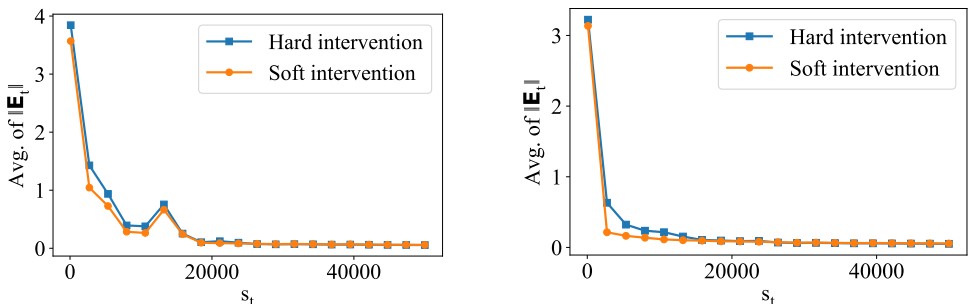

Figure 4: Results of the error term $\mathbf{E}_t$ on different value of $s_t$. *Left:* $d = 10$. *Right:* $d = 75$.

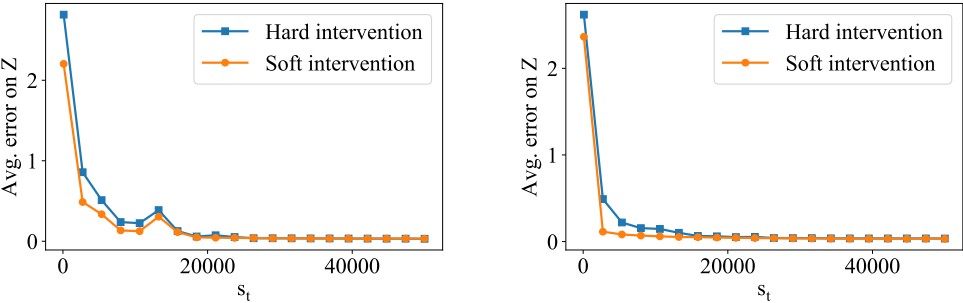

Figure 5: Results of average estimation error of $Z$ on different value of $s_t$. *Left:* $d = 10$. *Right:* $d = 75$.

**Performance of CRL when varying $n$ and $d$.** In Table 2, we provide additional experiments to evaluate the performance of CRL across different $n$, $d$, and intervention types, plotted against sample size $s_t$. These results show that the CRL performance of our RO-CRL algorithm is mostly consistent across variations in system parameters: Variable recovery errors decay with increasing number of samples $s_t$ consistently with the theoretical rate of $1/s_t$ (Theorem 2) across all settings. Observation dimension $d$ does not affect results thanks to a dimensionality reduction step. Finally, increasing $n$ from 5 to 8 does not lead to a significant drop in variable recovery performance. However, graph recovery performance significantly degrades for $n = 8$ under soft interventions, which can be explained by noting that the "true" graph under soft interventions is the transitive closure, which includes a high number of indirect edges. Such indirect effects are hard to track from the precision matrices, which makes the graph recovery for soft interventions more difficult.

Table 2: CRL performance versus $n$, $d$ and intervention type.

(a) Latent recovery (MSE).

| $n$ | $d$ | Int. type | $s_t$=1000 | 2000 | 4000 | 8000 | 16000 |
|---|---|---|---|---|---|---|---|
| 5 | 5 | Soft | 0.067 | 0.052 | 0.034 | 0.026 | 0.018 |
| 5 | 25 | Soft | 0.201 | 0.189 | 0.170 | 0.162 | 0.156 |
| 8 | 8 | Soft | 0.136 | 0.092 | 0.069 | 0.047 | 0.035 |
| 5 | 5 | Hard | 0.218 | 0.165 | 0.069 | 0.047 | 0.017 |
| 5 | 25 | Hard | 0.378 | 0.320 | 0.232 | 0.186 | 0.147 |
| 8 | 8 | Hard | 0.246 | 0.176 | 0.099 | 0.047 | 0.034 |

(b) Graph recovery rate.

| $n$ | $d$ | Int. type | $s_t$=1000 | 2000 | 4000 | 8000 | 16000 |
|---|---|---|---|---|---|---|---|
| 5 | 5 | Soft | 0.00 | 0.22 | 0.80 | 0.94 | 1.00 |
| 5 | 25 | Soft | 0.02 | 0.10 | 0.64 | 0.80 | 0.94 |
| 8 | 8 | Soft | 0.00 | 0.04 | 0.00 | 0.00 | 0.00 |
| 5 | 5 | Hard | 0.02 | 0.10 | 0.68 | 0.84 | 1.00 |
| 5 | 25 | Hard | 0.00 | 0.08 | 0.50 | 0.82 | 0.98 |
| 8 | 8 | Hard | 0.00 | 0.08 | 0.50 | 0.96 | 1.00 |

**Varying observation dimension $d$:**   Thanks to dimensionality reduction techniques and the forced-exploration schedule, CRL's regret remains essentially unchanged as we increase the feature dimension $d$ from 5 to 100 while keeping the latent dimension at $n = 5$. This observation is consistent with some previous findings in CRL under linear transformations [3] that the performance of CRL is not that sensitive to the feature dimension $d$, and this behavior does not contradict our lower bound analysis, which is derived under the error bounds from CRL. Table 3 reports the corresponding runtime for different $d$ at a fixed time horizon $T$, and the additional computational cost introduced by higher $d$ remains well within practical limits.

Table 3: Average runtime (in minutes) with varying $d$ when $n = 5$

| $d$ | 5 | 10 | 25 | 50 | 75 | 100 |
|---|---|---|---|---|---|---|
| soft | 48.55 | 47.19 | 53.85 | 48.01 | 48.11 | 48.55 |
| hard | 63.12 | 65.83 | 67.66 | 67.24 | 67.54 | 68.32 |

**Varying latent dimension $n$:**   (which also affects the $u_t$) for fixed $d = 10$. From Table 4, we observe that the cumulative regret grows by increasing $n$. We note the increase stems from two factors: (i) exploration cost for CRL varies with $n$, and (ii) the reward range becomes larger as $n$ increases. These trends will be more clearly visualized in the figures to be included in the final version.

Table 4: Cumulative regret with different latent dimensions $n$ when $d = 10$

| $n$ | 3 | 5 | 7 | 9 |
|---|---|---|---|---|
| soft | 1524 | 4832 | 13298 | 37787 |
| hard | 1789 | 5382 | 14677 | 37830 |

Table 5: Cumulative regret across different algorithms

| Algo | RO-CRL | UCB | modified RO-CRL |
|---|---|---|---|
| hard | 5382 | 15690 | 10132 |
| soft | 4832 | 16320 | 9726 |

**Baselines.** We compare RO-CRL with two baselines: vanilla UCB and a modified version of RO-CRL, where CRL is performed for a fixed fraction of the sample budget, after which the bandit proceeds without accounting for latent recovery error. Table 5 reports the cumulative regret at the given final horizon $T$ for different algorithms. As shown, both baselines perform significantly worse than RO-CRL. The performance gap between RO-CRL and Modified RO-CRL will continue to increase with longer time horizons.

**Assumption Violations.**

- **Assumption 1:** We run experiments where Assumption 1 is violated and provide the regret and variable recovery results in Tables 6–7. First, we note that graph recovery consistently fails (the graph recovery rate is exactly 0) in this setting. However, Table 6 shows that latent variable recovery still works: Indeed, its analysis is independent of Assumption 1. In Table 7, we evaluate cumulative regret versus time, and observe that the bandit algorithm does not achieve a sublinear regret after the forced exploration phase ($t \geq 18000$ in this case), meaning that the RO-CRL algorithm requires Assumption 1 to appropriately capture the true uncertainties and make good decisions.

Table 6: Cumulative regret under Assumption 1 violation ($n = d = 5$)

| $t$ | 3000 | 6000 | 9000 | 12000 | 15000 | 18000 | 21000 | 24000 |
|---|---|---|---|---|---|---|---|---|
| Regret | 0 | 2000 | 4000 | 4000 | 6000 | 6000 | 11000 | 16000 |

Table 7: Latent variable recovery MSE under Assumption 1 violation ($n = d = 5$)

| $s_t$ | 1000 | 2000 | 4000 | 8000 | 16000 |
|---|---|---|---|---|---|
| MSE | 0.134 | 0.123 | 0.104 | 0.097 | 0.092 |

- **Boundedness violations** We conduct experiments that deliberately violate two key assumptions: bounded noise and known parameter bounds. The specific violations are as follows: - Noise violation: We replace the bounded noise with unbounded Gaussian noise. - Parameter violation: Instead of properly rescaling the system, we multiply all parameters by a factor of 10 to simulate the violation. Note that such a scaling can be offset by properly scaling the latent variables. Therefore, in these experiments, we explicitly set a fixed scale of the latent variables.

  As shown in Table 8, both violations result in linearly growing regret after the forced exploration phase ($t \geq 18000$ in this case). This result aligns with theoretical expectations: violating the noise boundedness or known-parameter assumptions leads to confidence radii that no longer appropriately capture the true uncertainty, thereby causing increasingly poor decisions.

Table 8: Cumulative regret under bound violation ($n = d = 5$)

| $t$ | 6000 | 12000 | 15000 | 18000 | 19000 | 20000 | 21000 |
|---|---|---|---|---|---|---|---|
| noise | 1909 | 4272 | 5382 | 5382 | 7176 | 8970 | 10764 |
| parameters | 421447 | 505197 | 516290 | 516290 | 685092 | 857133 | 1029174 |

## D  Additional Notations

In this section, we present the notations that will be useful in our analyses.

Given a symmetric matrix $\mathbf{A} \in \mathbb{R}^{d \times d}$, we denote the vector of eigenvalues of $\mathbf{A}$ ordered in descending order by $\boldsymbol{\lambda}(\mathbf{A}) \in \mathbb{R}^d$ and the matrix of eigenvectors by $\mathbf{Q}(\mathbf{A}) \in \mathbb{R}^{d \times d}$ such that $\mathbf{A} = \mathbf{Q}(\mathbf{A}) \cdot \text{diag}(\boldsymbol{\lambda}(\mathbf{A})) \cdot \mathbf{Q}(\mathbf{A})^\top$.

Table 9: Table of Notations in Main Body

| Notation | Description |
|---|---|
| $Z$ | Latent variable |
| $\mathbf{B}, \mathbf{B}^*$ | Observational/Interventional weight matrices |
| $\varepsilon$ | noises |
| $\nu/\nu^*$ | mean of noises |
| $X$ | Observable variable |
| $U(Z) = \theta^\top Z + \varepsilon_U$ | Observable reward |
| $\mathbf{G}$ | Transforms |
| $Z[i], X[i]$ | $i$-th random variables |
| $Z_t, X_t$ | data vector at time $t$ |
| $\mathbf{a} \in \mathcal{A}$ | Interventions |
| $\mathbf{H}_t$ | Estimate of $\mathbf{G}^\dagger$ at time $t$ |
| $\hat{\mathbf{Z}}_t = \mathbf{H}_t X_t$ | Estimate of sample $\mathbf{Z}_t$ at time $t$ |
| $\mathcal{A}_0$ | Set of null and atomic intervention |
| $\mathbf{A}_t, \mathbf{A}_t^*$ | Estimates of $\mathbf{A}, \mathbf{A}^*$ at time $t$ |
| $\zeta_t$ | Cumulative estimation error at time $t$ |

## D.1 Precision Matrices

In this paper, we will consider *precision matrix differences* as our learning signal for CRL, similarly to [2]. The precision matrix of a distribution is the inverse of its covariance matrix. In case of the latent variables, $Z$, let us denote the pre-intervention precision matrix by $\Theta$. Note that the implicit linear SEM can also be written explicitly as

$$Z = (\mathbf{I} - \mathbf{B})^{-1} \cdot \varepsilon . \tag{40}$$

Let us denote the vector $\mathbf{v}$ as the variances of entries of $\varepsilon$, and $\mathbf{v}^*$ as its interventional counterpart $\varepsilon^*$. Using this formulation, the precision matrix is given by

$$\Theta^Z = (\mathbf{I} - \mathbf{B})^\top \cdot \mathrm{diag}(1/\mathbf{v}) \cdot (\mathbf{I} - \mathbf{B}) = \sum_{i \in [n]} (\mathbf{v}[i])^{-1} ([\mathbf{I} - \mathbf{B}]_i)^\top [\mathbf{I} - \mathbf{B}]_i . \tag{41}$$

Let us construct two matrices $\mathbf{K}, \mathbf{K}^* \in \mathbb{R}^{N \times N}$ row by row as

$$[\mathbf{K}]_i \triangleq (\mathbf{v}[i])^{-1/2} [\mathbf{I} - \mathbf{B}]_i , \qquad [\mathbf{K}^*]_i \triangleq (\mathbf{v}^*[i])^{-1/2} [\mathbf{I} - \mathbf{B}^*]_i , \tag{42}$$

such that term $i$ in (41) is equal to $([\mathbf{K}_i])^\top [\mathbf{K}]_i$. Since this is only a function of the generation mechanism of node $i$, the precision matrix of the latent variables under action $\mathbf{a} \in \mathcal{A}$, denoted by $\Theta_{\mathbf{a}}^Z$, is given by

$$\Theta_{\mathbf{a}}^Z = \sum_{i \notin \mathbf{a}} ([\mathbf{K}]_i)^\top [\mathbf{K}]_i + \sum_{i \in \mathbf{a}} ([\mathbf{K}^*]_i)^\top [\mathbf{K}^*]_i . \tag{43}$$

Therefore, $\mathbf{K}, \mathbf{K}^*$ fully parameterize the precision matrices of the latent variables under any action $\mathbf{a} \in \mathcal{A}$. Finally, note that the precision matrices of *observed* variables, denoted by $\Theta_{\mathbf{a}}$ for action $\mathbf{a} \in \mathcal{A}$, are given by

$$\Theta_{\mathbf{a}} = \mathbf{G}^{\dagger\top} \Theta_{\mathbf{a}}^Z \mathbf{G}^\dagger . \tag{44}$$

Thus, defining $\mathbf{Q} = \mathbf{K}\mathbf{G}^\dagger$ and $\mathbf{Q}^* = \mathbf{K}^*\mathbf{G}^\dagger$, both in $\mathbb{R}^{N \times D}$, the observed precision matrices are given by

$$\Theta_{\mathbf{a}} = \sum_{i \notin \mathbf{a}} ([\mathbf{Q}]_i)^\top [\mathbf{Q}]_i + \sum_{i \in \mathbf{a}} ([\mathbf{Q}^*]_i)^\top [\mathbf{Q}^*]_i . \tag{45}$$

Similarly to the latent case, $\mathbf{Q}$ and $\mathbf{Q}^*$ fully parameterize the observed precision matrices $\{\Theta_{\mathbf{a}} : \mathbf{a} \in \mathcal{A}\}$. In this paper, the CRL algorithms we design solely depend on estimates of $\mathbf{Q}$ and $\mathbf{Q}^*$.

## D.2 Causal Bandit Notations

We define the row of weights under intervention as

$$[\mathbf{B_a}]_i = \mathbb{1}\{i \in \mathbf{a}_t\} [\mathbf{B}^*]_i + \mathbb{1}\{i \notin \mathbf{a}_t\} [\mathbf{B}]_i . \tag{46}$$

First, we provide notations that are useful in our analyses. We denote the singular values of a matrix $\mathbf{A} \in \mathbb{R}^{M \times N}$, where $M \geq N$, by

$$\sigma_1(\mathbf{A}) \geq \sigma_2(\mathbf{A}) \geq \cdots \geq \sigma_N(\mathbf{A}) . \tag{47}$$

In the proofs, we often work with zero-padded vectors and corresponding matrices. As a result, the matrices that contain these vectors have non-trivial *null space* leading to zero singular values. In such cases, we use the *effective* smallest singular value that is non-zero. We denote the *effective* largest and smallest eigenvalues that correspond to effective dimensions of a positive semidefinite matrix $\mathbf{A}$ with rank $k$ by

$$\sigma_{\max}(\mathbf{A}) \triangleq \sigma_1(\mathbf{A}) , \quad \text{and} \quad \sigma_{\min}(\mathbf{A}) \triangleq \sigma_k(\mathbf{A}) . \tag{48}$$

For a square matrix $\mathbf{U} = \mathbf{A}\mathbf{A}^\top \in \mathbb{R}^{N \times N}$, we denote the *effective* largest and smallest eigenvalues by

$$\lambda_{\max}(\mathbf{U}) \triangleq \lambda_{\max}(\mathbf{A}\mathbf{A}^\top) = \sigma_{\max}^2(\mathbf{A}) , \tag{49}$$

$$\text{and} \quad \lambda_{\min}(\mathbf{U}) \triangleq \lambda_{\min}(\mathbf{A}\mathbf{A}^\top) = \sigma_{\min}^2(\mathbf{A}) . \tag{50}$$

Then we construct data matrices that are closely related to Gram matrices. At time $t \in \mathbb{N}$ and for any node $i \in [n]$, the data matrices $\mathbf{U}_{i,t} \in \mathbb{R}^{n \times t}$ and $\mathbf{U}_{i,t}^* \in \mathbb{R}^{n \times t}$ consist of the weighted observational and interventional data, respectively. Specifically, we define

$$\mathbf{U}_{i,t} \triangleq [\hat{\mathbf{Z}}_t]_{\mathsf{pa}_t(i)} \sqrt{\mathbf{W}_{i,t}} , \quad \text{and} \quad \left[\mathbf{U}_{i,t}^*\right]_s \triangleq [\hat{\mathbf{Z}}_t]_{\mathsf{pa}_t(i)} \sqrt{\mathbf{W}_{i,t}^*} , \tag{51}$$

where we denote $\sqrt{\mathbf{A}}$ as the square root of the matrix $\mathbf{A}$.

We denote the relevant data matrices for node $i \in [n]$ under intervention $\mathbf{a} \in \mathcal{A}$ by

$$\mathbf{U}_{i,\mathbf{a},t} \triangleq \mathbb{1}\{i \notin \mathbf{a}\}\mathbf{U}_{i,t} + \mathbb{1}\{i \in \mathbf{a}\}\mathbf{U}_{i,t}^* , \quad \text{and} \quad \mathbf{V}_{i,\mathbf{a},t} \triangleq \mathbb{1}\{i \notin \mathbf{a}\}\mathbf{V}_{i,t} + \mathbb{1}\{i \in \mathbf{a}\}\mathbf{V}_{i,t}^* . \tag{52}$$

Combining (51) and (52), we have

$$\mathbf{V}_{i,\mathbf{a},t} = \mathbf{U}_{i,\mathbf{a},t}\mathbf{U}_{i,\mathbf{a},t}^\top + \mathbf{I}_n . \tag{53}$$

Similarly we define the data matrices that are related to $\tilde{\mathbf{V}}_{i,\mathbf{a},t}$ as

$$\tilde{\mathbf{U}}_{i,t} \triangleq [\hat{\mathbf{Z}}_t]_{\mathsf{pa}_t(i)} \mathbf{W}_{i,t} , \quad \text{and} \quad \tilde{\mathbf{U}}_{i,t}^* \triangleq [\hat{\mathbf{Z}}_t]_{\mathsf{pa}_t(i)} \mathbf{W}_{i,t}^* . \tag{54}$$

The relevant data matrices for node $i \in [n]$ under intervention $\mathbf{a} \in \mathcal{A}$ are

$$\tilde{\mathbf{U}}_{i,\mathbf{a},t} \triangleq \mathbb{1}\{i \notin \mathbf{a}\}\tilde{\mathbf{U}}_{i,t} + \mathbb{1}\{i \in \mathbf{a}\}\tilde{\mathbf{U}}_{i,t}^* , \tag{55}$$

$$\text{and} \quad \tilde{\mathbf{V}}_{i,\mathbf{a},t} = \tilde{\mathbf{U}}_{i,\mathbf{a},t}\tilde{\mathbf{U}}_{i,\mathbf{a},t}^\top + \mathbf{I}_n . \tag{56}$$

Define $N_{i,t}^*$ as the number of times that node $i \in [n]$ is intervened, and $N_{i,t}$ as its complement, i.e.,

$$N_{i,t}^* \triangleq \sum_{s=1}^{t} \mathbb{1}\{i \in \mathbf{a}_s\} , \quad \text{and} \quad N_{i,t} \triangleq t - N_{i,t}^* . \tag{57}$$

Accordingly, for any $i \in [n]$, $t \in \mathbb{N}$ and $\mathbf{a} \in \mathcal{A}$, define

$$N_{i,\mathbf{a},t} \triangleq \mathbb{1}\{i \in \mathbf{a}\}N_{i,t}^* + \mathbb{1}\{i \notin \mathbf{a}\}N_{i,t} , \tag{58}$$

To proceed, we define the second-moment matrices and their *effective* largest and smallest eigenvalues as

$$\Sigma_{i,\mathbf{a},t} \triangleq \mathbb{E}_{\mathbf{a}} \hat{Z}[\mathsf{pa}_t(i)] \, \hat{Z}[\mathsf{pa}_t(i)]^\top , \tag{59}$$

$$\kappa_{\min,t} \triangleq \min_{i \in [n], \mathbf{a} \in \mathcal{A}} \sigma_{\min}(\Sigma_{i,\mathbf{a},t}) , \tag{60}$$

$$\kappa_{\max,t} \triangleq \max_{i \in [n], \mathbf{a} \in \mathcal{A}} \sigma_{\max}(\Sigma_{i,\mathbf{a},t}) , \tag{61}$$

where $\kappa_{\min} > 0$ is guaranteed since there is no deterministic relation between nodes and their patients. These variables are inherent to the system and remain unknown to the learner.

Lastly, we define $\tilde{\mathbf{A}}_{\mathbf{a},t}$ and $\tilde{\theta}_{\mathbf{a},t}$ as the weights that attains $\mathrm{UCB}_{\mathbf{a},t}$, i.e.,

$$(\tilde{\mathbf{A}}_{\mathbf{a},t}, \tilde{\theta}_{\mathbf{a},t}) = \underset{\{\tilde{\mathbf{A}}, \tilde{\theta}\} \in \mathcal{C}_{\mathbf{a}_{t+1}, t}}{\arg\max} \left\langle \tilde{\theta} , \sum_{\ell=0}^{L_t} \tilde{\mathbf{A}}^{\ell} \cdot \hat{\nu}_{\mathbf{a}_{t+1}} \right\rangle . \tag{62}$$

Accordingly, we define the auxiliary variable $\tilde{Z}_t$ generated according to the following SEM

$$\tilde{Z}_t = \tilde{\mathbf{A}}_{\mathbf{a}_t,t} \tilde{Z}_t + \varepsilon_t , \tag{63}$$

with the exact same realization of the noise $\varepsilon_t$ and the UCB is calculated as

$$\tilde{U}(\tilde{Z}_t) \triangleq \mathrm{UCB}_{\mathbf{a},t} = \tilde{\theta}_{\mathbf{a},t}^{\top} \tilde{Z}_t + \epsilon_{U,t} . \tag{64}$$

# E Analysis of CRL Steps

In this section, we first prove the high probability error bounds for recovering the latent variables and graph using finite samples. For notational clarity, we drop $\infty$ from infinite-sample estimators and use these forms without a $t$ subscript as infinite sample limits.

## E.1 Proof of Infinite-sample Guarantees

In the infinite sample limit $s_t \to \infty$, covariance and precision matrix estimates converge to their true values. This enables us to derive identifiability guarantees without additional error terms. First, let us start by proving the main motivating property of precision differences.

**Proof of Lemma 1.** Using the expansion (43), we note that the precision difference between null and single-node interventional distributions is very structured.

$$\mathbf{R}_i^Z \triangleq \Theta_{\emptyset}^Z - \Theta_{\{i\}}^Z = ([\mathbf{K}]_i)^{\top}[\mathbf{K}]_i - ([\mathbf{K}^*]_i)^{\top}[\mathbf{K}^*]_i . \tag{65}$$

Using the definition of $\mathbf{K}, \mathbf{K}^*$ in (42), row $i$ of either of these matrices have non-zero entries only in coordinates $i$ or $\mathsf{pa}(i)$. Therefore, only the principal submatrix at coordinates $(\{i\} \cup \mathsf{pa}(i), \{i\} \cup \mathsf{pa}(i))$ can be non-zero in $\mathbf{R}_i^Z$, therefore, for any row $j$, the norm $\|[\mathbf{R}_i^Z]_j\|_2 \neq 0$ implies $j \in \{i\} \cup \mathsf{pa}(i)$.

Using this fact, we can prove the second part of the lemma. Since $\mathbf{R}_i = (\mathbf{G}^{\dagger})^{\top} \mathbf{R}_i^Z (\mathbf{G}^{\dagger})$, row sparsity of $\mathbf{R}_i^Z$ immediately implies that

$$\mathsf{col}(\mathbf{R}_i) \subseteq \mathsf{col}((\mathbf{G}^{\dagger})^{\top} \mathbf{R}_i^Z) \subseteq \mathsf{span}\{[\mathbf{G}^{\dagger}]_j : j \in \{i\} \cup \mathsf{pa}(i)\} . \tag{66}$$

**Rank-2 assumption.** Next, let us investigate $\mathbf{R}_i^Z$ and $\mathbf{R}_i$ under Assumption 1. This assumption simply ensures that $[\mathbf{K}]_i$ and $[\mathbf{K}^*]_i$ have different directions for all $i \in [n]$ that is not the root node, which implies the following.

**Lemma 2.** *Under Assumption 1, the precision matrix difference $\mathbf{R}_i^Z$ has rank 1 if and only if $i$ is a root node, and has rank 2 otherwise. The rank of $\mathbf{R}_i$ is equal to the rank of $\mathbf{R}_i^Z$.*

*Proof:* Assumption 1, that is, $[\mathbf{B}]_i \neq [\mathbf{B}^*]_i$ is used directly with the definitions of $[\mathbf{K}]_i = (\mathbf{v}[i])^{-1/2}[\mathbf{I} - \mathbf{B}]_i$, and $[\mathbf{K}^*]_i = (\mathbf{v}^*[i])^{-1/2}[\mathbf{I} - \mathbf{B}^*]_i$. If $i$ is a root node, both $[\mathbf{B}]_i = [\mathbf{B}^*]_i = 0$, but we must change the noise variance, which yields a rank-1 update to the matrix, with only the $(i,i)$-th element being non-zero. If $i$ *not* a root node, then Assumption 1 ensures that $\mathbf{e}_i - [\mathbf{B}]_i$ and $\mathbf{e}_i - [\mathbf{B}^*]_i$ have different directions, thus the overall difference of outer products yields a rank-2 matrix. ∎

**Soft interventions.** Proof of the soft intervention inverse transform estimation results are identical to [3, Lemma 4]. For graph estimation, we follow closely to proof of [3, Lemma 5] as follows.

In the infinite sample regime, using soft interventions, we recover the latent variables up to mixing with parent nodes, i.e.,

$$\hat{\mathbf{Z}} = \mathbf{H}\mathbf{X} = \mathbf{C}_{\mathrm{S}}\mathbf{Z} , \tag{67}$$

where $\mathbf{C}_{\mathrm{S}} = \mathbf{H}\mathbf{G}$ with $\mathbf{C}_{\mathrm{S}}[i,j] \neq 0$ only if $j \in \{i\} \cup \mathsf{pa}(i)$. The "estimated latent" precision matrix difference is computed via

$$\hat{\mathbf{R}}_i^Z = [\mathbf{H}]^{\dagger\top} \mathbf{R}_i [\mathbf{H}]^{\dagger} = [\mathbf{C}_{\mathrm{S}}]^{-\top} \mathbf{R}_i^Z [\mathbf{C}_{\mathrm{S}}]^{-1} . \tag{68}$$

Using [3, Lemma 13], the up-to-parents mixing in $\mathbf{C}_S$ results in a up-to-descendants mixing in $[\mathbf{C}_S]^{-\top}$, that is, $[\mathbf{C}_S]^{-\top}[i,j] \neq 0$ only if $j \in \{i\} \cup \mathsf{de}(i)$. On the other hand, using Lemma 1, only non-zero rows of $\mathbf{R}_i^Z$ are $j \in \{i\} \cup \mathsf{pa}(i)$. This implies that all rows and columns $j \notin \{i\} \cup \mathsf{an}(i)$ will be zero in $\hat{\mathbf{R}}_i^Z$. Therefore, the estimated graph $\hat{\mathcal{G}}$ is at worst a supergraph of the transitive closure of the true graph $\mathcal{G}$.

**Hard interventions.**    Proof of the hard intervention post-processing steps in the infinite sample regime is identical to [3, Proof of Theorem 3].

## E.2    Proof of Finite-sample Guarantees

In this section, we start by defining the error bounds for sample covariance and precision matrix estimation [27].

**Lemma 3.** *With probability $1 - \delta$, the maximum error term in sample covariance matrices $\Sigma_{\mathbf{a},t}$ for $\mathbf{a} \in \mathcal{A}_0$ is bounded by*

$$\max_{\mathbf{a} \in \mathcal{A}_0} \|\Sigma_{\mathbf{a},t} - \Sigma_{\mathbf{a}}\| \leq C_\Sigma \cdot \left( \frac{d + \log(1/\delta)}{s_t} \right)^{1/2} . \tag{69}$$

*Similarly, for a bounded condition number for covariance matrices, the maximum error in the sample precision difference matrices $\mathbf{R}_i$ for $i \in [n]$ is upper bounded by*

$$\max_{i \in [n]} \|\mathbf{R}_{i,t} - \mathbf{R}_i\| \leq C_{\mathbf{R}} \cdot \left( \frac{d + \log(1/\delta)}{s_t} \right)^{1/2} . \tag{70}$$

The proof strategy for finite-sample guarantees follows closely to [4]: We first show that infinite-sample guarantees can be recovered with low error if the estimation errors are low enough. Next, we use the error bounds of the specific precision difference estimator to derive overall error bounds for the overall CRL procedure.

**Inverse transform estimation.**    In inverse transform estimation, the estimate $\mathbf{H}_t$ is constructed row by row as the principal eigenvectors from finite-sample precision differences $\mathbf{R}_{i,t}$. We have the following result on the stability of this estimation procedure.

**Lemma 4.** *Denote the principal eigenvector of $\mathbf{R}_i$ by $\mathbf{H}_i$ and that of $\mathbf{R}_{i,t}$ as $\mathbf{H}_{i,t}$. Denote the minimum separation of the top two eigenvalues of $\mathbf{R}_i$ by $\eta^*$, that is,*

$$\eta^* = \min_{i \in [n]} \boldsymbol{\lambda}(\mathbf{R}_i)_1 - \boldsymbol{\lambda}(\mathbf{R}_i)_2 . \tag{71}$$

*Using Davis–Kahan symmetric $\sin \theta$ theorem [28], when $\|\mathbf{R}_i - \mathbf{R}_{i,t}\| \leq \eta^*/2$, we have*

$$\|\mathbf{H}_{i,t}^\top \mathbf{H}_{i,t} - \mathbf{H}_i^\top \mathbf{H}_i\| \leq \frac{2}{\eta^*} \|\mathbf{R}_i - \mathbf{R}_{i,t}\| . \tag{72}$$

Following [4, Lemma 17], the inverse transform estimation is upper bounded by

$$\|\mathbf{H}_t - \mathbf{H}\| \leq \frac{2\sqrt{n}}{\eta^*} \max_{i \in [n]} \|\mathbf{R}_i - \mathbf{R}_{i,t}\| , \tag{73}$$

and the overall transformation error $\mathbf{E}_t$ in Definition 1 is upper bounded by

$$\|\mathbf{E}_t\| \leq \frac{2\sqrt{n}}{\eta^*} \max_{i \in [n]} \|\mathbf{R}_i - \mathbf{R}_{i,t}\| . \tag{74}$$

**Graph estimation.**    The graph estimation procedure consists of two steps: Computing the estimated latent precision difference matrices and thresholding them to recover edges. Let's first focus on the error bounds on the computation side, which then yield error upper bounds in order to achieve perfect graph recovery.

The equation for computing the estimated latent precision differences is

$$\hat{\mathbf{R}}_i^Z = \mathbf{H}^{\dagger\top} \mathbf{R}_i \mathbf{H}^\dagger , \tag{75}$$

and similarly for finite-sample counterparts. A perturbative error bound on the pseudoinverse term is given by, when $\|\mathbf{H}_t - \mathbf{H}\| \leq 1/(2\|\mathbf{H}^\dagger\|)$,

$$\|\mathbf{H}_t^\dagger - \mathbf{H}^\dagger\| \leq 4\|\mathbf{H}^\dagger\|^2\|\mathbf{H}_t - \mathbf{H}\| . \tag{76}$$

Therefore, a bound on the estimation error in $\hat{\mathbf{R}}_{i,t}^Z$ is given by, when $\|\mathbf{H}_t^\dagger - \mathbf{H}^\dagger\| \leq \|\mathbf{H}^\dagger\|$,

$$\|\hat{\mathbf{R}}_{i,t}^Z - \hat{\mathbf{R}}_i^Z\| \leq 4\|\mathbf{H}^\dagger\|(\|\mathbf{H}^\dagger\|\|\mathbf{R}_{i,t} - \mathbf{R}_i\| + \|\mathbf{H}_t^\dagger - \mathbf{H}^\dagger\|\|\mathbf{R}_i\|) . \tag{77}$$

The spectral norm bound is a natural upper bound on the $\ell_2$ norm of any row of a matrix. In other words, under the same circumstances, we have, for any $j \in [n]$,

$$\|(\hat{\mathbf{R}}_{i,t}^Z - \hat{\mathbf{R}}_i^Z)_j\| \leq 4\|\mathbf{H}^\dagger\|(\|\mathbf{H}^\dagger\|\|\mathbf{R}_{i,t} - \mathbf{R}_i\| + \|\mathbf{H}_t^\dagger - \mathbf{H}^\dagger\|\|\mathbf{R}_i\|) . \tag{78}$$

Since the minimum nonzero entry of $\hat{\mathbf{R}}_i^Z$ is $\gamma^*$, it suffices for the error to be below

$$\max_{i\in[n]} 4\|\mathbf{H}^\dagger\|(\|\mathbf{H}^\dagger\|\|\mathbf{R}_{i,t} - \mathbf{R}_i\| + \|\mathbf{H}_t^\dagger - \mathbf{H}^\dagger\|\|\mathbf{R}_i\|) \leq \gamma^*/2 \tag{79}$$

to estimate all the edges in $\hat{\mathcal{G}}$ correctly, i.e., to ensure $\hat{\mathcal{G}}_t = \hat{\mathcal{G}}$. By shifting constants, this means

$$\max_{i\in[n]} \|\mathbf{R}_{i,t} - \mathbf{R}_i\| \lesssim \frac{\eta^*\gamma^*}{\|\mathbf{H}^\dagger\|^2} \tag{80}$$

is sufficient for (i) correct (soft) graph recovery, and (ii) for the error bound in (74) to hold.

**Hard interventions.**   For the post-processing in hard interventions, we use the linear minimum mean square error estimator, which is defined via a linear algebraic equation

$$\mathbf{\Xi}_t[i, \hat{\mathrm{pa}}_t(i)] = (\hat{\Sigma}_{\{i\},t}^Z[i, \hat{\mathrm{pa}}_t(i)]) \cdot (\hat{\Sigma}_{\{i\},t}^Z[\hat{\mathrm{pa}}_t(i), \hat{\mathrm{pa}}_t(i)])^{-1} . \tag{81}$$

This part has three possible sources of error: Estimation of $\mathbf{H}_t/\hat{Z}_t$, finite sample estimation of covariance matrices, and incorrect graph estimation. Since with bounded error the graph will be correctly estimated, we focus on the other two. Specifically, $\hat{\Sigma}_{\{i\},t}^Z$ is actually defined via

$$\hat{\Sigma}_{\{i\},t}^Z = \mathbf{H}_t\Sigma_{\{i\},t}\mathbf{H}_t^\top , \tag{82}$$

which, whenever $\|\Sigma_{\{i\},t} - \Sigma_{\{i\}}\| \leq \|\Sigma_{\{i\}}\|$, has an error upper bound

$$\|\hat{\Sigma}_{\{i\},t}^Z - \hat{\Sigma}_{\{i\}}^Z\| \leq 4\|\mathbf{H}\|(\|\Sigma_{\{i\}}\|\|\mathbf{H}_t - \mathbf{H}\| + \|\mathbf{H}\|\|\Sigma_{\{i\},t} - \Sigma_{\{i\}}\|) . \tag{83}$$

Given that $(\hat{\Sigma}_{\{i\}}^Z[\hat{\mathrm{pa}}(i), \hat{\mathrm{pa}}(i)])$ has bounded condition number, collecting constants, we get

$$\mathbf{\Xi}_t[i, \hat{\mathrm{pa}}_t(i)] \lesssim \max_{i\in[n]} \|\mathbf{R}_{i,t} - \mathbf{R}_i\| , \tag{84}$$

and therefore, after update $\mathbf{H}_t \leftarrow (\mathbf{I}_n - \mathbf{\Xi}_t)\mathbf{H}_t$, the inverse transform estimate error becomes

$$\|\mathbf{E}_t\| \lesssim \max_{i\in[n]} \|\mathbf{R}_{i,t} - \mathbf{R}_i\| . \tag{85}$$

If we use the error bound in Lemma 3, with probability $1 - \delta$, we have

$$\|\mathbf{E}_t\| \lesssim \left(\frac{d + \log(1/\delta)}{s_t}\right)^{1/2} , \tag{86}$$

Since the order of the error did not change, the required error level for graph recovery also remains in the same order.

In summary, ensuring that the precision difference estimation error is upper-bounded by

$$\max_{i\in[n]} \|\mathbf{R}_{i,t} - \mathbf{R}_i\| \lesssim \frac{\eta^*\gamma^*}{\|\mathbf{H}^\dagger\|^2} \tag{87}$$

ensures that (i) the graph can be correctly identified, and (ii) the variables to be recovered with error term in the same order as the precision difference errors, under both hard and soft interventions. This means that we can simply unify the constants by choosing the worst case, and use Lemma 3 to prove Theorem 2, which is restated here for the sake of completeness.

**Theorem 5** (Sample complexity). *For any instant $t$ of RO-CRL that satisfies $s_t \geq N(\epsilon, \delta)$, where*

$$N(\epsilon, \delta) \triangleq C^2 \max\{\epsilon^{-2}, \epsilon_{\max}^{-2}\} \left(d + \log 1/\delta\right), \tag{88}$$

*under Assumption 1, the estimate $\mathbf{H}_t$ constructed in (11) (or (19)) and estimate $\hat{\mathcal{G}}_t$ constructed in (15) ensure $(\epsilon, \delta)$–PAC recovery of $\mathbf{Z}_t$ and $\mathcal{G}$ under soft (or hard) interventions specified in Definition 1. This implies that with probability at least $1 - \delta$, the error term $\mathbf{E}_t$ specified in Definition 1 under both hard and soft interventions satisfies*

$$\|\mathbf{E}_t\|_2^2 \leq C^2 \left(d + \log(1/\delta)\right)/s_t . \tag{89}$$

*Proof:* Note that many intermediary results require the precision difference estimation errors to be upper-bounded. As such, we provide error bounds where the error is required to be below a certain threshold. Then, the error bounds in Lemma 3 can be mapped to sample complexity statements to be used in our setting: With probability $1 - \delta$, the maximum error term in the sample covariance and precision matrices is bounded by $\epsilon$ if the instance satisfies

$$s_t \geq C_{\mathrm{sc}} \max\{\epsilon^{-2}, \epsilon_{\max}^{-2}\}(d + \log(1/\delta)) \tag{90}$$

That is, if $s_t$ satisfies the above, for any $(\epsilon, \delta)$, the following error bounds hold with probability at least $1 - \delta$.

$$\|\mathbf{R}_{i,t} - \mathbf{R}_i\| \leq \epsilon, \quad \|\Sigma_{i,t} - \Sigma_i\| \leq \epsilon . \tag{91}$$

Then, given that $\epsilon_{\max}$ corresponds to the maximum tolerable error level of analysis provided in this section, we get the error bounds and sample complexity statements for the CRL objectives provided in the theorem statement. ∎

## F  Proof of Regret Upper Bound (Theorem 3)

In this section, we prove the regret upper bound in Theorem 3 in three parts. For simplicity in notations, we provide the analysis for the setting in which the noise mean is known. The regret bounds for the more realistic setting in which the noise mean is unknown follow the same steps in a straightforward way by padding a dummy node to the graph and latent variables and noises, introducing one extra degree of freedom.

**Part 1.** we provide a decomposition of node-level utilities in Section F.2, which provides the intuition behind our UCB construction in (31).

**Part 2.** Then we establish a more general bound than Theorem 3 as Theorem 6, which holds under mild under-sampling conditions. Its proof consists of four steps:

1. We show the system and its estimation error remain bounded in Section F.1, and there is a transformed linear SEM that $\hat{\mathbf{Z}}_\infty$ satisfies.

2. We show high probability ellipsoidal confidence sets for the parameter estimates $\mathbf{A}_t$, $\mathbf{A}_t^*$ and $\theta_t$ in Section F.3.

3. We quantify how these uncertainties propagate along the causal paths in Section F.4.

4. Combining the above, we provide the high probability regret bounds in Section F.5.

**Part 3.** Finally, leveraging Theorem 6 in **Part 2**, we complete the proof of Theorem 3 by choosing an appropriate $f_t(\hat{\mathcal{G}}_t)$ to balance exploration and exploitation.

We will first prove all parts under *known transformed mean setting*, that is $\hat{\nu}_{\mathbf{a}}$ for $\mathbf{a} \in \mathcal{A}$, and then discuss how it generalizes to *unknown transformed mean* setting. We emphasize that, although our proof follows the high-level structure of [23], it departs crucially in how we define the utility function and account for uncertainty in both $Z[\mathsf{pa}[i]]$ and $Z[i]$, a direct use of their formulation leads to suboptimal bounds. Moreover, we derive instance-dependent regret upper bounds that generalize those in [23]. Such refined bounds are essential in the transitive closure setting, where the maximum in-degree can be of the same order as $n$.

### F.1 Bounded System and Error Bound

#### F.1.1 Known Transformed Mean

**Transformed SEM.** As CRL, even in the infinite sample regime, only recovers the variables up to scaling, the estimates $\hat{\mathbf{Z}}_\infty$ do not obey the original SEM $Z = \mathbf{B}Z + \varepsilon$. Instead, we obtain the following system.

**Lemma 5.** *As a result of Theorem 1, the estimated latent variables $\hat{\mathbf{Z}}_\infty$ are Markov with respect to the estimated graph $\hat{\mathcal{G}}_\infty$. Specifically, there exist weight matrices $\mathbf{A}, \mathbf{A}^* \in \mathbb{R}^{n \times n}$ such that*

$$\mathbf{A} = \mathbf{I}_n - \Lambda(\mathbf{I}_n - \mathbf{B})\mathbf{C}^{-1} , \quad and \quad \mathbf{A}^* = \mathbf{I}_n - \Lambda(\mathbf{I}_n - \mathbf{B}^*)\mathbf{C}^{-1} , \tag{92}$$

*where $\Lambda$ is a diagonal matrix chosen so that $\mathbf{A}$ and $\mathbf{A}^*$ share the same support as $\mathbf{B}$ and $\mathbf{B}^*$. As a result, $\mathbf{A}[i,j]$ and $\mathbf{A}^*[i,j]$ are non-zero only if $j \in \mathsf{pa}_\infty(i)$. Finally, there exist exogenous-noise vectors $\hat{\varepsilon}_\infty, \hat{\varepsilon}_\infty^*$ (with independent entries) such that $\hat{\mathbf{Z}}_\infty$ follows a linear SEM. For any time $t \in [\mathbb{N}]$ and $i \notin \mathbf{a}_t$, the estimated variables follows the following linear SEM*

$$\hat{\mathbf{Z}}_\infty[i,t] = [\mathbf{A}]_i \cdot \hat{\mathbf{Z}}_\infty[:,t] + \hat{\varepsilon}_\infty[i,t] , \tag{93}$$

*Similarly, $i \in \mathbf{a}_t$ alters the generating mechanism for node $i$ to*

$$\hat{\mathbf{Z}}_\infty[i,t] = [\mathbf{A}^*]_i \hat{\mathbf{Z}}_\infty[i,t] + \hat{\varepsilon}_\infty[i,t] , \tag{94}$$

*where we have at each time $t \in \mathbb{N}$, the mean of the noises satisfies $\mathbb{E}[\hat{\varepsilon}_\infty[:,t]] = \hat{\nu}_t$ under observation and $\mathbb{E}[\hat{\varepsilon}_\infty[:,t]] = \hat{\nu}_t^*$ under intervention. And we use the term*

$$\nu_{\mathbf{a}}[i] = \mathbb{1}\{i \in \mathbf{a}\} \nu^*[i] + \mathbb{1}\{i \notin \mathbf{a}_t\} \nu[i] . \tag{95}$$

*Proof:* Similar to (40), we know SEM follows

$$Z = (\mathbf{I}_n - \mathbf{B})^{-1}\varepsilon . \tag{96}$$

So under the infinite sample estimate and under $t \in \mathbb{N}$ with $i \notin \mathbf{a}_t$, we have the following relations

$$\hat{\mathbf{Z}}_\infty[:,t] = \mathbf{H}_\infty \mathbf{X}_\infty[:,t] \tag{97}$$
$$= \mathbf{C}\mathbf{Z}_\infty[:,t] \tag{98}$$
$$= \mathbf{C}(\mathbf{I}_n - \mathbf{B})^{-1}\varepsilon \tag{99}$$
$$\triangleq (\mathbf{I}_n - \mathbf{A})^{-1}\Lambda\varepsilon . \tag{100}$$

So that we have the relation

$$\mathbf{I}_n - \mathbf{A} = \Lambda(\mathbf{I}_n - \mathbf{B})\mathbf{C}^{-1} . \tag{101}$$

Rearrange (101) that we get

$$\mathbf{A} = \mathbf{I}_n - \Lambda(\mathbf{I}_n - \mathbf{B})\mathbf{C}^{-1} . \tag{102}$$

And we know that $\hat{\varepsilon} = \Lambda\varepsilon$. Similarly, the result holds for $\mathbf{A}^*$ under $i \in \mathbf{a}_t$. ∎

Similarly, the utility can be calculated as

$$U(Z) = \hat{\theta}^\top \hat{Z} + \varepsilon_U , \quad \text{with} \quad \hat{\theta} \triangleq \theta\mathbf{C}^{-1} . \tag{103}$$

As $\mathbf{A}$ and $\mathbf{A}^*$ are problem dependent constant matrices with finite element, there exists $m_A \in \mathbb{R}^+$ such that $|\mathbf{A}[i,j]| \le m_A$ and $|\mathbf{A}^*[i,j]| \le m_A$ and $|\hat{\theta}[i]| \le m_A$ for $i,j \in [n]$.

To simplify notation, we omit redefining other quantities (e.g. the intervention mean $\mu_{\mathbf{a}}$), which remain almost unchanged.

**Cumulative estimation error.** In Theorem 2 we characterized high-probability bounds on $\mathbf{E}_t$, which captures the error in estimating $\mathbf{Z}_t$. Recall that we have scheduled $\delta_t = \frac{6\delta}{\pi^2 t^2}$ as in Section 4 and impose the following condition on forced exploration $T_0$ and $f_t(\hat{\mathcal{G}}_t)$ for general regret upper bound.

$$T_0 \ge N\left(\epsilon_{\max}, \delta_{nT_0}\right) , \quad \text{and} \quad f_t(\hat{\mathcal{G}}_t) \ge N(\epsilon_{\max}, \delta_t) . \tag{104}$$

We note that the setting we adopt for Theorem 3 later satisfies the above condition.

We know that at time $t$, Algorithm 1 uses the estimates $\mathbf{H}_t$ to estimate $\hat{\mathbf{Z}}_t$ only when there are no under-explored interventions. That is, we have for $\mathbf{a} \in \mathcal{A}_0$ the following condition holds

$$N_{\mathbf{a},t} \ \geq \ f_t(\hat{\mathcal{G}}_t) \geq N(\epsilon_{\max}, \delta_t) \ . \tag{105}$$

Then according to Theorem 2, we have with probability at least $1 - \frac{6\delta}{\pi^2 t^2}$, the following error bound

$$\|\mathbf{E}_t\| \leq C \sqrt{\frac{d + \log(1/\delta_t)}{f_t(\hat{\mathcal{G}}_t)}} \ . \tag{106}$$

Define the error of estimating $\hat{\mathbf{Z}}_t$, $\Delta_t \in \mathbb{R}^{n \times t}$ by

$$\Delta_t \ \triangleq \ (\mathbf{H}_t - \mathbf{H}_\infty)\mathbf{X}_t \ . \tag{107}$$

For each $s \in [t]$, we have the following 2-norm bound for the error bound for the estimates for $Z_s$ at time $t$ as follows.

$$\|\Delta_t[:, s]\|_2 \leq \|\mathbf{E}_t\|_2 \|Z_s\|_2 \leq m \, C \sqrt{\frac{d + \log(1/\delta_t)}{f_t(\hat{\mathcal{G}}_t)}} \ , \tag{108}$$

where the first inequality holds is due to the triangle inequality for norm, and the second inequality holds is due to (106) and the boundedness of latent variables $Z$.

We define the estimation error of sample $s \in [t]$ at time $t \in \mathbb{N}$ as

$$\hat{\mathbf{Z}}_t[:, s] \ = \ \hat{\mathbf{Z}}_\infty[:, s] + \Delta_t[:, s] \ . \tag{109}$$

Plugging (109) into the SEM in (93) and (94), We obtain at time $s \in [t]$

$$\hat{\mathbf{Z}}_t[:, s] = \mathbf{A}_{\mathbf{a}_s} \hat{\mathbf{Z}}_\infty[:, s] + (\mathbf{I}_n - \mathbf{A}_{\mathbf{a}_s})\Delta_t[:, s] + \hat{\varepsilon}_\infty[:, s] \ . \tag{110}$$

Hence the error term $\mathbf{e}_t \in R^{n \times t}$ in the finite sample SEM is defined as

$$\mathbf{e}_t[:, s] \ \triangleq \ (\mathbf{I}_n - \mathbf{A}_{\mathbf{a}_s})\Delta_t[:, s] \ . \tag{111}$$

We can bound the above error for $i \in [n]$ and $s \in [t]$ as

$$|\mathbf{e}_t[i, s]| \leq \left\| [\mathbf{I}_{n+1} - \mathbf{A}_{\mathbf{a}_s}]_i \, \Delta_t[:, s] \right\|_2 \tag{112}$$

$$\leq \|[\mathbf{I}_{n+1} - \mathbf{A}_{\mathbf{a}_s}]\|_2 \|\Delta_t[:, s]\|_2 \tag{113}$$

$$\leq m \, C' \sqrt{\frac{d + \log(1/\delta_t)}{f_t(\hat{\mathcal{G}}_t)}} \ , \tag{114}$$

where we set $C' = C \cdot \max_{\mathbf{a} \in \mathcal{A}} \max_{i \in [n]} \|[\mathbf{I}_{n+1} - \mathbf{A}_{\mathbf{a}_s}]_i\|_2$, which is an instant-dependent constant.

We define the cumulative estimation error bound $\zeta_t$ at time $t \geq nT_0$ as

$$\zeta_t = t \, m \, C' \sqrt{\frac{d + \log(1/\delta_t)}{f_t(\hat{\mathcal{G}}_t)}} \ , \tag{115}$$

where $C'$ is an instance-dependent constant. This choice allows the following condition to hold.

$$\zeta_t \geq \sum_{s=1}^{t} |\mathbf{e}_t[i, s]| \ , \quad \forall i \in [n] \ . \tag{116}$$

**Remark 3.** *For the efficient RO-CRL , we need to set the cumulative estimation error bound $\zeta_t$ as*

$$\zeta_t' = \sum_{s \in [t]} m \, C' \sqrt{\frac{d + \log(1/\delta_s)}{f_s(\hat{\mathcal{G}}_t)}} \tag{117}$$

*Or one can do more adaptive based on when you reset the whole estimates. The regret order of efficient RO-CRL will be the same as RO-CRL but with a larger constant multiplier.*

**Bounded variables.** We have the following for the estimates for $t \in \mathbb{N}$

$$\|\hat{\mathbf{Z}}_\infty[:, t]\| = \|\mathbf{C}\mathbf{Z}_\infty[:, t]\| \le \|\mathbf{C}\|_2 \|\mathbf{Z}_\infty[:, t]\|_2 \le \|\mathbf{C}\|_2 \, m \,, \tag{118}$$

where the last inequality is due to the bounded variable $\|Z\| \le m$

Similarly, for all $s \in [t]$, we have the following bound for $\hat{\mathbf{Z}}_t[:, s]$ as

$$\|\hat{\mathbf{Z}}_t[:, s]\| \le \|\hat{\mathbf{Z}}_\infty[:, s] + \Delta_t[:, s]\| \le \|\hat{Z}_t\| + \|\Delta_t[:, s]\| \le \tilde{m} \,, \tag{119}$$

where we have defined

$$\tilde{m} = \left( \|\mathbf{C}\|_2 + C' \sqrt{\frac{d + \log(1/\delta_t)}{f_t(\hat{\mathcal{G}}_t)}} \right) m = \tilde{\mathcal{O}}\left( \left( 1 + \sqrt{\frac{d}{f_t(\hat{\mathcal{G}}_t)}} \right) m \right) . \tag{120}$$

Recall from (104) that

$$f_t(\mathcal{G}_t) \ge N(\epsilon_{\max}, \delta_t) = \tilde{\mathcal{O}}(d) \,. \tag{121}$$

Hence, we have

$$\tilde{m} = \tilde{\mathcal{O}}(m) \,. \tag{122}$$

### F.1.2 Generalization to Unknown Mean Setting

To handle an unknown post-transform noise mean, we augment the graph $\hat{\mathcal{G}}_t$ with a dummy node, prepend a 1 to each variable and noise vector, and adjust the weight matrices accordingly. In such a case, we define

$$Z^{\mathrm{P}} = \begin{bmatrix} 1 \\ Z \end{bmatrix} . \tag{123}$$

Subsequently, the estimate at time $t$ is given by

$$\hat{\mathbf{Z}}_t^{\mathrm{P}} = \begin{bmatrix} \mathbf{1}_t \\ \hat{\mathbf{Z}}_t \end{bmatrix} . \tag{124}$$

Analogous to Lemma 5, for all $t \in \mathbb{N}$ with $\mathbf{a}_t = \emptyset$, the padded variables satisfy

$$\hat{\mathbf{Z}}_\infty^{\mathrm{P}}[:, t] = \mathbf{A}^{\mathrm{P}} \cdot \hat{\mathbf{Z}}^{\mathrm{P}}[:, t] + \hat{\varepsilon}_\infty^{\mathrm{P}}[:, t] \,, \tag{125}$$

where we have defined $\mathbf{A}^{\mathrm{P}} \in \mathbb{R}^{(n+1) \times (n+1)}$

$$\mathbf{A}^{\mathrm{P}} = \begin{bmatrix} 1 & \mathbf{0} \\ \hat{\nu} & \mathbf{A} \end{bmatrix} \,, \quad \text{and} \quad \hat{\varepsilon}_\infty^{\mathrm{P}}[:, t] = \begin{bmatrix} 1 \\ \hat{\varepsilon}_\infty[:, t] - \hat{\nu} \end{bmatrix} . \tag{126}$$

Similarly, we can define the weights and noises under intervention as

$$\mathbf{A}^{*\mathrm{P}} = \begin{bmatrix} 1 & \mathbf{0} \\ \hat{\nu}^* & \mathbf{A}^* \end{bmatrix} \,, \quad \text{and} \quad \hat{\varepsilon}_\infty^{\mathrm{P}}[:, t] = \begin{bmatrix} 1 \\ \hat{\varepsilon}_\infty[:, t] - \hat{\nu}^* \end{bmatrix} \,, \tag{127}$$

where the estimated mean values are

$$\mathbb{E}[\hat{\varepsilon}_\infty^{\mathrm{P}}[:, t]] = \begin{bmatrix} 1 \\ \mathbf{0} \end{bmatrix} . \tag{128}$$

The weights for the utility parameter can be appended as

$$\hat{\theta} = \begin{bmatrix} 0 \\ \theta \mathbf{C}^{-1} \end{bmatrix} . \tag{129}$$

Since the dummy node requires no estimation, the choice of $\zeta_t$ and the **cumulative estimation error bounds** remain valid. Consequently, identical order bounds apply to both $\hat{\mathbf{Z}}_\infty$ and $\hat{\mathbf{Z}}_t$, as only an extra dimension of 1 has been added.

## F.2 Decomposition of Node-level Utility

Similar to [21, Lemma 1], we present the following decomposition for the expected utility value. Our design of the mean value estimator in the UCB definition in (31) is based on this lemma.

**Corollary 1.** *When $\hat{\nu}_{\mathbf{a}}$ is known, for intervention $\mathbf{a} \in \mathcal{A}$, the expected utility is related to the noise vector $\varepsilon$ via*

$$\mu_{\mathbf{a}} = \left\langle \hat{\theta} , \sum_{\ell=0}^{L} \mathbf{A}_{\mathbf{a}}^{\ell} \cdot \nu_{\mathbf{a}} \right\rangle , \tag{130}$$

*where $\mathbf{A}^{\ell}$ denotes the $\ell$-th power of matrix $\mathbf{A}$.*

*Proof.* We first show the results for the expected value of $Z$ under intervention $\mathbf{a} \in \mathcal{A}$ as follows. We know from the linear SEMs that

$$\hat{\mathbf{Z}}_{\infty} = (\mathbf{I}_n - \mathbf{A}_{\mathbf{a}})^{-1} \varepsilon_{\infty} . \tag{131}$$

So, in linear SEMs, each latent random variable $Z_i$ can be specified as a linear function of the exogenous noise variables $\varepsilon$ via recursive substitution of the structural equations. And the inverse has a simple expansion since $\mathbf{A}_{\mathbf{a}}$ is strictly lower triangular. Specifically,

$$(\mathbf{I}_n - \mathbf{A}_{\mathbf{a}})^{-1} = \left( \mathbf{I}_n + \sum_{\ell=0}^{\infty} \mathbf{A}_{\mathbf{a}}^{\ell} \right) \tag{132}$$

$$= \left( \sum_{\ell=0}^{L} \mathbf{A}_{\mathbf{a}}^{\ell} \right) , \tag{133}$$

where (133) holds due to $L$ is the maximum path length and $\mathbf{A}^{\ell}$ becomes a zero matrix for $\ell \geq L+1$. Define the variable $\hat{Z} \triangleq \mathbf{H}_{\infty} X = \mathbf{A}_{\mathbf{a}} \hat{Z} + \varepsilon$ Hence, we obtain

$$\hat{Z} = \sum_{\ell=0}^{L} \mathbf{A}_{\mathbf{a}}^{\ell} \cdot \varepsilon . \tag{134}$$

Since $\varepsilon$ and $\mathbf{a}$ are independent, the expectation of each $\varepsilon_i$ is 0 for $i \in [n]$, and dummy noise. Then, we obtain the following results for the mean value

$$\mathbb{E}[\hat{Z}] = \sum_{\ell=0}^{L} \mathbb{E} \left[ \mathbf{A}_{\mathbf{a}}^{\ell} \cdot \varepsilon \right] \tag{135}$$

$$= \sum_{\ell=0}^{L} \sum_{i=1}^{n} \mathbf{A}_{\mathbf{a}}^{\ell}[:, i] \mathbb{E} \left[ \varepsilon_i \right] \tag{136}$$

$$= \sum_{\ell=0}^{L} \mathbf{A}_{\mathbf{a}}^{\ell} \cdot \nu_{\mathbf{a}} . \tag{137}$$

Then, by the definition of expected mean of utility in (2), we have

$$\mu_{\mathbf{a}} = \mathbb{E}_{\mathbf{a}}[U(Z)] \tag{138}$$

$$= \mathbb{E}_{\mathbf{a}}[\theta^{\top} Z + \varepsilon_U] \tag{139}$$

$$= \mathbb{E}_{\mathbf{a}}[\hat{\theta}^{\top} \hat{Z} + \varepsilon_U] \tag{140}$$

$$= \hat{\theta}^{\top} \mathbb{E}_{\mathbf{a}}[\hat{Z}] \tag{141}$$

$$= \left\langle \hat{\theta} , \sum_{\ell=0}^{L} \mathbf{A}_{\mathbf{a}}^{\ell} \cdot \nu_{\mathbf{a}} \right\rangle . \tag{142}$$

$\blacksquare$

**Under unknown transformed mean.** Under unknown transformed mean, we are working with $\mathbf{A}^{\mathrm{P}}$ and $\mathbf{A}^{*\mathrm{P}}$, we define the transformed weight matrix with padding as

$$\mathbf{A}_{\mathbf{a}}^{\mathrm{P}} = \mathbb{1}\{i \in \mathbf{a}_t\} \, \mathbf{A}^{*\mathrm{P}} + \mathbb{1}\{i \notin \mathbf{a}_t\} + \mathbf{A}^{\mathrm{P}} \, . \tag{143}$$

For the expected utility value under an unknown transformed mean, we have the following lemma.

**Lemma 6.** *Given intervention* $\mathbf{a} \in \mathcal{A}$*, the expected utility value depends to the noise vector* $\varepsilon$ *via*

$$\mu_{\mathbf{a}} = \left\langle \hat{\theta} \, , \, \sum_{\ell=0}^{L+1} \mathbf{A}_{\mathbf{a}}^{\mathrm{P}\ell}[:, 0] \right\rangle , \tag{144}$$

*where* $\mathbf{A}^{\ell}$ *denotes the* $\ell$*-th power of matrix* $\mathbf{A}$*.*

*Proof:* The first few steps for the proof of Lemma 1 still hold, with $\mathbf{A}_{\mathbf{a}}^{\mathrm{P}}$, and maximum length $L+1$ instead of $L$ due to the dummy node, the changes in the proof start from (137), where we have

$$\mathbb{E}[\hat{Z}^{\mathrm{P}}] = \sum_{\ell=0}^{L+1} \sum_{i=1}^{n} \mathbf{A}_{\mathbf{a}}^{\mathrm{P}\ell}[:, i] \mathbb{E}\left[\varepsilon_i\right] \tag{145}$$

$$= \sum_{\ell=0}^{L+1} \mathbf{A}_{\mathbf{a}}^{\mathrm{P}\ell}[:, 0] \, . \tag{146}$$

Then, as the expected mean of utility defined in (2), we have

$$\mu_{\mathbf{a}} = \mathbb{E}_{\mathbf{a}}[U(Z)] \tag{147}$$

$$= \mathbb{E}_{\mathbf{a}}[\theta^\top Z + \varepsilon_U] \tag{148}$$

$$= \mathbb{E}_{\mathbf{a}}[\hat{\theta}^\top \hat{Z}^{\mathrm{P}} + \varepsilon_U] \tag{149}$$

$$= \hat{\theta}^\top \mathbb{E}_{\mathbf{a}}[\hat{Z}] \tag{150}$$

$$= \left\langle \hat{\theta}, \sum_{\ell=0}^{L+1} \mathbf{A}^{\mathrm{P}\ell}[:, 0] \right\rangle \, . \tag{151}$$

∎

## F.3 Proof of Concentration Inequality

In this section, we provide the concentration inequality and then note how to extend it to the unknown noise mean setting. We notice that certain parts of the proof can be replaced by a non-time-uniform argument (which does not improve the rate), so we keep the same flow as previous results while focusing on the essential difference. We note we always work on the SEM associated with (93) and (94).

As the estimation quality is related to the degrees of freedom, we begin by introducing the in-degree of our graph estimates. We denote $d_{i,t}$ as the number of parents of node $i$ in $\hat{\mathcal{G}}_t$. Then, by Theorem 2 together with our choice of the confidence levels $\{\delta_t\}$, it holds with probability at least $1 - \delta$, $d_{i,t} = |\mathsf{pa}(i)|$ under hard intervention and $d_{i,t} = |\mathsf{an}(i)|$ under soft intervention with all $i \in [n]$ and $t \in [\mathbb{N}]$.

**Lemma 7** (Confidence ellipsoids). *With probability at least* $1 - 3\delta$*, for any node* $i \in [n]$ *and* $t \geq 1$*, we have*

$$\|[\mathbf{A}_t]_i - [\mathbf{A}]_i\|_{\mathbf{V}_{i,t}[\tilde{\mathbf{V}}_{i,t,t}]^{-1}\mathbf{V}_{i,t}} \leq \beta_{i,t}(\delta_t) \, , \tag{152}$$

$$\|[\mathbf{A}_t^*]_i - [\mathbf{A}^*]_i\|_{\mathbf{V}_{i,t}^*[\tilde{\mathbf{V}}_{i,t,t}^*]^{-1}\mathbf{V}_{i,t}^*} \leq \beta_{i,t}(\delta_t) \, , \tag{153}$$

$$and \quad \|\theta_t - \mathbf{C}^{-1}\theta\|_{\mathbf{V}_{\theta,t}[\tilde{\mathbf{V}}_{\theta,t,t}]^{-1}\mathbf{V}_{\theta,t}} \leq \beta_t(\delta_t) \, , \tag{154}$$

*where* $\tilde{m} = \tilde{\mathcal{O}}\big((1 + \sqrt{d/f_t(\hat{\mathcal{G}}_t)})m\big)$ *and*

$$\beta_{i,t}(\delta_t) \triangleq 1 + \sqrt{d_{i,t}} + \sqrt{2\log\left(n/\delta_t\right) + d_{i,t}\log\left(1 + \tilde{m}^2 t/d_{i,t}\zeta_t^2\right)} \tag{155}$$

$$and \quad \beta_t(\delta_t) \triangleq 1 + \sqrt{n} + \sqrt{2\log\left(1/\delta_t\right) + n\log\left(1 + \tilde{m}^2 t/n\zeta_t^2\right)} \, . \tag{156}$$

*Proof:* We observe that regressing $[\hat{\mathbf{Z}}_t]_i - [\mathbf{e}_t]_i$ on $[\hat{\mathbf{Z}}_t]_{\mathsf{pa}_t(i)}$, restricted to time indices under the observational mechanism, yields an unbiased estimator of $\mathbf{A}$. This connects the proof of our concentration lemma to prior results in robust causal bandits [23]. However, a key distinction lies in the fact that both random variables on the node $i$ and its parent nodes $\mathsf{pa}_t(i)$ have estimation errors. This introduces two sources of noise, which, if treated using previous techniques that assume noiseless anchors using regression $[\mathbf{Z}]_i$ on $[\mathbf{Z}]_{\mathsf{pa}_t(i)}$, would result in a multiplicative increase in the error. To avoid this, we use the regression of $[\hat{\mathbf{Z}}_t]_i - [\mathbf{e}_t]_i$ on $[\hat{\mathbf{Z}}_t]_{\mathsf{pa}_t(i)}$ as our anchor instead of relying on the true variables. A technic based on [29]

We will provide the proof corresponding to the observational weights $[\mathbf{A}_t]_i$, while the proof for the interventional weights $[\mathbf{A}_t^*]_i$ and the utility parameters $\theta$ follows similarly.

We prove it by first establishing it for a given time $t \in \mathbb{Z}$. For any node $i \in [n]$, we decompose the error in estimation $\|[\mathbf{A}_t]_i - \mathbf{A}_i\|_{\mathbf{V}_{i,t}[\tilde{\mathbf{V}}_{i,t,t}]^{-1}\mathbf{V}_{i,t}}$ for $s \in [t]$ as follows.

$$\|[\mathbf{A}_t]_i - \mathbf{A}_i\|_{\mathbf{V}_{i,t}[\tilde{\mathbf{V}}_{i,t,t}]^{-1}\mathbf{V}_{i,t}} \tag{157}$$

$$= \left\|[\mathbf{V}_{i,t}]^{-1}[\hat{\mathbf{Z}}_t]_{\mathsf{pa}_t(i)}^{\top}\mathbf{W}_{i,t}([\hat{\mathbf{Z}}_t]_i - \hat{\nu}) - \mathbf{A}_i\right\|_{\mathbf{V}_{i,t}[\tilde{\mathbf{V}}_{i,t,t}]^{-1}\mathbf{V}_{i,t}} \tag{158}$$

$$= \left\|[\mathbf{V}_{i,t}]^{-1}[\hat{\mathbf{Z}}_t]_{\mathsf{pa}_t(i)}^{\top}\mathbf{W}_{i,t}([\hat{\mathbf{Z}}_t]_i - [\mathbf{e}_t]_i - \hat{\nu} + [\mathbf{e}_t]_i) - \mathbf{A}_i\right\|_{\mathbf{V}_{i,t}[\tilde{\mathbf{V}}_{i,t,t}]^{-1}\mathbf{V}_{i,t}} \tag{159}$$

$$\leq \underbrace{\left\|[\hat{\mathbf{A}}_t]_i - \mathbf{A}_i\right\|_{\mathbf{V}_{i,t}[\tilde{\mathbf{V}}_{i,t,t}]^{-1}\mathbf{V}_{i,t}}}_{I_1:\text{ Stochastic and regularization error}} + \underbrace{\left\|[\hat{\mathbf{Z}}_t]_{\mathsf{pa}_t(i)}^{\top}\mathbf{W}_{i,t}[\mathbf{e}_t]_i\right\|_{[\tilde{\mathbf{V}}_{i,t,t}]^{-1}}}_{I_2:\text{ Fluctuation error}}. \tag{160}$$

where $\hat{\mathbf{A}}_t$ refers to the auxiliary estimators which correspond to the ridge regression estimator when knowing the estimation error $\mathbf{e}_t$ on $\hat{\mathbf{Z}}_t$, i.e.,

$$[\hat{\mathbf{A}}_t]_i = [\mathbf{V}_{i,t}]^{-1}[\hat{\mathbf{Z}}_t]_{\mathsf{pa}_t(i)}\mathbf{W}_{i,t}([\hat{\mathbf{Z}}_t]_i - [\mathbf{e}_t]_i - \hat{\nu}). \tag{161}$$

Next, we bound the two error terms $I_1$ and $I_2$.

**Bounding $I_1$.** The stochastic and regularization errors can be bounded by the following lemma. We notice we do not need the time uniform bounds, but non-time uniform bounds will not improve the regret order (See [30, Section 20] for a related example).

**Lemma 8.** *For all node $i \in [n]$, for given $t \in [T]$ with probability at least $1 - \delta_t$, we have*

$$I_1 = \left\|[\hat{\mathbf{A}}_t]_i - \mathbf{A}_i\right\|_{\mathbf{V}_{i,t}[\tilde{\mathbf{V}}_{i,t,t}]^{-1}\mathbf{V}_{i,t}} \leq \sqrt{d_{i,t}} + \sqrt{2\log\left(\frac{n}{\delta_t}\right) + d_{i,t}\log\left(1 + \frac{\hat{m}^2 t}{d_{i,t}\zeta_t^2}\right)}. \tag{162}$$

*Proof:* This lemma follows the result from [23], which is based on [31, Theorem 1]. ■

Based on Lemma 8 and our scheduling of $\delta_t$ such that $\sum_{t\in\mathbb{N}} \delta_t = \delta$, we immediately have the following lemma

**Lemma 9.** *For all node $i \in [n]$, with probability at least $1 - \delta$, for all $t \in \mathbb{N}$, we have*

$$I_1 = \left\|[\hat{\mathbf{A}}_t]_i - \mathbf{A}_i\right\|_{\mathbf{V}_{i,t}[\tilde{\mathbf{V}}_{i,t,t}]^{-1}\mathbf{V}_{i,t}} \leq \sqrt{d_{i,t}} + \sqrt{2\log\left(\frac{n}{\delta_t}\right) + d_{i,t}\log\left(1 + \frac{\hat{m}^2 t}{d_{i,t}\zeta_t^2}\right)}. \tag{163}$$

**Bounding $I_2$.** Now we need to bound the fluctuation error $I_2$, which can be decomposed as

$$I_2 = \left\|[\tilde{\mathbf{V}}_{i,t,t}]^{-1/2}[\hat{\mathbf{Z}}_t]_{\mathsf{pa}_t(i)}\mathbf{W}_{i,t}[\mathbf{e}_t]_i\right\| \tag{164}$$

$$= \left\|[\tilde{\mathbf{V}}_{i,t,t}]^{-1/2}\sum_{s\in[t],i\notin\mathbf{a}_s}\mathbf{W}_{i,t}[s,s]\,\hat{\mathbf{Z}}_t[\mathsf{pa}_t(i),s]\,\mathbf{e}_t[i,s]\right\| \tag{165}$$

$$\leq \sum_{s\in[t],i\notin\mathbf{a}_s}\mathbf{W}_{i,t}[s,s]\left\|[\tilde{\mathbf{V}}_{i,t,t}]^{-1/2}\hat{\mathbf{Z}}_t[\mathsf{pa}_t(i),s]\,\mathbf{e}_t[i,s]\right\| \tag{166}$$

$$= \sum_{s \in [t], i \notin \mathbf{a}_s} \mathbf{W}_{i,t}[s,s] \left\| [\tilde{\mathbf{V}}_{i,t,t}]^{-1/2} \hat{\mathbf{Z}}_t[\mathsf{pa}_t(i), s] \right\| |\mathbf{e}_t[i,s]| \tag{167}$$

$$\leq \sum_{s \in [t], i \notin \mathbf{a}_s} \mathbf{W}_{i,t}[s,s] |\mathbf{e}_t[i,s]| \left\| \hat{\mathbf{Z}}_t[\mathsf{pa}_t(i), s] \right\|_{[\tilde{\mathbf{V}}_{i,t,t}]^{-1}} \tag{168}$$

$$\leq \sum_{s \in [t], i \notin \mathbf{a}_s} \mathbf{W}_{i,t}[s,s] |\mathbf{e}_t[i,s]| \left\| \hat{\mathbf{Z}}_t[\mathsf{pa}_t(i), s] \right\|_{[\tilde{\mathbf{V}}_{i,t,s}]^{-1}} \tag{169}$$

$$\leq 1 , \tag{170}$$

where (164) and (165) follow the definition of weighted norm and weight matrix, (166) and (167) hold due to the triangle inequality and weights are non-negative, (169) holds due to $\|x\|_{[\tilde{\mathbf{V}}_{i,t}]^{-1}} \leq \|x\|_{[\tilde{\mathbf{V}}_{i,s}]^{-1}}$, and (170) is obtained using the definition of the weights and the property $\zeta_t \geq \sum_{s=1}^t |\mathbf{e}_t[i,s]|$ in (116).

Finally, substituting the results of Lemma 9 and (170), with probability at least $1 - \delta$, for all $t \geq 0$, we have

$$\|[\mathbf{A}_t]_i - \mathbf{A}_i\|_{\mathbf{V}_{i,t}[\tilde{\mathbf{V}}_{i,t,t}]^{-1}\mathbf{V}_{i,t}} \leq 1 + \sqrt{d_{i,t}} + \sqrt{2\log\left(\frac{1}{n\delta_t}\right) + d_{i,t}\log\left(1 + \frac{\hat{m}^2 t}{d_i \zeta_t^2}\right)} . \tag{171}$$

Similarly, for the estimators for interventional weights, with probability at least $1 - \delta$, for all $t \geq 0$, we have

$$\|[\mathbf{A}_t^*]_i - \mathbf{A}_i^*\|_{\mathbf{V}_{i,t}^*[\tilde{\mathbf{V}}_{i,t,t}^*]^{-1}\mathbf{V}_{i,t}^*} \leq 1 + \sqrt{d_{i,t}} + \sqrt{2\log\left(\frac{1}{n\delta_t}\right) + d_i\log\left(1 + \frac{\hat{m}^2 t}{d_{i,t}\zeta_t^2}\right)} . \tag{172}$$

A similar proof can be get for the utility $U$ where the in-degree is $n$ and only one concentration bound instead of $n$ is needed, hence, with probability at least $1 - \delta$, for all $t \geq 0$, we have

$$\|\theta_t - \hat{\theta}\|_{\mathbf{V}_{\theta,t}[\tilde{\mathbf{V}}_{\theta,t,t}]^{-1}\mathbf{V}_{\theta,t}} \leq 1 + \sqrt{n} + \sqrt{2\log\left(\frac{1}{\delta}\right) + n\log\left(1 + \frac{\hat{m}^2 t}{n\zeta_t^2}\right)} . \tag{173}$$

Combining the results in (171), (172) and (173) we complete the proof. ∎

Finally, in the unknown noise mean setting one simply replaces each $d_i$ by $d_i + 1$ (to account for the dummy node) and adjusts $\tilde{m}$ accordingly.

### F.4 Cumulative Estimation Error

Lemma 7 in the previous section provides high-probability error bounds on our estimators. Due to the causal structure, these errors accumulate and propagate along the causal paths, leading to the estimation error in the utility. So we provide the following optimistic cumulative estimation error for utility $U$. To start with, we define the detailed cumulative uncertainty $u_{\beta,t} \in \{u_{\beta,\mathrm{S}}, u_{\beta,\mathrm{H}}\}$.

$$u_{\beta,i,t} = \begin{cases} 0 & \text{if } i \text{ is a root node} \\ m_A \sum_{j \in \mathsf{pa}_t(i)} u_{\beta,j} + \beta_{i,t} & \text{otherwise} \end{cases} , \quad \text{and} \quad u_{\beta,t} = m_A \sum_{i=1}^n u_{\beta,i,t} + \beta_t . \tag{174}$$

**Lemma 10.** *If $\hat{\mathcal{G}}_t = \mathcal{G}$ or $\hat{\mathcal{G}}_t = \mathcal{G}_{\mathrm{tc}}$ for all $t \in [\mathbb{N}]$, and $[\mathbf{A}]_i \in \mathcal{C}_{i,t}$ and $[\mathbf{A}^*]_i \in \mathcal{C}_{i,t}^*$ for all $t \in \mathbb{N}$ and $i \in [n]$ and $\theta_t \in \mathcal{C}_{\theta,t}$ for all $t \in \mathbb{N}$, then we have*

$$\sum_{t=1}^T \mathbb{1}\{\mathcal{A}_t^{\mathrm{UE}} = \emptyset\} \mathbb{E}_{\mathbf{a}_t} \left| \tilde{U}(\tilde{Z}_t) - U(\hat{\mathbf{Z}}_t[:,t]) \right| \leq 2\hat{m}\mathcal{B} \, u_{\beta,T} , \tag{175}$$

*where $u = u_{\mathrm{H}}$ if $\hat{\mathcal{G}}_t = \mathcal{G}$ and $u = u_{\mathrm{S}}$ if $\hat{\mathcal{G}}_t = \mathcal{G}_{\mathrm{tc}}$, and we define the term*

$$\mathcal{B} = \frac{4\sqrt{\hat{m}\kappa_{\max}}}{\kappa_{\min}} \sqrt{T} + \frac{8}{\kappa_{\min}} \sqrt[4]{\frac{3T}{2}} + E_1 \tag{176}$$

$$+ \frac{4\hat{m}}{\kappa_{\min}} \log\left(\frac{\kappa_{\min}}{\hat{m}}\sqrt{\frac{T}{2}} + \alpha\hat{m}^2\right)\zeta_T , \tag{177}$$

*where* $\tau = \frac{\alpha^2\hat{m}^6}{\kappa_{\min}^2}$, $\alpha = \sqrt{\frac{16}{3}\log((\max_{i\in[n]} d_i + 1)T^{5/2}(T+1))}$ *and*

$$E_1 = 4\frac{\sqrt{\hat{m}}\kappa_{\max}}{\kappa_{\min}}\sqrt{\tau}\log\left(\sqrt{\frac{T}{2}} + \sqrt{\tau}\right) \tag{178}$$

$$+ 4\sqrt{\frac{\alpha\hat{m}^5}{\kappa_{\min}^3}}\log\left(\frac{\sqrt{\frac{1}{\tau}}\sqrt[4]{\frac{T}{2}} + \sqrt[4]{4} + 1}{\sqrt{\frac{1}{\tau}}\sqrt[4]{\frac{T}{2}} + \sqrt[4]{4} - 1}\right) \tag{179}$$

$$+ 8\tau\left(\frac{1}{\zeta_{(n+1)T_0}}\sqrt{\kappa_{\max}\tau + \alpha\hat{m}^2\sqrt{\tau}} + 1\right) \tag{180}$$

$$+ \frac{\hat{m}}{\zeta_{(n+1)T_0}T} + \frac{2\hat{m}}{3\zeta_{nT_0}} + 1 . \tag{181}$$

*Proof:* This proof is a cumulative estimation error at the node level of [23]. We first prove that when the conditions of the lemma hold, the latent variables $i \in [n]$ have a cumulative estimation error as follows.

**Lemma 11.** *If* $\hat{\mathcal{G}}_t = \mathcal{G}$ *or* $\hat{\mathcal{G}}_t = \mathcal{G}_{tc}$ *for all* $t \in [\mathbb{N}]$, *and* $[\mathbf{A}]_i \in \mathcal{C}_{i,t}$ *and* $[\mathbf{A}^*]_i \in \mathcal{C}_{i,t}^*$ *for all* $t \in \mathbb{N}$ *and* $i \in [n]$ *and* $\theta_t \in \mathcal{C}_{\theta,t}$ *for all* $t \in \mathbb{N}$, *then for* $i \in [n]$ *we have*

$$\sum_{t=1}^{T} \mathbb{1}\{\mathcal{A}_t^{\mathrm{UE}} = \emptyset\}\mathbb{E}_{\mathbf{a}_t}\left|\tilde{Z}_t[i] - \hat{\mathbf{Z}}_t[i,t]\right| \leq 2\hat{m}\mathcal{B}\, u_{\beta,i,T} , \tag{182}$$

*where* $u_i = u_{i,\mathrm{H}}$ *if* $\hat{\mathcal{G}}_t = \mathcal{G}$ *and* $u_i = u_{i,\mathrm{S}}$ *if* $\hat{\mathcal{G}}_t = \mathcal{G}_{tc}$ *that is defined in* (35).

*Proof:* We establish (182) via induction on the causal depth. We define the *causal depth* of node $i$ as the length of the longest directed causal path that ends at node $i \in [n]$ in $\hat{\mathcal{G}}_t$ and denote it by $L_i$.

**Base step:** $L_i = 1$. This is according to the same proof in [23] with a change on $\hat{m}$, and using the bounds for cumulative error that $\zeta_{nT_0} \leq \zeta_t \leq \zeta_T$ for $nT_0 \leq t \leq T$.

**Induction Step.** Assume that the property holds for causal depths up to $L_i = k$. We show that it will also be satisfied for $L_i = k + 1$.

For this purpose, we start with the following expansion and apply the triangular inequality to find an upper bound for it.

$$\sum_{t=1}^{T} \mathbb{1}\{\mathcal{A}_t^{\mathrm{UE}} = \emptyset\}\mathbb{E}_{\mathbf{a}_t}\left|\tilde{Z}_t[i] - \hat{\mathbf{Z}}_t[i,t]\right| \tag{183}$$

$$= \sum_{t=1}^{T} \mathbb{1}\{\mathcal{A}_t^{\mathrm{UE}} = \emptyset\}\mathbb{E}_{\mathbf{a}_t}\left|[\tilde{\mathbf{A}}_{\mathbf{a}_t,t}]_i\tilde{Z}_t[\mathsf{pa}_t(i)] - [\mathbf{A}_{\mathbf{a}_t}]_i\hat{\mathbf{Z}}_t[\mathsf{pa}_t(i),t]\right| \tag{184}$$

$$= \sum_{t=1}^{T} \mathbb{1}\{\mathcal{A}_t^{\mathrm{UE}} = \emptyset\}\mathbb{E}_{\mathbf{a}_t}\left|[\tilde{\mathbf{A}}_{\mathbf{a}_t,t}]_i\left(\tilde{Z}_t[\mathsf{pa}_t(i)] - \hat{\mathbf{Z}}_t[\mathsf{pa}_t(i),t]\right)\right|$$

$$+ \sum_{t=1}^{T} \mathbb{1}\{\mathcal{A}_t^{\mathrm{UE}} = \emptyset\}\mathbb{E}_{\mathbf{a}_t}\left|\left([\tilde{\mathbf{A}}_{\mathbf{a}_t,t}]_i - [\mathbf{A}_{\mathbf{a}_t}]_i\right)\hat{\mathbf{Z}}_t[\mathsf{pa}_t(i),t]\right| \tag{185}$$

$$\leq m_A \sum_{j\in\mathsf{pa}_t(i)}\sum_{t=1}^{T} \mathbb{1}\{\mathcal{A}_t^{\mathrm{UE}} = \emptyset\}\mathbb{E}_{\mathbf{a}_t}\left|\tilde{Z}_t[j] - \hat{\mathbf{Z}}_t[j,t]\right| + 2\hat{m}\beta_{i,T}\mathcal{B} . \tag{186}$$

where the transition to (185) holds due to the triangular inequality via adding and subtracting terms $[\tilde{\mathbf{A}}_{\mathbf{a}_t,t}]\hat{\mathbf{Z}}_t[\mathsf{pa}_t(i),t]$; and (186) holds since the triangle inequality of $L_2$ norm and $|\tilde{\mathbf{A}}_{\mathbf{a}_t,t}[i,j]| \leq m_A$ and similar proof as in the base step.

Next, we find an upper bound on the first summand in (186). We notice that the summation is taken over all parents of node $i$. Thus, we aim to find an upper bound for the error bound for each parent. Based on the induction assumption, for each node $j \in \mathsf{pa}_t(i)$, we have

$$\sum_{t=1}^{T} \mathbb{1}\{\mathcal{A}_t^{\mathrm{UE}} = \emptyset\} \mathbb{E}_{\mathbf{a}_t} \left| \tilde{Z}_t[j] - \hat{\mathbf{Z}}_t[j, t] \right| \leq 2\hat{m}\mathcal{B}u_{\beta,j,T} . \tag{187}$$

Subsequently, plugging (187) into (186), we obtain

$$\sum_{t=1}^{T} \mathbb{1}\{\mathcal{A}_t^{\mathrm{UE}} = \emptyset\} \sum_{j \in \mathsf{pa}_t(i)} \mathbb{E}_{\mathbf{a}_t} \left| \tilde{Z}_t[j] - \hat{\mathbf{Z}}_t[j, t] \right| \leq \sum_{j \in \mathsf{pa}_t(i)} 2\hat{m}\mathcal{B}u_{\beta,j,T} , \tag{188}$$

Combining the results in (186) and (188), we conclude

$$\sum_{t=1}^{T} \mathbb{1}\{\mathcal{A}_t^{\mathrm{UE}} = \emptyset\} \mathbb{E}_{\mathbf{a}_t} \left| \tilde{Z}[i] - \mathbf{Z}_t[i, t] \right| \leq 2\hat{m}\beta_T \mathcal{B}u_{\beta,i,T} , \tag{189}$$

which proved the desired results in (182). ∎

Now we are ready to prove the final result for the utility function.

$$\sum_{t=1}^{T} \mathbb{1}\{\mathcal{A}_t^{\mathrm{UE}} = \emptyset\} \mathbb{E}_{\mathbf{a}_t} \left| \tilde{U}_t(\tilde{Z}_t) - U(\hat{\mathbf{Z}}_t[:, t]) \right| \tag{190}$$

$$= \sum_{t=1}^{T} \mathbb{1}\{\mathcal{A}_t^{\mathrm{UE}} = \emptyset\} \mathbb{E}_{\mathbf{a}_t} \left| \tilde{\theta}_t^{\top} \tilde{Z}_t - \theta^{\top} \hat{\mathbf{Z}}_t[:, t] \right| \tag{191}$$

$$= \sum_{t=1}^{T} \mathbb{1}\{\mathcal{A}_t^{\mathrm{UE}} = \emptyset\} \mathbb{E}_{\mathbf{a}_t} \left| \tilde{\theta}_t^{\top} \left( \tilde{Z}_t - \hat{\mathbf{Z}}_t[:, t] \right) \right|$$

$$+ \sum_{t=1}^{T} \mathbb{1}\{\mathcal{A}_t^{\mathrm{UE}} = \emptyset\} \mathbb{E}_{\mathbf{a}_t} \left| \left( \tilde{\theta}_t - \theta \right)^{\top} \hat{\mathbf{Z}}_t[:, t] \right| \tag{192}$$

$$\leq m_A \sum_{i=1}^{n} \sum_{t=1}^{T} \mathbb{1}\{\mathcal{A}_t^{\mathrm{UE}} = \emptyset\} \mathbb{E}_{\mathbf{a}_t} \left| \tilde{Z}_t[i] - \hat{\mathbf{Z}}_t[i, t] \right| + 2\hat{m}\beta_{\theta,T}\mathcal{B} \tag{193}$$

$$\leq 2\hat{m}\mathcal{B}u_{\beta,T} . \tag{194}$$

Here we conclude the proof. ∎

### F.5 Proof of General Regret Bounds

**Theorem 6** (General Regret upper bound). *By setting confidence radius $\beta_{i,t}(\delta_t)$ and $\beta_t(\delta_t)$ according to Lemma 7, $f_t(\hat{\mathcal{G}}_t)$ and $T_0$ satisfy the condition in (104). If we define $f_T = \max_{t \in [T]} f_t(\hat{\mathcal{G}}_t)$, then with probability at least $1 - 4\delta$, the average cumulative regret of RO-CRL is upper bounded by*

$$\mathcal{R}_T \leq \tilde{\mathcal{O}}\left( nf_T + u\left( \sqrt{T} + d^{\frac{1}{2}} T f_T^{-\frac{1}{2}} \right) \right) . \tag{195}$$

*Proof:* We start by defining the event in which, over $T$ rounds, all confidence sets $\mathcal{C}_{i,t}$ contain the ground truth parameters $f_i$. Specifically, Now define the error events $\mathcal{E}_i$ and $\mathcal{E}_i^*$ for $i \in [n]$ for each estimator

$$\mathcal{E}_{\mathrm{CRL}} \triangleq \left\{ \forall t \in [T] : \text{Thorem 2 holds at time } t \right\} , \tag{196}$$

$$\mathcal{E}_i \triangleq \left\{ \forall t \in [T] : [\mathbf{A}]_i \in \mathcal{C}_{i,t} \right\} , \tag{197}$$

$$\mathcal{E}_i^* \triangleq \left\{ \forall t \in [T] : [\mathbf{A}^*]_i \in \mathcal{C}_{i,t}^* \right\} , \tag{198}$$

$$\mathcal{E}_\theta \triangleq \left\{ \forall t \in [T] : \hat{\theta} \in \mathcal{C}_{\theta,t} \right\}, \tag{199}$$

where the $\beta_{i,t}(\delta_t)$ and $\beta_t(\delta_t)$ is chosen as in (155) and (156). Accordingly, define the event that at least one of the confidence ellipsoids of estimators does not contain the true parameters at at least one time index

$$\mathcal{E} \triangleq \mathcal{E}_{\mathrm{CRL}} \bigcap \left( \bigcap_{i=1}^n \mathcal{E}_i \right) \bigcap \left( \bigcap_{i=1}^n \mathcal{E}_i^* \right) \bigcap \mathcal{E}_\theta . \tag{200}$$

By invoking the union bound on probability and Lemma 7, we have

$$\mathbb{P}(\mathcal{E}^{\mathrm{c}}) \leq \mathbb{P}(\mathcal{E}_{\mathrm{CRL}}^{\mathrm{c}}) + \sum_{i=1}^n \left( \mathbb{P}(\mathcal{E}_i^{\mathrm{c}}) + \mathbb{P}(\mathcal{E}_i^{*\mathrm{c}}) + \mathbb{P}(\mathcal{E}_\theta^{\mathrm{c}}) \right) \tag{201}$$

$$\leq \sum_{t \in \mathbb{Z}} \delta_t + \sum_{i=1}^n \sum_{t \in \mathbb{Z}} \left( \frac{\delta_t}{n} + \frac{\delta_t}{n} + \frac{\delta_t}{n} \right) = 4\delta . \tag{202}$$

Next, we decompose the regret defined in (5) under the events $\mathcal{E}$.

$$\mathbb{E}[\mathcal{R}_T] = \sum_{t=1}^T \mathbb{E}_{\mathbf{a}^*} \left[ U(Z_t) - \mathbb{E}_{\mathbf{a}_t} U(Z_t) \right] \tag{203}$$

$$= \sum_{t=1}^T \left[ \mathbb{1}\{\mathcal{A}_t^{\mathrm{UE}} \neq \emptyset\} \cdot 2m + \mathbb{1}\{\mathcal{A}_t^{\mathrm{UE}} = \emptyset\} \left( \mathrm{UCB}_{\mathbf{a}_t} - \mathbb{E}_{\mathbf{a}_t}[U(Z_t)] \right) \right] \tag{204}$$

$$\leq nf_T + \sum_{t=1}^T \mathbb{1}\{\mathcal{A}_t^{\mathrm{UE}} = \emptyset\} \mathbb{E} \left[ \mathrm{UCB}_{\mathbf{a}_t} - \mathbb{E}_{\mathbf{a}_t}[U(Z_t)] \right] \tag{205}$$

$$\leq nf_T + \sum_{t=1}^T \mathbb{1}\{\mathcal{A}_t^{\mathrm{UE}} = \emptyset\} \mathbb{E} \left[ \mathbb{E}_{\mathbf{a}_t}[\tilde{U}(\tilde{Z}_t)] - \mathbb{E}_{\mathbf{a}_t}[U(Z_t)] \right], \tag{206}$$

where we have used the set of inequalities

$$\mathbb{E}_{\mathbf{a}^*}[U(Z_t)] \leq \mathrm{UCB}_{\mathbf{a}^*}(t) \leq \mathrm{UCB}_{\mathbf{a}_t}(t) = \mathbb{E}_{\mathbf{a}_t}[\tilde{U}(\tilde{Z}_t)] . \tag{207}$$

Then using the lemma 10 and cumulative estimation $\zeta_t$ defined in (115), we have

$$\mathcal{R}_T \leq nf_T + 2\hat{m}\mathcal{B}u_{\beta,T} \tag{208}$$

$$= \tilde{\mathcal{O}} \left( nf_T + u(\sqrt{T} + d^{\frac{1}{2}}Tf_T^{-\frac{1}{2}}) \right) . \tag{209}$$

∎

We note that the same results can be obtained for the unknown transformed mean setting.

### F.6 Proof of Trade-off Upper Bounds (Theorem 3)

*Proof:* Based on the Theorem 6, the remaining is to balancing the two terms

$$nf_T + ud^{\frac{1}{2}}Tf_T^{-\frac{1}{2}} . \tag{210}$$

while satisfying the condition

$$f_t(\hat{\mathcal{G}}_t) \geq N(\epsilon_{\max}, \delta_t) . \tag{211}$$

Hence, by setting

$$f_t(\hat{\mathcal{G}}_t) = \max\{d^{\frac{1}{3}}n^{-\frac{2}{3}}u_t^{\frac{2}{3}}t^{\frac{2}{3}}, N(\epsilon_{\max}, \delta_t)\} , \tag{212}$$

where $u_t$ is defined as

$$u_{i,t} = \begin{cases} 0 & \text{if } i \text{ is a root node} \\ \sum_{j \in \mathsf{pa}_t(i)} u_{j,t} + \sqrt{|\mathsf{pa}_t(i)|} & \text{otherwise} \end{cases} , \quad \text{and} \quad u_t = \sum_{i=1}^n u_{i,t} + \sqrt{n} , \tag{213}$$

which establishes the desired result. ∎

We note that the same proof steps work for the unknown transformed mean setting with the change in $u$.

## F.7 Refined Upper Bound for Causal Bandit

**Graph-dependent Bound**   Under the causal bandit setting, since $U(Z) = Z[n]$, we eliminate all uncertainty in estimating the $n$-dimensional parameter $\theta$. In fact, we have $\zeta_t = 0$ for all $t \in [\mathbb{N}]$, so no forced exploration is needed. If we set $\zeta_t = 1$ (or set weight matrices to be $\mathbf{I}_n$), by applying Lemma 11 in Section F.4 together with the concentration inequality in Lemma 7, we immediately obtain our regret upper bound with high probability. From (209), we have with probability at least $1 - 2\delta$, the regret bound of modified RO-CRL for CB is

$$\mathcal{R}_T \leq \tilde{\mathcal{O}}\big(u_{\mathrm{H},n}\sqrt{T}\big) . \tag{214}$$

**Graph-independent Bound**   To get a graph-independent bound that corresponds to the maximum in-degree $d_{\mathcal{G}}$ and the maximum length of a causal path $L$. To match the corresponding lower bound that is in the *unknown transformed mean* setting, we show the bound for $u_{\mathrm{H},N}$ in this region. In particular, we have

$$u_{\mathrm{H},n} = d_{\mathcal{G}}^L + \sum_{\ell=1}^{L} d_{\mathcal{G}}^{\frac{2L-1}{2}} = d_{\mathcal{G}}^L + \frac{\sqrt{d_{\mathcal{G}}}(d_{\mathcal{G}}^L - 1)}{d_{\mathcal{G}} - 1} . \tag{215}$$

As we have the fact

$$\frac{\sqrt{d_{\mathcal{G}}}(d_{\mathcal{G}}^L - 1)}{d_{\mathcal{G}} - 1} \leq d_{\mathcal{G}}^L . \tag{216}$$

So we obtain

$$u_{\mathrm{H},n} = \mathcal{O}(d_{\mathcal{G}}^L) . \tag{217}$$

Hence, the regret bound for the modified RO-CRL for CB is

$$\mathcal{R}_T \leq \tilde{\mathcal{O}}\big(d_{\mathcal{G}}^L \sqrt{T}\big) . \tag{218}$$

# G   Proofs of Lower Bounds

**Equivalent definition of $p$.**   An alternative definition of $p$ is $p = 1 + \sum_{i=1}^{n}(p_i + 1)$ and $p_i$ is the number of causal path from noises to node $i$, which is defined recursively as follows (on $\mathcal{G}_{\mathrm{tr}}$).

- On $\mathcal{G}$:

$$p_i = \begin{cases} 1 & \text{if } i \text{ is the root node} \\ \sum_{j \in \mathsf{an}(i)} p_j + 1 & \text{otherwise} \end{cases} . \tag{219}$$

- On $\mathcal{G}_{\mathrm{tr}}$:

$$p_i = \begin{cases} 1 & \text{if } i \text{ is the root node} \\ \sum_{j \in \mathsf{an}(i)} p_j + 1 & \text{otherwise} \end{cases} . \tag{220}$$

## G.1   Graph-dependent Lower Bound for Causal Bandit

**Theorem 7.** *For any given graph $\mathcal{G}$ there exists a causal bandit instance on $\mathcal{G}$ such that the expected regret of any causal bandit algorithm is at least*

$$\mathcal{R}_T \; \geq \; \Omega(p_n\sqrt{T}) , \tag{221}$$

*where $p_n$ is defined in (219) on $\mathcal{G}$.*

Let $\Pi$ be the set of all policies on the set of stochastic bandit environments $\mathcal{I}$, which contains all the possible bandit instances sharing the same DAG $\mathcal{G}$ and satisfying the conditions. The minimax regret is defined as

$$\inf_{\pi \in \Pi} \sup_{\mathcal{I}_0 \in \mathcal{I}} [\mathcal{R}_T \mid \pi, \mathcal{I}_0] , \tag{222}$$

where $[\mathcal{R}_T \mid \pi, \mathcal{I}_0]$ denotes the expected regret of policy $\pi$ on the bandit instance $\mathcal{I}_0$. We will consider a set $\tilde{\mathcal{I}}$, instead of $\mathcal{I}$, that contains two bandit instances. By definition of minimax regret, a lower bound for the regret of any policy on $\tilde{\mathcal{I}}$ is also a lower bound for the minimax regret since

$$\inf_{\pi \in \Pi} \sup_{\mathcal{I}_0 \in \mathcal{I}} [\mathcal{R}_T \mid \pi, \mathcal{I}_0] \geq \inf_{\pi \in \Pi} \sup_{\mathcal{I}_0 \in \tilde{\mathcal{I}}} [\mathcal{R}_T \mid \pi, \mathcal{I}_0] . \tag{223}$$

Following this property, the central idea of the proof is as follows. Consider two linear SEM causal bandit instances that differ by a small fraction and are hard to distinguish. At the same time, we can construct them to have different optimal interventions, indicating that a selection policy cannot incur small regret for both at the same time under the same data realization. Note that the difference of the rewards, or equivalently the regrets, observed by these two bandit instances under the same intervention can be computed by tracing the effect of the differing edge parameter over all the paths that end at the reward node.

We consider two linear SEM causal bandit instances $I, \bar{I} \in \mathcal{I}_0$ that is parameterized by $I \triangleq \{\mathbf{B}, \mathbf{B}^*, \varepsilon\}$ and by $\bar{I} \triangleq \{\mathbf{B}, \mathbf{B}^*, \bar{\varepsilon}\}$. We note here that we assume the mean of the noise is conditionally independent, that is, it can be dependent on the intervention. We note that the algorithm discussed previously can work under this setting without any modifications. For each node $i \in [n]$ in $\mathcal{I}$, we have

$$\varepsilon_i = \begin{cases} \text{Bern}(1/2 + \delta) & \text{if } i \notin \mathbf{a} \\ \text{Bern}(1/2) & \text{if } i \in \mathbf{a} \end{cases}, \tag{224}$$

where $\text{Bern}(q)$ denotes the Bernoulli random distribution with probability $q$. The noise is reversed in the second bandit instance $\bar{I}$, which is

$$\bar{\varepsilon}_i = \begin{cases} \text{Bern}(1/2) & \text{if } i \notin \mathbf{a} \\ \text{Bern}(1/2 + \delta) & \text{if } i \in \mathbf{a} \end{cases}. \tag{225}$$

This is where the difference between the two bandit instances lies: if $(i \rightarrow j)$ is an edge in the graph, we have

$$[\mathbf{B}]_{i,j} = [\mathbf{B}^*]_{i,j} = 1 . \tag{226}$$

Except for this, all the rest are the same. Next, consider a fixed bandit policy $\pi$ that generates the following filtration over time

$$\mathcal{F}_t \triangleq \{\mathbf{a}_1, Z_1, \ldots, \mathbf{a}_t, Z_t\} . \tag{227}$$

The decision of $\pi$ at time $t$ is $\mathcal{F}_{t-1}$-measurable. Accordingly, define $\mathbb{P}_t$ and $\bar{\mathbb{P}}_t$ as the probability measures induced by $\mathcal{F}_t$ by $t$ rounds of interaction between $\pi$ and the two bandit instances $I$ and $\tilde{I}$. When it is clear from context, we use the shorthand terms $\mathbb{P}$ and $\bar{\mathbb{P}}$ for $\mathbb{P}_T$ and $\bar{\mathbb{P}}_T$, respectively. We will show that $\pi$ cannot suffer small regret in both instances at the same time and under the same filtration $\mathcal{F}_T$.

By Lemma 1, since all the elements of observational and interventional weights are non-negative, the optimal intervention is the one that maximizes the expected value of each noise. The optimal action between two bandit instances only differs. This means optimal intervention for $I$ is $\mathbf{a} = \emptyset$ and that for $\bar{I}$ is $\mathbf{a} = [n]$. Define $\mathcal{E}_{\text{lb}}^i$ as the event in which the decision on node $i$ is sup-optimal at least $\frac{T}{2}$ times after $T$ rounds on bandit instance $I$, i.e.,

$$\mathcal{E}_{\text{lb}}^i \triangleq \left\{ N_{i,t}^* \geq \frac{T}{2} \right\} , \quad \text{for} \quad i \in [n] . \tag{228}$$

We note that the event $\mathcal{E}_{\text{lb}}^i$ is defined on the $\sigma$-algebra defined by the filtration $\mathcal{F}_t$, that induces both $\mathbb{P}_t$ and $\bar{\mathbb{P}}_t$. We compute the expected instantaneous regret when node $i$ is chosen sub-optimal in the first bandit instance, and the total regret is the summation over these nodes. Note that each path passes a node that node $i$ contributes to the expected regret. Furthermore, since every weight is positive, in $\mathcal{I}$, when a suboptimal action is chosen, the impact on average regret is determined by the number of paths that start from the node $i$ and end at the reward node $n$. Then, by the definition of $\mathcal{E}_{\text{lb}}$, we have

$$[\mathcal{R}_t \mid \mathbb{P}] = \mathbb{E}_{\mathbb{P}} \left[ \sum_{t=1}^{T} \mu_\emptyset - \mu_{\mathbf{a}_t} \right] \tag{229}$$

$$= \mathbb{E}_{\mathbb{P}} \left[ \sum_{t=1}^{T} \sum_{i \in [n]} \mathbb{1}\{i \notin \mathbf{a}_t\} \delta(m_{i,n}+1) \right] \tag{230}$$

$$\geq \sum_{i \in [n]} \mathbb{P}(\mathcal{E}_{\mathrm{lb}}^i) \frac{T}{2} \delta(m_{i,n}+1) \, , \tag{231}$$

where (230) holds as we break down the regret and (231) holds due to the definition of $\mathcal{E}_{\mathrm{lb}}^j$ in (228).

Similarly, for $\bar{I}$, each node $i$ that is not intervened, it will occur at least $\delta(m_{i,n}+1)$ regret. Applying the same steps as in (229),-(231), we obtain

$$[\mathcal{R}_t \mid \bar{\mathbb{P}}] = \mathbb{E}_{\bar{\mathbb{P}}} \left[ \sum_{t=1}^{T} \mu_{[n]} - \mu_{\mathbf{a}_t} \right] \tag{232}$$

$$\geq \mathbb{E}_{\bar{\mathbb{P}}} \left[ \sum_{t \in [T]} \sum_{i \in [n]} \mathbb{1}\{i \in \mathbf{a}_t\} \delta(m_{i,n}+1) \right] \tag{233}$$

$$\geq \sum_{i \in [n]} \bar{\mathbb{P}}(\mathcal{E}_{\mathrm{lb}}^{i,\mathrm{c}}) \frac{T}{2} \delta(m_{i,n}+1) \, . \tag{234}$$

By combining (231) and (234) we have

$$[\mathcal{R}_t \mid \mathbb{P}] + [\mathcal{R}_t \mid \bar{\mathbb{P}}] \geq \frac{T}{2} \, \delta \sum_{i \in [n]} p_{i,n} [\mathbb{P}(\mathcal{E}_{\mathrm{lb}}^i) + \bar{\mathbb{P}}(\mathcal{E}_{\mathrm{lb}}^{i,\mathrm{c}})] \, . \tag{235}$$

Next, we characterize a lower bound on $\mathbb{P}(\mathcal{E}_{\mathrm{lb}}^i) + \bar{\mathbb{P}}(\mathcal{E}_{\mathrm{lb}}^{i,\mathrm{c}})$ for $i \in [n]$, which involves the Kullback-Leibler (KL) divergence between $\mathbb{P}$ and $\bar{\mathbb{P}}$, denoted by $\mathrm{D}_{\mathrm{KL}}(\mathbb{P} \parallel \bar{\mathbb{P}})$. For this purpose, we leverage the following theorem.

**Theorem 8** (Bretagnolle-Huber inequality). *Let $\mathbb{P}$ and $\bar{\mathbb{P}}$ be probability measures on the same measurable space $(\Omega, \mathcal{F})$ and let $A \in \mathcal{F}$ be an arbitrary event. Then,*

$$\mathbb{P}(A) + \bar{\mathbb{P}}(A^{\mathrm{c}}) \geq \frac{1}{2} \exp(-\mathrm{D}_{\mathrm{KL}}(\mathbb{P} \parallel \bar{\mathbb{P}})) \, . \tag{236}$$

By invoking Theorem 8, from (235) we obtain

$$[\mathcal{R}_t \mid \mathbb{P}] + [\mathcal{R}_t \mid \bar{\mathbb{P}}] \geq \frac{T}{2} \, \delta \sum_{i \in [n]} (m_{i,n}+1)[\mathbb{P}(\mathcal{E}_{\mathrm{lb}}^i) + \bar{\mathbb{P}}(\mathcal{E}_{\mathrm{lb}}^{i,\mathrm{c}})] \tag{237}$$

$$\geq \frac{T}{4} \, \delta \sum_{i \in [n]} (m_{i,n}+1) \exp(-\mathrm{D}_{\mathrm{KL}}(\mathbb{P} \parallel \bar{\mathbb{P}})) \, , \tag{238}$$

$$= \frac{T}{4} \, \delta p_n \exp(-\mathrm{D}_{\mathrm{KL}}(\mathbb{P} \parallel \bar{\mathbb{P}})) \, , \tag{239}$$

It remains to compute $\exp(-\mathrm{D}_{\mathrm{KL}}(\mathbb{P} \parallel \bar{\mathbb{P}}))$ to conclude our proof, for which we leverage the following result.

**Lemma 12.** *The KL divergence between $\mathbb{P}$ and $\bar{\mathbb{P}}$, the probability measures induced by $\mathcal{F}_t$ on $I$ and $\tilde{I}$, is equal to*

$$\mathrm{D}_{\mathrm{KL}}(\mathbb{P} \parallel \bar{\mathbb{P}}) = \frac{nT}{2} \log \left( \frac{1}{(1+2\delta)(1-2\delta)} \right) \, . \tag{240}$$

*Proof:* Note that a Bayesian network factorizes as

$$\mathbb{P}(Z[1], \dots, Z[n]) = \prod_{i=1}^{n} p_i(Z[i] \mid Z[\mathsf{pa}(i)]) \, . \tag{241}$$

Additionally, the two bandit instances differ only in the mechanism of the first layer. Then, $D_{KL}(\mathbb{P} \| \bar{\mathbb{P}})$ can be decomposed as

$$D_{KL}(\mathbb{P} \| \bar{\mathbb{P}}) = \sum_{i=1}^{N} D_{KL}\big(\mathbb{P}(Z[i] \mid Z[\mathsf{pa}(i)]) \| \bar{\mathbb{P}}(Z[i] \mid Z[\mathsf{pa}(i)])\big) . \tag{242}$$

By noting that the KL-divergence between two Bernoulli random variables with probabilities $p$ and $q$ is given by

$$D_{KL}(\mathrm{Bern}(r) \| \mathrm{Bern}(q)) = r \log\left(\frac{r}{q}\right) + (1-r) \log\left(\frac{1-r}{1-q}\right) . \tag{243}$$

Since give $Z[\mathsf{pa}(i)]$, the $Z[i]$ under $\mathbb{P}$ and $\bar{\mathbb{P}}$ are both shifted Bernoulli random variables. From the above, we obtain for node $i \in [n]$

$$D_{KL}(\mathbb{P}(Z[i] \mid Z[\mathsf{pa}(i)]) \| \bar{\mathbb{P}}(Z[i] \mid Z[\mathsf{pa}(i)]))$$

$$= \sum_{t \in [T]: i \notin \mathbf{a}_t} D_{KL}(\mathrm{Bern}(1/2 + \delta) \| \mathrm{Ber}(1/2)) \tag{244}$$

$$+ \sum_{t \in [T]: i \in \mathbf{a}_t} D_{KL}(\mathrm{Bern}(1/2) \| \mathrm{Ber}(1/2 + \delta)) \tag{245}$$

$$= \sum_{s=1}^{T} \mathbb{1}\{i \notin \mathbf{a}_t\} \left[ \left(\frac{1}{2} + \delta\right) \log(1 + 2\delta) + \left(\frac{1}{2} - \delta\right) \log(1 - 2\delta) \right] \tag{246}$$

$$+ \sum_{s=1}^{T} \mathbb{1}\{i \in \mathbf{a}_t\} \frac{1}{2} \log\left(\frac{1}{(1 + 2\delta)(1 - 2\delta)}\right) \tag{247}$$

$$< \frac{T}{2} \log\left(\frac{1}{(1 + 2\delta)(1 - 2\delta)}\right) , \tag{248}$$

where the last inequality holds since $(\frac{1}{2} + \delta) \log(1 + 2\delta) + (\frac{1}{2} - \delta) \log(1 - 2\delta) < \frac{1}{2} \log\left(\frac{1}{(1+2\delta)(1-2\delta)}\right)$ for $0 < \delta < 1/2$. And hence we have

$$D_{KL}(\mathbb{P} \| \bar{\mathbb{P}}) = n \frac{T}{2} \log\left(\frac{1}{(1 + 2\delta)(1 - 2\delta)}\right) . \tag{249}$$

If we choose $\delta = \frac{1}{\sqrt{T}}$ to balance the terms in the lower bound, we obtain

$$\max\{[\mathcal{R}_T \mid \mathbb{P}], [\mathcal{R}_T \mid \bar{\mathbb{P}}]\} \geq \frac{1}{2} \left([\mathcal{R}_T \mid \mathbb{P}] + [\mathcal{R}_T \mid \bar{\mathbb{P}}]\right) \tag{250}$$

$$\overset{(238)}{\geq} \frac{T}{8} p_n \delta \exp(-D_{KL}(\mathbb{P} \| \bar{\mathbb{P}})) \tag{251}$$

$$\overset{(248)}{\geq} \frac{T}{8} p_n \delta [(1 + 2\delta)(1 - 2\delta)]^{Tn/2} \tag{252}$$

$$= \frac{1}{8} p_n \sqrt{T} \times \left(1 - \frac{4}{T}\right)^{Tn/2} . \tag{253}$$

for $T \geq 5$, the term $\left(1 - \frac{4}{T}\right)^{Tn/2}$ is an increasing function of $T$. Hence, for $T \geq 5$, we have the lower bound $\left(1 - \frac{4}{T}\right)^{Tn/2} \geq 0.2^{2.5}$. By setting $c = \frac{1}{8} \times 0.2^{2.5}$, we have

$$\max\{[\mathcal{R}_T \mid \mathbb{P}], [\mathcal{R}_T \mid \bar{\mathbb{P}}]\} \geq c p_n \sqrt{T} . \tag{254}$$

### G.2 Graph-independent Lower Bound for Causal Bandit

To get a graph-independent bound that corresponds to the maximum in-degree $d_{\mathcal{G}}$ and the maximum length of a causal path $L$. When $L \neq 0$ and $d > 1$, we have

$$p_n = \sum_{\ell=0}^{L} d_{\mathcal{G}}^{\ell} = \frac{d_{\mathcal{G}}^{L+1} - 1}{d_{\mathcal{G}} - 1} . \tag{255}$$

As we have the fact that

$$\frac{d_{\mathcal{G}}^{L+1} - 1}{d_{\mathcal{G}} - 1} \leq 2d_{\mathcal{G}}^L \ . \tag{256}$$

So we obtain

$$p_n = \mathcal{O}(d_{\mathcal{G}}^L) \ . \tag{257}$$

Hence, the regret bound for the modified RO-CRL for CB is

$$\max\{[\mathcal{R}_T \mid \mathbb{P}], [\mathcal{R}_T \mid \bar{\mathbb{P}}]\} \geq c' d_{\mathcal{G}}^L \sqrt{T} \ . \tag{258}$$

### G.3 Proof of Reward-oriented CRL Lower Bound (Theorem 4)

In Section G.1, we have shown a lower bound of $\Omega(p_n \sqrt{T})$ for the causal bandit. A similar result can be obtained for reward-oriented CRL as follows.

**Corollary 2.** *When having the knowledge of $\mathcal{G}$ and $\mathbf{H}_\infty$, there exists a causal model instance on $\mathcal{G}_{\mathrm{tc}}$ such that the expected regret of any algorithm is at least*

$$\mathcal{R}_T \ \geq \ \Omega(p\sqrt{T}) \ , \tag{259}$$

*where now $p$ is defined on transitive closure $\mathcal{G}_{\mathrm{tc}}$.*

*Proof:* We set $\theta = \mathbf{1}_n$ for both instances $I$ and $\bar{I}$. The proof then proceeds almost the same as for Theorem 7, with the only modification being the way interventions affect the utility. In this setting, each path terminating at node $j \in [n]$ contributes to the overall utility. Hence, selecting the suboptimal intervention at node $\sum_{j \in [n]} (m_{i,j} + 1)$. As a counterpart of (237), we have

$$[\mathcal{R}_t \mid \mathbb{P}] + [\mathcal{R}_t \mid \bar{\mathbb{P}}] \geq \frac{T}{2} \, \delta \sum_{i \in [n]} \sum_{j \in [n]} (m_{i,j} + 1)[\mathbb{P}(\mathcal{E}_{\mathrm{lb}}^i) + \bar{\mathbb{P}}(\mathcal{E}_{\mathrm{lb}}^{i,\mathrm{c}})] \tag{260}$$

$$\geq \frac{T}{4} \, \delta \sum_{i \in [n]} \sum_{i \in [n]} (m_{i,j} + 1) \exp(-\mathrm{D_{KL}}(\mathbb{P} \parallel \bar{\mathbb{P}})) \ , \tag{261}$$

$$= \frac{T}{4} \, \delta p \exp(-\mathrm{D_{KL}}(\mathbb{P} \parallel \bar{\mathbb{P}})) \ , \tag{262}$$

where the last formulation provides the $p$ in the lower bound ∎

Now we have shown the regret lower bound under the perfect scenario on the error $\mathbf{E}_t = \mathbf{0}$. Now we construct two more instances of causal models with $\mathbf{E}_t$ occurring adversarially. We construct two instances of the causal model on $\mathcal{G}_{\mathrm{tc}}$ and demonstrate that under specific deviations, no algorithm can distinguish between them and the initial stage.

Let us examine the parameterization of the two causal models, referred to as $I' = \{\mathbf{B}, \mathbf{B}^*, \varepsilon\}$ and $\bar{I}' = \{\mathbf{B}, \mathbf{B}^*, \bar{\varepsilon}\}$. For the existing edges in graph $\mathcal{G}_{\mathrm{tc}}$, $(i, j)$ for $i < j$ and $i, j \in [n]$, we define

$$[\mathbf{B}]_{i,j} = [\mathbf{B}^*]_{i,j} = 1 \ . \tag{263}$$

For the noises, for $I'$ and $\bar{I}'$ we define

$$[\varepsilon_i \mid I'] \sim \left\{ \begin{array}{ll} \varepsilon_0 & \text{if } i \in \mathbf{a} \\ 0 & \text{if } i \notin \mathbf{a} \end{array} \right. , \quad \text{and} \quad [\varepsilon_i \mid \bar{I}'] \left\{ \begin{array}{ll} 0 & \text{if } i \in \mathbf{a} \\ \varepsilon_0 & \text{if } i \notin \mathbf{a} \end{array} \right. , \tag{264}$$

where $\varepsilon_0$ is a constant that is defined later. And we define the parameters $\theta = \mathbf{1}_n$ for both. Thus, the only difference between the two bandit instances lies in the mean of the noises. In the first causal model, the optimal action is when all the nodes are *intervened*. In contrast, in the second causal graph model, the best action is associated with all the nodes being *not intervened*.

We first consider any algorithm with forced exploration (or an under-sampling rule). Consider for time horizon $T$ that the algorithm forced each intervention in $\mathcal{A}_0$ to perform $N_T$ times, the Theorem 2 provides the upper bound for the estimation error

$$\|\mathbf{E}_t\|_2 = \mathcal{O}\left(\sqrt{\frac{d}{N_T}}\right) \ . \tag{265}$$

We notice $\varepsilon_0$ controls the scaling of the system, and hence, controls the scaling of estimates and constants. By setting $\varepsilon_0$ such that $\|\mathbf{C}\|_2 = \|\mathbf{E}_t\| = \mathcal{O}\left(\sqrt{\frac{d}{N_T}}\right)$, we have $\varepsilon_0 = \mathcal{O}\left(\sqrt{\frac{d}{N_T}}\right)$. We can set $\mathbf{E}_t$ adversarially as

$$\mathbf{E}_t = -\mathbf{C} \, . \tag{266}$$

In such cases $\mathbf{H}_t = \mathbf{0}$, so that all estimates $\hat{\mathbf{Z}}_t = \mathbf{0}$, no information is posted to the learner, and the learner cannot distinguish between these two instances. Consequently, there must exist a bandit instance at which the algorithm plays the sub-optimal choice on each node $i$ at least $T/2$ times. We note the sub-optimal intervention blocks the causal flow from $\mathsf{an}(i)$ to $\mathsf{de}(i) \cup \{i\}$ to the reward. Hence, we have the following reward decomposition

$$[\mathcal{R}_T \mid I'] = \sum_{t=(n+1)N_T}^{T} \sum_{i \in [n]} \mathbb{1}\{i \notin \mathbf{a}_t\} \sum_{j \in \mathsf{an}(i)} \left(m_{j,i}+1+\sum_{k \in \mathsf{de}(i)} \mathbb{1}\{k \in \mathbf{a}_t\}(m_{j,k}+1)\right)\varepsilon_0 \, . \tag{267}$$

where $\mathsf{de}(i)$ is the descendants set of node $i$ and we note the term $\mathbb{1}\{k \notin \mathbf{a}_t\}$ is used to avoid counting any path more than once. By dropping the non-negative term associated with the $\mathbb{1}\{k \notin \mathbf{a}_t\}$, we can lower bound the regret by

$$[\mathcal{R}_T \mid I'] \geq \sum_{t=(n+1)N_T}^{T} \sum_{i \in [n]} \mathbb{1}\{i \notin \mathbf{a}_t\} \sum_{j \in \mathsf{an}(i)} (m_{j,i} + 1)\varepsilon_0 \, . \tag{268}$$

Similarly, we have

$$[\mathcal{R}_T \mid \bar{I}'] \geq \sum_{t=(n+1)N_T}^{T} \sum_{i \in [n]} \mathbb{1}\{i \in \mathbf{a}_t\} \sum_{j \in \mathsf{an}(i)} (m_{j,i} + 1)\varepsilon_0 \, . \tag{269}$$

By combining (268) and (269), we obtain

$$[\mathcal{R}_T \mid I'] + [\mathcal{R}_T \mid \bar{I}'] = pT\varepsilon_0 \, , \tag{270}$$

Hence, among these two causal graph instances, there will be at least one instance that incurs a regret of

$$\mathcal{R}_T \geq \Omega(pT\sqrt{d/N_T}) \, . \tag{271}$$

At the same time, the forced exploration period will incur a regret of order $nN_T$. Combining the two we have

$$\mathcal{R}_T \geq \Omega(nN_T + pT\sqrt{d/N_T}) \, , \tag{272}$$

Lastly, we need a trade-off between the two. By setting $N_T$ to be on the order

$$N_T = d^{1/3}n^{-2/3}p^{2/3}T^{2/3} \, . \tag{273}$$

This balances the two terms and provides the result.

$$\mathcal{R}_T \geq \Omega\big(d^{\frac{1}{3}}n^{\frac{1}{3}}p^{\frac{2}{3}}T^{\frac{2}{3}}\big) \, . \tag{274}$$

We note that the reward-oriented CRL method without forced exploration cannot leverage Theorem 2 to obtain meaningful regret bounds, as at all times we can have $\mathbf{H}_t = \mathbf{0}$ and the regret scales linearly with $T$ under the setting that $\varepsilon_0 = 1$.

Finally, combining the results in (259) and (274), we know that there exist a causal graph instance in $I, \bar{I}, I'$ and $\bar{I}'$ such that

$$\mathcal{R}_T \geq \Omega\big(d^{\frac{1}{3}}n^{\frac{1}{3}}p^{\frac{2}{3}}T^{\frac{2}{3}} + p\sqrt{T}\big) \, . \tag{275}$$

## H  Multiple Interventions

In this section, we state what changes are needed to adapt RO-CRL for multiple interventions per node. We focus on *soft interventions*, where *hard interventions* can be viewed as a special case of this. To maintain consistency, we redefine several key terms.

### H.1 Latent Data-generating Process and Intervention

Consider the case where there are $s$ possible distinct intervention mechanisms on each node $i \in [n]$. We denote $[\mathbf{B}^k]_i$ as the weight for intervention $k$ at node $i$, and define $\mathbf{B}^k$ as $\{\mathbf{B}^k = \{[\mathbf{B}^k]_i \mid i \in [n]\}$, the full set of such weights. We reserve $\mathbf{B}^0 = \mathbf{B}$ for observational weights. Similarly, we can define $\nu^k$ for $k \in \{0\} \cup [s]$. And we define the intervention as a vector $\mathbf{a} \in \mathbb{R}^{[n]}$ in this case as $\mathbf{a}[i] \in \{0\} \cup [s]$. The intervention space has cardinality of $|\mathcal{A}| = (s+1)^n$ instead of $2^n$. Under intervention $\mathbf{a} \in \mathcal{A}$, the $Z$ follows the SEM

$$Z = \mathbf{B_a} Z + \varepsilon \,, \tag{276}$$

where we have

$$[\mathbf{B_a}]_i = \mathbf{B}^{\mathbf{a}[i]} \,. \tag{277}$$

### H.2 Algorithm Modification

Theorem 2 requires only one intervention per node, and its sample-complexity and error bounds remain unchanged. Hence, the CRL component of the algorithm requires no modification.

For UCB selection,*we let** $[\mathbf{A}_t^k]_i$ **denote the robust estimate for each intervention $k \in \{0\} \cup [s]$ as

$$[\mathbf{A}_t^k]_i \triangleq [\mathbf{V}_{i,t}^k]^{-1}[\hat{\mathbf{Z}}_t]^\top_{\mathsf{pa}_t(i)} \mathbf{W}_{i,t}^k [\hat{\mathbf{Z}}_t]_i - \hat{\nu}^k[i] \,, \quad \forall k \in \{0\} \cup [s] \,. \tag{278}$$

where we have defined the *weighted and doubly weighted Gram matrices* as

$$\mathbf{V}_{i,t}^k \triangleq [\hat{\mathbf{Z}}_t]^\top_{\mathsf{pa}_t(i)} \mathbf{W}_{i,t}^k [\hat{\mathbf{Z}}_t]_{\mathsf{pa}_t(i)} + \mathbf{I}_n \,, \quad \text{and} \quad \tilde{\mathbf{V}}_{i,t}^k \triangleq [\hat{\mathbf{Z}}_t]^\top_{\mathsf{pa}_t(i)} \mathbf{W}_{i,t}^{k2} [\hat{\mathbf{Z}}_t]_{\mathsf{pa}_t(i)} + \mathbf{I}_n \,. \tag{279}$$

**Weight designs.** The diagonal elements for weight matrices are defined as

$$\mathbf{W}_{i,t}^k[s,s] \triangleq \mathbb{1}\{\mathbf{a}_t[i] = k\} \min\left\{ \frac{1}{\zeta_t} \,, \frac{1}{\zeta_t \|\hat{Z}_s[\mathsf{pa}_t(i)]\|_{[\tilde{\mathbf{V}}_{i,t}^k]^{-1}}} \right\} \,, \tag{280}$$

**Confidence ellipsoids.** After performing estimation in each round, we construct the following confidence ellipsoids for $k \in [s]$

$$\mathcal{C}_{i,t}^k \triangleq \left\{ \xi : \left\| \xi - [\mathbf{A}_{t-1}^k]_i \right\|_{\mathbf{V}_{i,t-1}^k [\tilde{\mathbf{V}}_{i,t-1}^k]^{-1} \mathbf{V}_{i,t-1}^k} \le \beta_{i,t}(\delta) \right\} \,, \tag{281}$$

### H.3 Changes in the Regret Bounds

**Theorem 9** (Regret upper bound). *Under Assumptions 1–3, with probability at least $1 - 4\delta$ , the average cumulative regret of RO-CRL is upper bounded by*

$$\mathcal{R}_T \le \tilde{\mathcal{O}}\big(s^{\frac{2}{3}} d^{\frac{1}{3}} n^{\frac{1}{3}} u^{\frac{2}{3}} T^{\frac{2}{3}} + u(\sqrt{sT} + \sqrt[4]{s^3 T})\big) \,, \tag{282}$$

*where we set $u = u_\mathrm{S}$ for soft interventions and $u = u_\mathrm{H}$ for hard interventions..*

The estimates $\theta_t$ for $\theta$ and the confidence ellipsoids $\mathcal{C}_{\theta,i}$ remain the same. We will skip some unimportant parts and focus on the changes under this setting. We refer the reader to [23] for detailed steps. We notice the change mainly due to the changes in the following lemma and how we choose the confidence levels. We note that when $m = 1$, the theorem reduces to the setting discussed in the main paper.

Now we discuss the changes needed for the proof steps discussed in Section F.

First, Lemma 6 and the discussion in Section F.1 still hold. Second, Lemma 7 in Section F.3 holds for all $k \in [s]$ with mild change of

$$\beta_{i,t}(\delta_t) \triangleq 1 + \sqrt{d_{i,t}} + \sqrt{2 \log(kn/\delta_t) + d_{i,t} \log(1 + \tilde{m}^2 t / d_{i,t} \zeta_t^2)} \,. \tag{283}$$

Now, we modify Lemma 10 to the following lemma.

**Algorithm 2** Reward-oriented CRL (RO-CRL) for multiple interventions

---

1: **Forced exploration.** Sample $T_0$ times for each intervention $\mathbf{a} \in \mathcal{A}_0$.
2: **for** $t = (n+1)T_0, \ldots$ **do**
3:     $\triangleright$ Under-sampling rule
4:     **if** $\mathcal{A}_t^{UE} \neq \emptyset$ **then**
5:         **Pull** $\mathbf{a}_t$ random sample from $\mathcal{A}_t^{UE}$
6:     **else**
7:         $\triangleright$ Latent recovery
8:         Update the inverse transform estimate $\mathbf{H}_t$ via (11)
9:         Estimate $\hat{\mathbf{Z}}_t$ according to $\hat{\mathbf{Z}}_t = \mathbf{H}_t \mathbf{X}_t$
10:       Update the graph estimate $\hat{\mathcal{G}}_t$ via (15)
11:       **if** hard interventions **then**
12:           Update the inverse transform estimate again using (19)
13:           Update $\hat{\mathbf{Z}}_t$ according to $\hat{\mathbf{Z}}_t = \mathbf{H}_t \mathbf{X}_t$
14:           Update the graph estimate again $\hat{\mathcal{G}}_t$ via (15)
15:       $\triangleright$ Parameter estimation
16:       Set weight matrix $\mathbf{W}_{i,t}^k$ according to (280) and $\mathbf{W}_{\theta,t}$ according to (25).
17:       Update $\mathbf{A}_t^k$ and $\theta_t$ according to (278) and (26), respectively
18:       Set $\mathbf{A}_t^0 = \mathbf{0}$ under hard intervention.
19:       $\triangleright$ UCB selection
20:       Compute $\text{UCB}_{\mathbf{a},t}$ according to (31) for $\mathbf{a} \in \mathcal{A}$.
21:       **Pull** $\mathbf{a}_{t+1} = \arg\max_{\mathbf{a} \in \mathcal{A}} \text{UCB}_{\mathbf{a},t}$
22:     Observe $X_t$ and $U(Z_t)$

---

**Lemma 13.** *If $\hat{\mathcal{G}}_t = \mathcal{G}$ or $\hat{\mathcal{G}}_t = \mathcal{G}_{\text{tc}}$ for all $t \in [\mathbb{N}]$, and $[\mathbf{A}]_i \in \mathcal{C}_{i,t}$ and $[\mathbf{A}^*]_i \in \mathcal{C}_{i,t}^*$ for all $t \in \mathbb{N}$ and $i \in [n]$ and $\theta_t \in \mathcal{C}_{\theta,t}$ for all $t \in \mathbb{N}$, then we have*

$$\sum_{t=1}^{T} \mathbb{1}\{\mathcal{A}_t^{\text{UE}} = \emptyset\} \mathbb{E}_{\mathbf{a}_t} \left| \tilde{U}(\tilde{Z}_t) - U(\hat{\mathbf{Z}}_t[:,t]) \right| \leq 2\hat{m}\mathcal{B} \, u_{\beta,T} \times \sqrt{s} \, . \tag{284}$$

*Proof:* The changes to the proof are mainly in the **Base step**, where we aim to prove the following. For node $i \in [n]$ with causal depth $L_i = 1$, we show that

$$\sum_{t=1}^{T} \mathbb{E}_{\mathbf{a}_t} \left| \tilde{Z}_t[i] - \hat{Z}_t[i] \right| \leq 2m u_{\beta,i} \mathcal{B}' \, , \tag{285}$$

where $\mathcal{B}'$ is defined as

$$\mathcal{B}' \leq \frac{4\sqrt{m\kappa_{\max}}}{\kappa_{\min}} \sqrt{T(s+1)} + \frac{8}{\kappa_{\min}} \sqrt[4]{\frac{3(s+1)^3 T}{2}} + E_1 \tag{286}$$

$$+ (s+1)\frac{4m}{\kappa_{\min}} \log\left(\frac{\kappa_{\min}}{m}\sqrt{\frac{T}{s+1}} + \alpha m^2\right) \zeta_T \, . \tag{287}$$

and $E_1$ is defined as

$$E_1 = (s+1)\left(4\frac{\sqrt{m\kappa_{\max}}}{\kappa_{\min}}\sqrt{\tau}\log\left(\sqrt{\frac{T}{s+1}} + \sqrt{\tau}\right)\right. \tag{288}$$

$$+ 4\sqrt{\frac{\alpha m^5}{\kappa_{\min}^3}} \log\left(\frac{\sqrt{\frac{1}{\tau}}\sqrt[4]{\frac{T}{s+1}} + \sqrt[4]{4} + 1}{\sqrt{\frac{1}{\tau}}\sqrt[4]{\frac{T}{s+1}} + \sqrt[4]{4} - 1}\right) \tag{289}$$

$$+ 8\tau\left(\frac{1}{\zeta_1}\sqrt{\kappa_{\max}\tau} + \alpha m^2\sqrt{\tau} + 1\right) \tag{290}$$

$$+ \frac{m}{\zeta_{(n+1)T_0} T} + \frac{2m}{3\zeta_{(n+1)T_0}} + 1 \Bigg) . \tag{291}$$

When the causal path of a node is $L_i = 1$, according to SEM defined in (93), we have the following expansion:

$$\sum_{t=1}^{T} \mathbb{E}_{\mathbf{a}_t} \left| \tilde{Z}_t[i] - \hat{Z}_t[i] \right| = \sum_{t=1}^{T} \mathbb{E}_{\mathbf{a}_t} \left| [\tilde{\mathbf{A}}_{\mathbf{a}_t,t}]_i^\top \hat{Z}_t[\mathsf{pa}(i)] - [\mathbf{A}_{\mathbf{a}_t}]_i^\top \hat{Z}_t[\mathsf{pa}(i)] \right| \tag{292}$$

$$\leq \sum_{t=1}^{T} \mathbb{E}_{\mathbf{a}_t} \sup_{b_1,b_2 \in \mathcal{C}_{i,\mathbf{a}_t,t}} \| b_1 - b_2 \|_{\mathbf{V}_{i,\mathbf{a}_t,t} [\tilde{\mathbf{V}}_{i,\mathbf{a}_t,t}]^{-1} \mathbf{V}_{i,\mathbf{a}_t,t}} \tag{293}$$

$$\times \left\| \hat{Z}_t[\mathsf{pa}(i)] \right\|_{\left[ \mathbf{V}_{i,\mathbf{a}_t,t} [\tilde{\mathbf{V}}_{i,\mathbf{a}_t,t}]^{-1} \mathbf{V}_{i,\mathbf{a}_t,t} \right]^{-1}} \tag{294}$$

$$\leq 2\hat{m}\beta_{i,T} \times \sum_{t=1}^{T} \lambda_{i,t} , \tag{295}$$

where we define

$$\lambda_{i,t} \triangleq \frac{\sqrt{\lambda_{\max}\left( \tilde{\mathbf{V}}_{i,\mathbf{a}_t,t} \right)}}{\lambda_{\min}\left( \mathbf{V}_{i,\mathbf{a}_t,t} \right)} , \tag{296}$$

∎

Then, the weights we defined are bounded in the range $[\frac{1}{\zeta_t \hat{m}}, \frac{1}{\zeta_t}]$ for $t \in \mathbb{N}$. Leverage these bounds if we define the following constants. Note, we drop the index $t$ for these constants, but they are a variable of time.

$$\kappa'_{\min} \triangleq \frac{1}{\zeta_t \hat{m}} \kappa_{\min} \quad \text{and} \quad \kappa'_{\max} \triangleq \frac{1}{\zeta_t} \kappa_{\max} , \tag{297}$$

$$\tilde{\kappa}_{\min} \triangleq \frac{1}{\zeta_t^2 \hat{m}^2} \kappa_{\min} \quad \text{and} \quad \tilde{\kappa}_{\max} \triangleq \frac{1}{\zeta_t^2} \kappa_{\max} , \tag{298}$$

$$m' \triangleq \frac{1}{\sqrt{\zeta_t}} \hat{m} \quad \text{and} \quad \tilde{m} \triangleq \frac{1}{\zeta_t} . \tag{299}$$

In order to proceed, we need upper and lower bounds for the maximum and minimum singular values of $\mathbf{U}_{i,\mathbf{a}_t,t}$. These bounds depend on the number of non-zero rows of $\mathbf{U}_{i,\mathbf{a}_t,t}$ matrices, which equals the values of the random variable $N_{i,\mathbf{a}_t,t}$. Let us define the constant

$$\gamma_n \triangleq \max\left\{ \alpha m^2 \sqrt{n}, \alpha^2 m^2 \right\} , \tag{300}$$

$$\gamma'_n \triangleq \max\left\{ \alpha m'^2 \sqrt{n}, \alpha^2 m'^2 \right\} , \tag{301}$$

$$\tilde{\gamma}_n \triangleq \max\left\{ \alpha \tilde{m}^2 \sqrt{n}, \alpha^2 \tilde{m}^2 \right\} , \quad \forall n \in [T] . \tag{302}$$

Then for every $t \in [T]$, and $n \in [t]$, we define the error events corresponding to the maximum and minimum singular values of $\mathbf{U}_{i,t}$ and $\tilde{\mathbf{U}}_{i,t}$ as

$$\mathcal{E}_{i,n,t} \triangleq \Bigg\{ N_{i,t} = n \quad \text{and} \quad \Big\{ \sigma_{\min}\left( \mathbf{U}_{i,t} \right) \leq \sqrt{\max\left\{ 0, n\kappa'_{\min} - \gamma'_n \right\}}$$

$$\text{or } \sigma_{\max}\left( \mathbf{U}_{i,t} \right) \geq \sqrt{n\kappa'_{\max} + \gamma'_n} \Big\} \Bigg\} , \tag{303}$$

$$\tilde{\mathcal{E}}_{i,n,t} \triangleq \Bigg\{ N_{i,t} = n \quad \text{and} \quad \Big\{ \sigma_{\min}\left( \tilde{\mathbf{U}}_{i,t} \right) \leq \sqrt{\max\left\{ 0, n\tilde{\kappa}_{\min} - \tilde{\gamma}_n \right\}}$$

$$\text{or } \sigma_{\max}\left( \tilde{\mathbf{U}}_{i,t} \right) \geq \sqrt{n\tilde{\kappa}_{\max} + \tilde{\gamma}_n} \Big\} \Bigg\} , \tag{304}$$

Similarly, we can define $\mathcal{E}^*_{i,n,t}$ and $\tilde{\mathcal{E}}^*_{i,n,t}$ by replacing $N_{i,t}$ and $\mathbf{U}_{i,t}$ (or $\tilde{\mathbf{U}}_{i,t}$) by $N^*_{i,t}$ and $\tilde{\mathbf{U}}_{i,t}$ (or $\tilde{\mathbf{U}}^*_{i,t}$), respectively.

As a result of [23, Lemma 8], if we define the union error event $\mathcal{E}_{i,\cup}$ as

$$\mathcal{E}_{i,\cup} \triangleq \{\exists\,(t,n) : t \in [T], n \in [t],\ \mathcal{E}_{i,n,t} \text{ or } \mathcal{E}^*_{i,n,t}\} \,. \tag{305}$$

We have

$$\mathbb{P}(\mathcal{E}_{i,\cup}) \leq 2NT(T+1)(d_i+1)\exp\left(-\frac{3\alpha^2}{16}\right) \,. \tag{306}$$

**Bounding term** $\mathbb{E}\left[\mathbb{1}\{\mathcal{E}_{i,\cup}\}\sum_{t=1}^{T}\lambda_{i,t}\right]$. Same as [23], we use the fact that $\zeta_t$ increase with time $t$ and obtain

$$\mathbb{E}\left[\mathbb{1}\{\mathcal{E}_{i,\cup}\}\sum_{t=1}^{T}\sqrt{m^2 t + 1}\right] < \frac{m}{\zeta_{(n+1)T_0}T} + \frac{2m}{3\zeta_{(n+1)T_0}} + 1 \,. \tag{307}$$

**Bounding** $\mathbb{E}\left[\mathbb{1}\{\mathcal{E}^c_{i,\cup}\}\sum_{t=1}^{T}\lambda_{i,t}\right]$. We define the function $h(x)$ as

$$h(x) \triangleq \frac{\sqrt{x\tilde{\kappa}_{\max} + \tilde{\gamma}_n + 1}}{\max\{0, x\kappa'_{\min} - \gamma'_n\} + 1} \,, \quad x > 0 \,. \tag{308}$$

And we define the $g$ function when $x > \tau$ as follows.

$$g(x,\zeta_t) \triangleq \frac{\sqrt{x\kappa_{\max} + \alpha\hat{m}^2\sqrt{x}}}{x\kappa_{\min}/\hat{m} - \alpha\hat{m}^2\sqrt{x}} + \frac{\zeta_t}{x\kappa_{\min}/\hat{m} - \alpha\hat{m}^2\sqrt{x}} \,. \tag{309}$$

We have the following theorem to show the monotonicity and relation of $h(x)$ and $g(x)$.

**Lemma 14.** *[23, Lemma 10] $h(x,\zeta_t)$ and $g(x,\zeta_t)$ are both decreasing functions of $x$ when $x > \tau$ and $h(x) < g(x)$, where $\tau$ is defined as $\frac{\alpha^2\hat{m}^6}{\kappa^2_{\min}}$.*

Now we are ready to bound the last term

$$\mathbb{E}\left[\mathbb{1}\{\mathcal{E}^c_{i,\cup}\}\sum_{t=1}^{T}\lambda_{i,t}\right] \leq \mathbb{E}\sum_{t=1}^{T} h(N_{i,\mathbf{a}_t,t}) \,. \tag{310}$$

We define the set of time indices at which the chosen actions are under-explored as

$$\mathcal{H}_i \triangleq \{t \in [T] \mid N_{i,\mathbf{a}_t,t} \leq 4\tau\} \,. \tag{311}$$

It can be readily verified that $|\mathcal{H}_i| \leq 8\tau$. Furthermore, when $x \in \mathcal{H}_i$, we have

$$h(x) \leq \frac{1}{\zeta_{(n+1)T_0}}\sqrt{\kappa_{\max}\tau + \alpha\hat{m}^2\sqrt{\tau}} + 1 \,, \ x \leq \tau \,. \tag{312}$$

Then we can bound the summation when $\mathcal{H}_i$ occurs as follows.

$$\mathbb{E}\sum_{t=1}^{T}\mathbb{1}\{t \in \mathcal{H}_i\}h(N_{i,\mathbf{a}_t}(t)) \leq 8\tau\left(\frac{1}{\zeta_{(n+1)T_0}}\sqrt{\kappa_{\max}\tau + \alpha\hat{m}^2\sqrt{\tau}} + 1\right) \,. \tag{313}$$

Now we only need to bound the remaining part when $t \notin \mathcal{H}_i$

$$\mathbb{E}\sum_{t=1}^{T}\mathbb{1}\{t \in \mathcal{H}^c_i\}h(N_{i,\mathbf{a}_t,t}) \,. \tag{314}$$

Note that when $t \in \mathcal{H}^c_i$, we have $N_{i,\mathbf{a}_t,t} > \tau$ and

$$h(N_{i,\mathbf{a}_t,\zeta_t}) \leq g(N_{i,\mathbf{a}_t,t},\zeta_t) \,. \tag{315}$$

Now we discuss the major changes. We define the number of times that node $i$ is under mechanism $k \in \{0\} \cup [s]$ at time $t \in [\mathbb{N}]$ as

$$N_{i,k,t} = \sum_{s=1}^{t} \mathbb{1}\{\mathbf{a}_s[i] = k\} \tag{316}$$

Using the above results and noting that $\mathcal{H}_i^c$ excludes those samples from initial rounds, we obtain

$$\sum_{t=1}^{T} \mathbb{1}\{t \in \mathcal{H}_i^c\} h(N_{i,\mathbf{a}_t,t}) \le \sum_{t=1}^{T} \mathbb{1}\{t \in \mathcal{H}_i^c\} g(N_{i,\mathbf{a}_t,t}) \tag{317}$$

$$\le \sum_{k \in [s]} \sum_{n=4\tau+1}^{N_{i,k,T}} g(n, \zeta_t) . \tag{318}$$

We bound the discrete sums through integrals and define

$$G_\tau(y) = \int_{x=4\tau}^{y} g(x)dx , \quad y \ge 4\tau . \tag{319}$$

Since $g(x)$ is a positive, decreasing function, for any $k \in \mathbb{N}, k \ge 4\tau + 1$ we have

$$\sum_{n=4\tau+1}^{k} g(n) \le \int_{x=4\tau}^{k} g(x)dx = G_\tau(k) . \tag{320}$$

Then, the summation in (318) is upper bounded by

$$\sum_{k \in [s]} \sum_{n=4\tau+1}^{N_{i,k,t}} g(n) \le \sum_{k \in [s]} G_\tau(N_{i,k,T}) . \tag{321}$$

Since $g(x)$ is positive and decreasing, and $G(y)$ is defined as an integral of the $g$ function with a positive first derivative and negative second derivative, it can be deduced that $G$ is a concave function. Thus, we have

$$\sum_{k \in [s]} G_\tau(N_{i,k,T}) \le (s+1)G_\tau\left(\frac{T}{s+1}\right) . \tag{322}$$

Next, we proceed to establish an upper bound for the function $G$, which can be upper bounded as

$$G_\tau\left(\frac{T}{s+1} + 4\tau\right) = \int_{x=4\tau}^{\frac{T}{s+1}+4\tau} g(x)\mathrm{d}x \tag{323}$$

$$\le \int_{x=4\tau}^{\frac{T}{s+1}+4\tau} \sqrt{\frac{m^2 \kappa_{\max}}{\kappa_{\min}}} \frac{1}{\sqrt{x\kappa_{\min}} - \sqrt{\tau\kappa_{\min}}}\mathrm{d}x$$

$$+ \int_{x=4\tau}^{\frac{T}{s+1}+4\tau} \sqrt{\alpha m^2(1 + \frac{m\kappa_{\max}}{\kappa_{\min}})} \frac{x^{1/4}}{x\kappa_{\min}/m - \alpha m^2\sqrt{x}}\mathrm{d}x$$

$$+ \int_{x=4\tau}^{\frac{T}{s+1}+4\tau} \frac{\zeta_T}{x\kappa_{\min}/m - \alpha m^2\sqrt{x}}\mathrm{d}x \tag{324}$$

$$\le 2\frac{\sqrt{m\kappa_{\max}}}{\kappa_{\min}}\left(\sqrt{\frac{T}{s+1}} + \sqrt{\tau}\log\left(\sqrt{\frac{T}{s+1}} + \sqrt{\tau}\right)\right)$$

$$+ \frac{4}{\kappa_{\min}}\sqrt[4]{\frac{T}{s+1}} + 2\sqrt{\frac{\alpha m^5}{\kappa_{\min}^3}}\log\left(\frac{\sqrt{\frac{1}{\tau}}\sqrt[4]{\frac{T}{s+1}} + \sqrt[4]{4} + 1}{\sqrt{\frac{1}{\tau}}\sqrt[4]{\frac{T}{s+1}} + \sqrt[4]{4} - 1}\right)$$

$$+ \frac{2m\log\left(\frac{\kappa_{\min}}{m}\sqrt{\frac{T}{s+1}} + \alpha m^2\right)}{\kappa_{\min}}\zeta_T . \tag{325}$$

where (324) is due to the inequality $\sqrt{x+y} \leq \sqrt{x} + \sqrt{y}$ and $\sqrt{x-y} \geq \sqrt{x} - \frac{y}{\sqrt{x}}$ when $x \geq y$, and we use closed-form integral and discard positive terms in (325). Combining the results in (307), (313), and (325), let $E_1$ denote the accumulation of terms that exhibit at most logarithmic growth rates with respect to $T$ and $\zeta_T$.

$$E_1 = (s+1)\left( 4\frac{\sqrt{m\kappa_{\max}}}{\kappa_{\min}}\sqrt{\tau}\log\left( \sqrt{\frac{T}{s+1}} + \sqrt{\tau} \right) \right. \tag{326}$$

$$+ 4\sqrt{\frac{\alpha m^5}{\kappa_{\min}^3}}\log\left( \frac{\sqrt{\frac{1}{\tau}}\sqrt[4]{\frac{T}{s+1}} + \sqrt[4]{4} + 1}{\sqrt{\frac{1}{\tau}}\sqrt[4]{\frac{T}{s+1}} + \sqrt[4]{4} - 1} \right) \tag{327}$$

$$+ 8\tau\left( \frac{1}{\zeta_1}\sqrt{\kappa_{\max}\tau + \alpha m^2\sqrt{\tau}} + 1 \right) \tag{328}$$

$$\left. + \frac{m}{\zeta_{(n+1)T_0}T} + \frac{2m}{3\zeta_{(n+1)T_0}} + 1 \right). \tag{329}$$

Therefore, the final result for the bound is

$$\mathbb{E}\left[ \sum_{t=1}^T \lambda_{i,t} \right] \leq \frac{4\sqrt{m\kappa_{\max}}}{\kappa_{\min}}\sqrt{T(s+1)} + \frac{8}{\kappa_{\min}}\sqrt[4]{\frac{3(s+1)^3 T}{2}} + E_1 \tag{330}$$

$$+ (s+1)\frac{4m}{\kappa_{\min}}\log\left( \frac{\kappa_{\min}}{m}\sqrt{\frac{T}{s+1}} + \alpha m^2 \right)\zeta_T. \tag{331}$$

And the rest of the proof remains almost the same, with only a difference in confidence constants.

## I   Remark on *do* interventions

In this section, we discuss the results for *do* interventions, a further restricted type of intervention. It is a subclass of hard interventions in which intervention on node $i$ removes both ancestral connections and the randomness of $Z[i]$ and sets the noise variable $\varepsilon^*[i]$ to a fixed known value.

**Remark 4** (*do* interventions)**.** *Under do interventions, there exists a set of $n+1$ interventions such that getting one sample from each suffices to construct $\mathbf{H}_\infty$ for which $\mathbf{H}_\infty\mathbf{G}$ is a full-rank diagonal matrix. The resulting regret bounds depend on the causal model assumptions. For instance, in the standard setting where each latent variable $Z[i] \in [k]$ for $i \in [k]$ with intervention space $\mathcal{A} = [k]^n$ and the conditional distributions $p_i(Z[i] \mid Z[\mathrm{pa}[i]])$ are arbitrary, all latent variables contribute to the utility node. In this case, the problem reduces to a multi-armed bandit, with a matching regret bounds scale as $\Theta(\sqrt{k^n T})$ even without the latent recovery.*

## J   Broader impacts, Limitations and Further Discussions

This paper is purely theoretical, and the experiments only use synthetic data, so we are not aware of any negative societal impacts. Our principal limitation is the assumption of linearity. Specifically, we assume linear structural-equation models, linear transformation models, and linear utility functions. However, these assumptions are standard in both causal bandit and causal representation learning work under soft interventions, with the non-linear setting remaining largely unexplored. Extending reward-oriented CRL beyond linear models, therefore, remains an important avenue for future research.

We note that even in the simpler contexts (e.g., CRL or CB), the linear versus nonlinear models are investigated extensively. Here, we outline how each component can be individually generalized. To provide a concrete methodology for extensions, we first note that our current analysis consists of two parts: (i) analyzing finite-sample guarantees of a CRL algorithm, and (ii) analyzing how the CRL guarantees translate to reward guarantees. We discuss how the same decomposition can address nonlinear models as well:

- **Latent SEM:** It is possible to generalize the algorithms and performance guarantees to nonlinear SEMs. While the overall RO-CRL framework and pipeline remain intact, such a generalization has two distinct implications: one for the CRL guarantees and another for the CB guarantees. On the CRL side, the subroutine can be replaced with methods designed for general SEMs that also provide finite-sample guarantees (e.g., [4]). As a result, the performance guarantees stated in Theorem 2 would need to be adjusted accordingly. On the bandit side, the UCB rule and reward estimators can similarly be replaced with their counterparts developed for nonlinear causal bandits (e.g., [20, 24]). Such generalization, however, is highly non-trivial and even in the simpler contexts (e.g., standalone CRL and CB), the linear and nonlinear settings are investigated separately, each with its exclusive technical challenges.

- **Transformation:** There are existing results for CRL under nonlinear transformations [10–12]. However, finite-sample analysis of CRL under general transformations is still an open problem. Nevertheless, we note that any future finite-sample result can be integrated into our RO-CRL framework to extend it to nonlinear transformations: Given that the estimator provides a graph and variables recovery error bound similar to Theorem 2, these estimates and error bounds can be used by the robust causal bandit algorithm. Similar to SEM generalization, as long as the setting does not violate the CB assumptions, the overall pipeline would work.

- **Utility:** Our utility-function assumption is a direct extension of the latent SEM one. Specifically, the framework can be modularly generalized to accommodate non-linear utilities when adopting the robust non-linear estimates in bandit settings. The key open question, therefore, is the same one that arises for latent-SEM causal-bandit methods: How robust are these algorithms to violations in the underlying variable?

In all three cases, the structure of the analysis, error control in CRL, followed by robust bandit optimization, remains valid. The main requirement is that the substituted components provide compatible error guarantees.

**Potential use cases for RO-CRL .**    Finally, we list some domains where latent interventions are plausible and RO-CRL could be applied.

- *Robotics:* Consider a robot operating in an environment and observed through high-dimensional sensory data such as images. Interventional CRL enables recovery of interpretable latent factors (e.g., joint angles) that causally generate visual data. CRL is particularly suitable for robotics in vision-based contexts because high-dimensional observations (images) obscure the causal variables and robots have known or assumed causal structure (e.g., a joint causes an end-effector position). In robotics, interventions on the latent variables can be achieved physically or via simulation. Physical interventions include, for instance, directly changing a joint angle or velocity (e.g., move joint 3 by $+5°$) and moving the robot gripper to a fixed position while leaving other joints free. In this context, an example downstream objective could be moving the robot arm to a specific configuration using only the utility values and learned latent factors as control input.

- *Genomics:* In gene regulatory networks, we observe high-dimensional expression data, typically measured from bulk or single-cell RNA sequencing. These arise from lower-dimensional latent variables representing unobserved biological drivers such as transcription factor activities, regulatory protein states, or signaling pathway activations. CRL is a natural fit for gene regulatory analysis because gene expression is influenced by a structured, latent causal process involving transcriptional and post-transcriptional regulation. Standard representation learning techniques fail to recover biologically meaningful variables, often mixing causal and non-causal factors. Node-level interventions in the CRL framework correspond to perturbing individual latent regulatory variables. In gene networks, such interventions are biologically realizable via gene knockouts or CRISPR interference to silence specific genes or transcription factors and overexpression systems to activate regulators, to name a few. In this context, an example downstream objective could be to optimize a certain type of biological response over the set of some possible genetic modifications.

