# OpenReview forum: "Reward-oriented Causal Representation Learning"
_NeurIPS.cc/2025/Conference — NeurIPS 2025 poster_

### Official Review · Reviewer_w3y6 · 2025-06-20

**Clarity:** 2
**Significance:** 2
**Originality:** 3
**Rating:** 5
**Confidence:** 2

**Summary:**

The paper proposes a new reward-oriented causal representation learning framework where the goal is to learn a latent representation well enough for some downstream task. The framework is formally described for soft and hard interventions, with corresponding identifiability guarantees under a linear SEM.

The RO-CRL algorithm is then presented as a method for recovering latent variables and discovering the optimal interventions. The algorithm first tests each atomic intervention a few times to get initial estimates of the inverse of $\mathbf{G}$ and $\mathcal{G}$. Using the estimates and reward, the next intervention is selected according to a UCB-based rule. The authors then provide regret upper bounds for their algorithm and a general lower bound on CRL algorithms, showing that their method is close to optimal.

**Questions:**

- Given the above, what would be an example of a real world scenario where we can intervene on each of the nodes in the causal graph but we still need to learn the latent variables occurs?
  - In the robot example given in the introduction, it feels like if we are able to intervene on each of the joint angles, then we'd have access to their values.
- How does the recovery of latent factors compare with the normal CRL setting?
- How computationally expensive is the algorithm?

It is clear that considerable theoretical work was done. If the motivation and the setup were made clearer, I would be willing to increase my score.

**Ethical Concerns:**

["NO or VERY MINOR ethics concerns only"]

**Final Justification:**

The paper contains significant theoretical work. From the rebuttal, my main issue regarding the viability of the setting has been addressed and I appreciate the additional explanations. The fixing of typos and additional discussion of relevant points will strengthen the submission: I have thus raised my score from 3->5.

**Limitations:**

Yes.

**Paper Formatting Concerns:**

None.

**Quality:**

3

**Strengths And Weaknesses:**

With the caveat that it is possible I misunderstood some parts of the submission as it was quite technical:

**Strengths**
- The paper is well-structured, though could benefit from additional diagrams to better explain the algorithm/idea.
- The idea is novel and interesting, potentially sparking a new direction for future CRL research.
- The paper contains significant theoretical contributions consisting of both novel identifiability results and new bounds.
- Even without the representation learning part, the work contains state-of-the-art regret bounds for the causal bandit setting.

**Weaknesses**
- The motivation behind the idea could be clearer. While the notion of concentrating on the downstream task instead of perfect identifiability makes sense, it is unclear in what scenarios you'd be able to intervene on every single node in the causal SEM but not have access to the latent variables. In a sense, being able to intervene in this manner implies knowledge of at least what the latent variables are.
- The algorithm could use further experimental evidence.
- The restriction to linear SEM, transformation models and utility functions is somewhat restrictive.

There is a large number of typos/grammatical mistakes (e.g. just from the first two pages):
- "to the extended needed" -> "to the extent needed"
- "Aiming at establish the possibility/impossibility results" -> "Aimed at establishing possibility/impossibility results"
- "enable more improved" -> "enable improved"

---

> ### Author Rebuttal · Authors · 2025-07-31
>
> **Q1. Setting and motivation.** The observation that having access to single-node interventions on latent variables means at least knowing the variable labels is astute. In fact, in CRL, the recovered variables are precisely identified with their corresponding interventions. This enables the recovery of variables with known names/semantics but unknown *values* from high-dimensional observations, which leads to the second part of the question: When do latent variables exist but not directly observable?
>
> These concerns broadly apply to interventional CRL; we will clarify them and include a summary in the revised manuscript.
>
> Interventional CRL has become one of the key branches of CRL. Its literature initially focused on the fundamental identifiability guarantees under stylized models (e.g., single-node interventions), and has since grown in various dimensions (e.g., unknown multi-node interventions). Whether the latent variables can be intervened (either individually or as a group) depends on the application domain. Below, we briefly review two such domains in which interventions on latent variables are plausible.
> - **Robotics:** Consider a robot operating in an environment and observed through high-dimensional sensory data such as images. Interventional CRL enables recovery of interpretable latent factors (e.g., joint angles) that causally generate visual data. CRL is particularly suitable for robotics in vision-based contexts because *high-dimensional observations* (images) obscure the causal variables and *robots* have known or assumed causal structure (e.g., a joint​ causes an end-effector position​). In robotics, interventions on the latent variables can be achieved physically or via simulation. Physical interventions include, for instance, directly changing a joint angle or velocity (e.g., move joint 3 by +5°) and moving the robot gripper to a fixed position while leaving other joints free.
> - **Genomics:** In gene regulatory networks, we observe high-dimensional expression data, typically measured from bulk or single-cell RNA sequencing. These arise from lower-dimensional latent variables representing unobserved biological drivers such as transcription factor activities, regulatory protein states, or signaling pathway activations. CRL is a natural fit for gene regulatory analysis because gene expression is influenced by a structured, latent causal process involving transcriptional and post-transcriptional regulation. Standard representation learning techniques fail to recover biologically meaningful variables, often mixing causal and non-causal factors. Node-level interventions in the CRL framework correspond to perturbing individual latent regulatory variables. In gene networks, such interventions are biologically realizable via *gene knockouts* or *CRISPR interference* to silence specific genes or transcription factors and *overexpression systems* to activate regulators, to name a few.
>
> **Q2. Comparison with normal CRL.** CRL identifiability guarantees are generally presented for the asymptotic regime of having an infinite amount of data. In our RO-CRL framework, we have to move away from this stylized assumption and work with a finite number of samples. As a special case of our setting, if we set our horizon $T \to \infty$, the latent recovery of our RO-CRL recovers the state-of-the-art known identifiability results under both soft and hard interventions (e.g., see [5, Algorithm 1]). With finite samples, however, RO-CRL prioritizes maximizing the cumulative utility. While shrinking the confidence radii requires progressively tightening the latent factor recovery error bounds, it also requires adaptive selection to track the uncertainty in estimates. Therefore, the overall latent error bounds are looser than having used all samples in a normal CRL algorithm. Conversely, if we were to separate CRL and bandit steps, i.e., first do CRL for a fixed portion of the samples then apply bandit, the cumulative regret can grow linearly in $T$ due to constant recovery errors after CRL pure exploration.
>
> **Q3. Computational complexity.** Computational cost per step can be broken down into two parts:
> 1. **CRL:** CRL routine depends on matrix inversions ($O(d^3)$), which can be expensive for large $d$. However, since the transformation is linear, we can detect the supporting subspace and project the samples to it, effectively reducing the observation dimension $d$ to $n$, and yielding an overall per-step complexity of $O(n^4)$.
> 2. **UCB-based selection:** The computational bottleneck is the intervention‑selection step: the UCB is intractable on general causal graphs [25, §3.3]. In practice, we may adopt the following procedure.
>     - Non‑negative edge weights and noise (shifted system): Because all rewards are monotone in their parents’ variables, to calculate the UCB, we can maximize nodes sequentially in causal order and use the closed‑form in linear bandit. The resulting worst‑case complexity is $O(n\*d\_{\hat G\_t}^3)$, where $n$ is the number of latent dimensions and $d\_{\hat G\_t}$ is the maximum in-degree in $\hat G\_t$.
>     - UCB upper bounds: [25] proposes techniques that efficiently compute an upper bound on the UCB, which we could incorporate into our approach. But the upper bound is loose and their robustness to imperfect latent recovery is unknown.
>
> **W1. Intervention on the latent space.** We appreciate the reviewer’s careful attention to the viability of interventions. As discussed in response to Q1, viability of interventions highly depends on the application and there are important ones that naturally fit the scope of interventional CRL and are subjects of ongoing studies via CRL.
>
> **W2. Further experimental evidence.** We provide additional experiments to evaluate the performance of our algorithm across different $n$ and $d$. See Tables R1–4 in our response to reviewer FrNT. Overall, we observe that the regret (Theorem 3) and latent recovery (Theorem 2) results are both validated across variations in problem parameters. We also perform experiments to see the robustness of our algorithm to the assumptions we adopt (Assumptions 1–3). See Tables R5–7 in our response to reviewer FrNT. We observe that violation of any of these assumptions results in linear regret, which aligns with theoretical expectations.
>
> **W3. Linear SEM, transformation, and rewards.** While linear models represent an important class of causal models, transformations, and rewards, we do agree with the reviewer that generalization to nonlinear settings is also important. We note that even in the simpler contexts (e.g., CRL or CB), the linear versus nonlinear models are investigated extensively. Here, we outline how each component can be individually generalized. To provide a concrete methodology for extensions, we first note that our current analysis consists of two parts: (i) analyzing finite-sample guarantees of a CRL algorithm, and (ii) analyzing how the CRL guarantees translate to reward guarantees. We discuss how the same decomposition can address nonlinear models as well:
> - **Latent SEM:** It is possible to generalize the algorithms and performance guarantees to nonlinear SEMs. While the overall RO-CRL framework and pipeline remain intact, such a generalization has two distinct implications: one for the CRL guarantees and another for the CB guarantees. Setting aside the technical details, here is a high-level description of how this generalization works. On the CRL side, the CRL subroutine can be replaced with methods designed for general SEMs that also provide finite-sample guarantees (e.g., [11]). As a result, the performance guarantees stated in Theorem 2 would need to be adjusted accordingly. On the bandit side, the UCB rule and reward estimators can similarly be replaced with their counterparts developed for nonlinear causal bandits (e.g., [20, 23]). Such generalization, however, is highly non-trivial and even in the simpler contexts (e.g., standalone CRL and CB), the linear and nonlinear settings are investigated separately, each with its exclusive technical challenges.
> - **Transformation:** There are existing results for CRL under nonlinear transformations [9, 10, 13]. However, finite-sample analysis of CRL under general transformations is – to our knowledge – still an open problem. Nevertheless, we note that any future finite-sample result can be integrated into our RO-CRL framework to extend it to nonlinear transformations: Given that the estimator provides a graph and variables recovery error bound similar to Theorem 2, these estimates and error bounds can be used by the robust causal bandit algorithm. Similar to SEM generalization, as long as the setting does not violate the CB assumptions, the overall pipeline would work.
> - **Utility:** Our utility‑function assumption is a direct extension of the latent SEM one. Specifically, the framework can be modularly generalized to accommodate non‑linear utilities when adopting the robust non-linear estimates in bandit settings. The key open question, therefore, is the same one that arises for latent‑SEM causal‑bandit methods: How robust are these algorithms to violations in the underlying variable?
>
> In all three cases, the structure of the analysis – error control in CRL followed by robust bandit optimization – remains valid. The main requirement is that the substituted components provide compatible error guarantees.
>
> **Typos.** We thank the reviewer, we will do a thorough proofread to clean the paper.
>
> **Algorithm/idea exposition.** If the paper is accepted, we will use the extra page in camera‑ready to add a high-level description of the data pipeline to make the setting clearer and highlight the algorithm’s main steps and central idea. The diagram will include the following parts:
> - CRL subroutine that is called after each exploratory sample
> - The key idea of UCB-selection
> - Annotation on the link between estimates and original parameters
> - Idea behind under-sampling rule

---

> > ### Comment · Reviewer_w3y6 · 2025-08-01
> >
> > I thank the authors for their thorough clarifications. The examples and additional explanations are particularly helpful in addressing my largest concern with the paper.
> >
> > If I understand correctly, through this framework, one can learn a generalizable mapping from high-dimensional observations to latent variables. This mapping can be learned from say a simulator (where the mapping is not particularly useful as we have access to the latent variables) but then applied in real world systems using the previously simulated robot (say to estimate joint angles, velocity, etc. from a camera-feed).
> >
> > While the algorithm still seems expensive in larger domains, I understand that this is somewhat unavoidable and the theoretical contributions are significant regardless
> >
> > The additional experiments as well as description of how the work could be extended to non-linear systems are appreciated.
> >
> > I will increase my score accordingly.

---

### Official Review · Reviewer_J78L · 2025-06-26

**Clarity:** 2
**Significance:** 3
**Originality:** 3
**Rating:** 4
**Confidence:** 3

**Summary:**

This paper introduces a novel framework called Reward-oriented Causal Representation Learning (RO-CRL), which rethinks the goal of causal representation learning (CRL). Rather than aiming for perfect identification of latent variables and their causal relations, the authors propose to learn the minimal representation sufficient to optimize a downstream reward function. The paper formalizes this notion in the context of linear structural equation models (SEMs) and linear transformations from latent to observed space, with linear utility functions. A key contribution is a regret-minimizing algorithm that adaptively explores interventions to identify the optimal intervention with respect to the unknown reward. The algorithm is analyzed under both soft and hard interventions, achieving nearly tight regret bounds. The paper also presents finite-sample identifiability results and shows that the proposed framework generalizes and improves upon causal bandit methods.

**Questions:**

Have the authors considered how the RO-CRL framework could be extended to non-linear SEMs or transformations? Even a qualitative discussion on potential challenges or future directions would strengthen the broader relevance of the work.

The paper would benefit from a more visible main-text discussion of empirical results, perhaps with illustrative toy examples that demonstrate key phenomena (e.g., how coarse representations suffice for specific rewards). Are the empirical claims consistent across different intervention types and graph structures?

The algorithm relies on known bounds for parameter magnitudes and noise levels. How sensitive is RO-CRL to violations or inaccuracies in these bounds?

**Ethical Concerns:**

["NO or VERY MINOR ethics concerns only"]

**Final Justification:**

As posted on the rebuttal, the reviewers have provided an exhaustive reply, addressing some of the doubts I initially had.

**Limitations:**

Yes

**Quality:**

2

**Strengths And Weaknesses:**

The paper demonstrates high quality in its theoretical development, offering a detailed analysis of the RO-CRL algorithm. The manuscript is overall well-written. However, the structure of the paper is somewhat unusual and a bit confusing. For instance, the notation is explained at the end of the introduction.  Regarding significance, the shift from complete identifiability to task-oriented sufficiency is relevant, addressing common limitations of traditional CRL approaches when deployed in real-world decision-making systems. The originality of the work lies in formalizing the objective of utility-maximizing representation learning within a causal inference framework and in developing a regret-minimizing algorithm that explicitly avoids unnecessary modeling detail.

However, there are some limitations. The main restriction is the reliance on linearity assumptions across the SEM, transformation, and utility function, which may reduce applicability to real-world domains. While these assumptions simplify the analysis, a discussion on potential extensions to more general settings would be relevant. The empirical evaluation is limited and relegated to the appendix, which makes it harder to assess the practical effectiveness of the method. Overall, there is a gap between theory and empirical interpretation.

---

> ### Author Rebuttal · Authors · 2025-07-31
>
> **Q1. Extension to nonlinear setting.** Thank you for the chance to discuss the potential to extend to nonlinear settings. While linear models represent an important class of causal models, transformations, and rewards, we do agree with the reviewer that generalization to nonlinear settings is also important. Our framework sets the foundations for RO-CRL and we outline how it can accommodate nonlinear SEM, nonlinear transformation, and nonlinear rewards. To provide a concrete methodology for extensions, we first note that our current analysis consists of two parts: (i) analyzing finite-sample guarantees of a CRL algorithm, and (ii) analyzing how the CRL guarantees translate to reward guarantees. We discuss how the same decomposition can address nonlinear models as well:
> - **Latent SEM:** It is possible to generalize the algorithms and performance guarantees to nonlinear SEMs. While the overall RO-CRL framework and pipeline remain intact, such a generalization has two distinct implications: one for the CRL guarantees and another for the CB guarantees. Setting aside the technical details, here is a high-level description of how this generalization works. On the CRL side, the CRL subroutine can be replaced with methods designed for general SEMs. For example, [11] provides an algorithm along with finite-sample guarantees applicable to general SEMs. As a result, the performance guarantees stated in Theorem 2 would need to be adjusted accordingly. On the bandit side, the UCB rule and reward estimators can similarly be replaced with their counterparts developed for nonlinear causal bandits (for instance, those based on Gaussian processes or neural networks [20, 23]). Such generalization, however, is highly non-trivial and even in the simpler contexts (e.g., standalone CRL and CB), the linear and nonlinear settings are investigated separately, each with its exclusive technical challenges.
> - **Transformation:** There are existing results for CRL under nonlinear transformations [9, 10, 13]. However, finite-sample analysis of CRL under general transformations is – to our knowledge – still an open problem. Nevertheless, we note that any future finite-sample result can be integrated into our RO-CRL framework to extend it to nonlinear transformations: Given that the estimator provides a graph and variables recovery error bound similar to Theorem 2, these estimates and error bounds can be used by the robust causal bandit algorithm. Similar to SEM generalization, as long as the setting does not violate the CB assumptions, the overall pipeline would work.
> - **Utility:** Our utility‑function assumption is a direct extension of the latent SEM assumption. Specifically, the framework can be modularly generalized to accommodate non‑linear utilities when adopting the robust non-linear estimates in bandit settings. The key open question, therefore, is the same one that arises for latent‑SEM causal‑bandit methods: How robust are these algorithms to violations in the underlying variable?
>
> **Q2. Main-text empirical discussions.** Thank you for the suggestion. Given that the final version allows for one extra page in the main body, we will dedicate that extra page to move the key empirical observations to the main body, we will provide more discussions on them, and will also add new empirical results. More specifically, we have additional experimental settings to evaluate the performance of RO-CRL across variations in the latent dimension, observed dimension, and intervention types in random graphs (We kindly refer the reviewer to **Additional experiments** response to reviewer FrNT). Note that the tables in this response will be replaced by figures in the revised paper (the responses don’t allow images).
>
> ““**Section 6. Experiments**
>
> In this section, besides discussing and previous evaluations, we also provide new ones to assess the regret and latent recovery performance of RO-CRL across different intervention types and graph sizes. In all settings, we repeat the experiments 50 times.
>
> First, we demonstrate that RO-CRL shows sublinear regret, as established in Theorem 3. In Figure 4, we compare the regret of RO-CRL under soft (left) and hard interventions (right) on the same graph with $n=5$. We can see under both settings, after the forced exploration period (shaded as yellow), the algorithm converges and shows sublinear regret, validating results in Theorem 3. Thanks to the dimension-reduction step and the forced-exploration schedule, CRL's regret remains essentially unchanged as we increase the feature dimension $d$ from 5 to 100. Table R1 reports the corresponding runtime for each setting at a fixed time horizon $T$, and the additional computational cost introduced by higher $d$ remains well within practical limits. Finally, in Table R2, we show cumulative regret across different latent dimensions $n$. We note that all settings show sublinear regrets in the plot. Furthermore,  In Table R3, we compare the RO-CRL with other bandit algorithms.
>
> Next, we validate the latent recovery performance of RO-CRL. Figure 1 shows the graph recovery rate versus the sample size $s\_t$ for $n=5$. As expected, it is observed that the recovery improves with more samples $s\_t$. Furthermore, hard interventions (which are stronger than soft) consistently yield higher recovery rates than soft interventions. Figure 3 shows the average estimation error on latent variables Z for varying $s\_t$. We can see both terms decay, which conforms to the theoretical decay $1/s\_t$ rate established in Theorem 2. We next show in Table R4 that these observations are consistent across variations in $n$, $d$, intervention type.””
>
> We do agree that illustrative examples would help readers to interpret the results more easily, and will include a toy example over the 2-node chain graph where the UCB criterion converges despite the variable recovery not being perfect. This will demonstrate that perfect recovery is not strictly required for good downstream task performance. We cannot include images in the response due to conference rules, however, the example consists of a full specification of the true and estimated latent variables (the means, edge weights, utility weights), and shows the calculations for UCB of the suboptimal arms dominated by the optimal arm’s true mean.
>
> **Q3. Known bounds for parameter magnitudes and noise levels.** Thank you for the thoughtful question. We first address it from a theoretical perspective, and then provide additional experiments to assess the sensitivity to the assumed bounds. The boundedness of the weight matrices is not a critical issue. In the CRL setting, latent variables can be recovered only up to a scaling factor, and this level of recovery does not compromise recovery of X (or any downstream task that uses X). In fact, recovery up to scaling recovers X as well as perfectly estimating Z. This is not limited to linear transformation, but to get the intuition across, we discuss the reason in the context of linear transformations in which X=G⋅Z. Such a transformation is equivalent to another system with the model X=(aG)⋅(Z/a) for any nonzero scalar a. Therefore, if we recover Z/a instead of the ground-truth Z, we can compensate for the scaling by estimating the transformation as a instead of G. In this sense, recovery of the latent variables is exact up to scaling, that is, any element of the set {Z/a:a∈R} is equally valid. Consequently, imposing boundedness on Z does not compromise the results. This boundedness can, in fact, be equivalently modeled by requiring the weight matrices that govern the dependencies among the coordinates of Z to be bounded.
>
> We perform additional experiments and provide regret and variable recovery results under violations of Assumption 1 and boundedness assumptions in Tables R5–7 in the response to reviewer FrNT.
> - **Assumption 1 violations:** We observe that graph recovery consistently fails (the graph recovery rate is exactly 0), which is consistent with our answer to Q3. However, Table R5 shows that latent variable recovery still works, which is consistent with the fact that its analysis is independent of Assumption 1. In Table R6, we evaluate cumulative regret versus time, and observe that the bandit algorithm does not achieve a sublinear regret after the forced exploration phase ($t \geq 18000$ in this case), meaning that RO-CRL algorithm requires Assumption 1 to ensure such sublinear regret results.
> - **Boundedness violations:** Both violations in noise and parameter bounds will lead to linear regret after the forced exploration phase. This aligns with theoretical expectations: violating either of them leads to confidence radii that no longer appropriately capture the true uncertainty, thereby causing increasingly poor decisions.

---

> > ### Comment · Reviewer_J78L · 2025-08-01
> >
> > Thank you for your response. Since you addressed my concerns, I've decided to increase my score.

---

### Official Review · Reviewer_FrNT · 2025-07-04

**Clarity:** 3
**Significance:** 4
**Originality:** 3
**Rating:** 5
**Confidence:** 2

**Summary:**

This paper tackles the problem of learning causal representations (causal graphs and structural equations) in a setting where the goal is not to learn a representation that inverts the data generating process, but that can accurately model which interventions maximize a reward. The analysis is restricted to the setting where both the causal and reward models are linear. An algorithm for inference of the latent parameters in the limited time horizon is derived, as well as associated regret bounds. Some preliminary experimental results are obtained to validate the method.

**Questions:**

- In line 31, a large number of papers from the same authors are cited. Please be careful about self-citation and try including citations from a more diverse set of authors
- As a suggestion for readability, I strongly suggest not using bold typeface to put emphasis, it is distracting for the reader. I suggest using italics if necessary

**Ethical Concerns:**

["NO or VERY MINOR ethics concerns only"]

**Final Justification:**

I acknowledge the authors rebuttal and am looking forward to see the paper with the promised changes. I am happy to recommend this paper for acceptance.

**Limitations:**

- Limitations should be discussed in the main text and not in the appendix

**Paper Formatting Concerns:**

No.

**Quality:**

3

**Strengths And Weaknesses:**

## Strengths
- This paper formulates a novel and interesting problem of learning causal representations with the goal of maximizing a reward instead of inverting the data generating process. This could have important theoretical and practical applications
- The theoretical analysis seems impressive and generalizes some existing results. It importantly operates in the finite sample setting

## Weaknesses
- I would like to see some experimental validation of the regret bounds. Figure 4 does suggest sublinear regret, but it would be great to have a more detailed analysis
- It would also be interesting to have experimental comparison with other appropriate bandit algorithms
- The paper is clear, but a bit heavy to read. I suggest trying to include a figure that explains the main algorithm. This could greatly help understanding
- There is also little discussion of practical examples of applications of this framework. I suggest trying to include some to ground the discussion and highlight possible uses.

---

> ### Author Rebuttal · Authors · 2025-07-31
>
> **Q1. Citations.** We appreciate this feedback. In the revision, we will expand the discussions and to include more general discussion about causal representation learning and its latest developments.
>
> **Q2. Bold typeface.** Thank you for noting this. We will revise this accordingly.
>
> **W1. Experimental validation of the regret bounds.** Thank you for the suggestion. We incorporate new empirical results to validate the regret bounds. More specifically, we provide the following additional results:
> - Varying observation dimension $d$: Thanks to dimensionality reduction techniques and the forced-exploration schedule, CRL's regret remains essentially unchanged as we increase the feature dimension $d$ from 5 to 100 while keeping the latent dimension at $n=5$. This observation is consistent with some previous findings in CRL under linear transformations [5] that the performance of CRL is not that sensitive to the feature dimension $d$, and this behavior does not contradict our lower bound analysis, which is derived under the error bounds from CRL. Table R1 reports the corresponding runtime for different $d$at a fixed time horizon $T$, and the additional computational cost introduced by higher $d$ remains well within practical limits.
>
> Table R1: Average runtime (in minutes) with varying $d$ when $n=5$
> |$d$|5|10|25|50|75|100|
> |-|-|-|-|-|-|-|
> |soft|48.55|47.19|53.85|48.01|48.11|48.55|
> |hard|63.12|65.83|67.66|67.24|67.54|68.32|
>
> - Varying latent dimension $n$ (which also affects the $u$t) for fixed $d=10$. From table 2, we observe that the cumulative regret grows by increasing $n$. We note the increase stems from two factors: (i) exploration cost for CRL varies with $n$, and (ii) the reward range becomes larger as $n$ increases These trends will be more clearly visualized in the figures to be included in the final version.
>
> Table R2: Cumulative regret with different latent dimensions $n$ when $d=10$
> |$n$|3|5|7|9|
> |-|-|-|-|-|
> |soft|1524|4832|13298|37787|
> |hard|1789|5382|14677|37830|
>
> **W2. Comparison with other bandit algorithms.** We have included additional results to compare against two baselines: vanilla UCB and a modified version of RO-CRL, where CRL is performed for a fixed fraction of the sample budget, after which the bandit proceeds without accounting for latent recovery error. Table R3 reports the cumulative regret at the given final horizon $T$ for different algorithms. As shown, both baselines perform significantly worse than RO-CRL. The performance gap between RO-CRL and Modified RO-CRL will continue to increase with longer time horizons.
>
> **Table R3:** Cumulative regret across different algorithms
> |Algorithm|RO-CRL|UCB|modified RO-CRL|
> |-|-|-|-|
> |hard|5382|15690|10132|
> |soft|4832|16320|9726|
>
> **W3. Include a figure that explains the main algorithm.** We will provide a schematic description of the pipeline of the processes to highlight the main steps of the framework and algorithm. This will, specifically, constitute the following key pieces:
> - CRL subroutine that is called after each exploratory sample
> - The key processes involved in UCB-based selection of interventions
> - Annotation on the link between estimates and original parameters
> - Idea behind the under-sampling rule
>
> **W4. Practical examples of applications.** We provide two potential use case examples for the RO-CRL framework and will include them in the revised paper.
> - **Robotics:** Consider a robot operating in an environment that is observed through high-dimensional sensory data such as images. Interventional CRL offers a way to recover interpretable, disentangled latent factors (e.g., joint angles, end-effector positions) that causally generate visual data. CRL is particularly suitable for robotics in vision-based contexts because *high-dimensional observations* (images) obscure the causal variables and *robots* have known or assumed causal structure (e.g., a joint​ causes an end-effector position​). In robotics, interventions on the latent variables can be achieved physically or via simulation. Physical interventions include, for instance, directly changing a joint angle or velocity (e.g., move joint 3 by +5°) and moving the robot gripper to a fixed position while leaving other joints free. In this context, an example downstream objective could be moving the robot arm to a specific configuration using only the utility values and learned latent factors as control input.
> - **Genomics:** In gene regulatory networks, we observe high-dimensional expression data, typically measured from bulk or single-cell RNA sequencing. These observations are assumed to arise from a set of lower-dimensional latent variables, which represent unobserved biological drivers such as transcription factor activities, regulatory protein states, or signaling pathway activations. CRL is a natural fit for gene regulatory analysis because gene expression is influenced by a structured, latent causal process involving transcriptional and post-transcriptional regulation. Standard representation learning techniques fail to recover biologically meaningful variables, often mixing causal and non-causal factors. Node-level interventions in the CRL framework correspond to perturbing individual latent regulatory variables. In gene networks, such interventions are biologically realizable via *gene knockouts* or *CRISPR interference* to silence specific genes or transcription factors and *overexpression systems* to activate regulators, to name a few. In this context, an example downstream objective could be to optimize a certain type of biological response over the set of some possible genetic modifications.
>
> **Limitations.** We will move the limitations paragraph to the conclusion.
>
> -----
>
> **Additional Experiments.** Finally, we include additional experiments.
>
> - **Performance of CRL when varying $n$ and $d$.** In Table R4, we provide additional experiments to evaluate the performance of CRL across different $n$, $d$ and intervention types, plotted against sample size $s\_t$. These results show that CRL performance of our RO-CRL algorithm is mostly consistent across variations in system parameters: Variable recovery errors decay with increasing number of samples $s\_t$ consistently with the theoretical rate of $1/s\_t$ (Theorem 2) across all settings. Observation dimension $d$ does not affect results thanks to a dimensionality reduction step. Finally, increasing $n$ from 5 to 8 does not lead to a significant drop in variable recovery performance. However, graph recovery performance significantly degrades for $n=8$ under soft interventions, which can be explained by noting that the “true” graph under soft is the transitive closure which includes high numbers of indirect edges. Such indirect effects are hard to track from the precision matrices, which makes the graph recovery for soft interventions more difficult.
>
> **Table R4:** CRL performance versus $n$, $d$ and intervention type.
>
> (a) Latent recovery (MSE):
>
> |$n$|$d$|Int. type|$s\_t =$ 1000|2000|4000|8000|16000|
> |-|-|-|-|-|-|-|-|
> |5|5|Soft|0.067|0.052|0.034|0.026|0.018|
> |5|25|Soft|0.201|0.189|0.170|0.162|0.156|
> |8|8|Soft|0.136|0.092|0.069|0.047|0.035|
> |5|5|Hard|0.218|0.165|0.069|0.047|0.017|
> |5|25|Hard|0.378|0.320|0.232|0.186|0.147|
> |8|8|Hard|0.246|0.176|0.099|0.047|0.034|
>
> (b) Graph recovery rate:
>
> |$n$|$d$|Int. type|$s\_t =$ 1000|2000|4000|8000|16000|
> |-|-|-|-|-|-|-|-|
> |5|5|Soft|0.00|0.22|0.80|0.94|1.00|
> |5|25|Soft|0.02|0.10|0.64|0.80|0.94|
> |8|8|Soft|0.00|0.04|0.00|0.00|0.00|
> |5|5|Hard|0.02|0.10|0.68|0.84|1.00|
> |5|25|Hard|0.00|0.08|0.50|0.82|0.98|
> |8|8|Hard|0.00|0.08|0.50|0.96|1.00|
>
> - **Assumption Violations.**
>   - **Assumption 1:** We run experiments where Assumption 1 is violated and provide the regret and variable recovery results in Tables R5–6. First, we note that graph recovery consistently fails (the graph recovery rate is exactly 0) in this setting, which is consistent with our answer to Q3 of reviewer Z2af. However, Table R5 shows that latent variable recovery still works: Indeed, its analysis is independent of Assumption 1. In Table R6, we evaluate cumulative regret versus time, and observe that the bandit algorithm does not achieve a sublinear regret after the forced exploration phase ($t \geq 18000$ in this case), meaning that RO-CRL algorithm requires Assumption 1 to appropriately capture the true uncertainties and make good decisions.
>
> **Table R5:** Cumulative regret under Assumption 1 violation ($n=d=5$)
>
> |$t$|3000|6000|9000|12000|15000|18000|21000|24000|
> |-|-|-|-|-|-|-|-|-|
> |Regret|0|2000|4000|4000|6000|6000|11000|16000|
>
> **Table R6:** Latent variable recovery MSE under Assumption 1 violation ($n=d=5$)
>
> |$s\_t$|1000|2000|4000|8000|16000|
> |-|-|-|-|-|-|
> |MSE|0.134|0.123|0.104|0.097|0.092|
>
> - **Boundedness violations:** We conduct experiments that deliberately violate two key assumptions: bounded noise and known parameter bounds. The specific violations are as follows:
>     - Noise violation: We replace the bounded noise with unbounded Gaussian noise.
>     - Parameter violation: Instead of proper rescaling the system, we multiply all parameters by a factor of 10 to simulate the violation. Note that such a scaling can be offset by properly scaling the latent variables (see e.g. response to J78L, Q3). Therefore, in these experiments, we explicitly set a fixed scale of the latent variables.
>
>     As shown in Table R7, both violations result in linearly growing regret after the forced exploration phase ($t \geq 18000$ in this case). This result aligns with theoretical expectations: violating the noise boundedness or known-parameter assumptions leads to confidence radii that no longer appropriately capture the true uncertainty, thereby causing increasingly poor decisions.
>
> **Table R7:** Cumulative regret under bound violation ($n=d=5$)
> |$t$|6000|12000|15000|18000|19000|20000|21000|
> |-|-|-|-|-|-|-|-|
> |noise|1909|4272|5382|5382|7176|8970|10764|
> |parameters|421447|505197|516290|516290|685092|857133|1029174|

---

> > ### Comment · Reviewer_FrNT · 2025-08-01
> >
> > I acknowledge the authors rebuttal and am looking forward to see the paper with the promised changes. I am happy to recommend this paper for acceptance.

---

### Official Review · Reviewer_dqb6 · 2025-07-11

**Clarity:** 2
**Significance:** 2
**Originality:** 3
**Rating:** 4
**Confidence:** 2

**Summary:**

This work considers the problem of causal representation learning from a reward-oriented perspective, where, contrary to prior works which optimize for perfect identifiability, they optimize for the coarsest level of identifiability for a specific downstream task. Focuses on the specific task of identifying an optimal set of interventions as specified by a linear utility function in terms of the latent variables. Considers linear SEMs with linear transformations to the latent space. Provides an algorithm, RO-CRL, for the reward-oriented CRL task under hard and soft interventions and specifies regret bounds for both settings.

**Questions:**

1. The SEM, transformation and utility function all being linear seems restrictive. Could these results be extended to any nonlinear setting?
2. Is the dimension of the latent space assumed to be known a priori? If the causal variables are inherently latent, how could you know how many there are?
3. Similarly, why can we assume that interventions on single latent variables aren’t actually affecting many latent variables?

**Ethical Concerns:**

["NO or VERY MINOR ethics concerns only"]

**Final Justification:**

The authors sufficiently addressed my concerns outlined in Weakness 1 and 2. While I believe this method to be very interesting and important for the field of CRL, which can sometimes feel impractical due to the infeasibility of many assumptions needed for perfect identifiability, I have questions regarding how this method will work in practice. For instance, I am not convinced that knowing latent dimensions and latent factors a priori is realistic or that methods could even have the ability to intervene on all latent variables. Real-world experiments would have been great, but I understand the difficulty given the lack of datasets.

Ultimately, I believe the work is interesting enough and has enough potential to be accepted, but I still remain underconfident in my evaluation.

**Limitations:**

yes

**Quality:**

2

**Strengths And Weaknesses:**

Strengths:
1. This work addresses a key gap in CRL literature by developing methods that achieve identifiability only to the extent necessary for maximizing performance on downstream tasks. Current works focus primarily on perfect identifiability, requiring an excessive and arguably unrealistic number of interventions, thus making this work potentially more applicable.
2. All proofs and theoretical findings seem correct.
3. Paper was generally well written and enjoyable to read.

Weaknesses:
1. My primary concern is that the paper claims that prior works focused on perfect identifiability employ an ‘excessive’ number of interventions to achieve perfect identifiability; however [3] (as cited in the manuscript) indicates that a single atomic intervention on each node is sufficient for identifiability, where RO-CRL enforces more interventions than that in just stage 1of the algorithm. This seems misleading and questions whether this work is really capable or discovering meaningful causal representations more efficiently.
2. Stage 1a and 1b of RO-CRL seems to indicate the ability to take both hard and soft interventions as desired; however, given the interventions are on unknown latent variables, it seems restrictive to think that you could know entirely whether an intervention is hard or soft.
3. The work includes very limited synthetic experimental results and no real-world experimental results. It would be very interesting to see a real world example of how your approach could learn representations geared for a specific downstream task.

---

> ### Author Rebuttal · Authors · 2025-07-31
>
> **Q1. Generalize to non-linear settings.** Thank you for the question. While linear models represent an important class of causal models, transformations, and rewards, we do agree with the reviewer that generalization to nonlinear settings is also important. We note that even in the simpler contexts (e.g., CRL or CB), the linear versus nonlinear models are investigated extensively. Here, we outline how the linear SEM, linear transformation, and linear utility can be individually generalized. To provide a concrete methodology for extensions, we first note that our current analysis consists of two parts: (i) analyzing finite-sample guarantees of a CRL algorithm, and (ii) analyzing how the CRL guarantees translate to reward guarantees. We discuss how the same decomposition can address nonlinear models as well:
> - **Latent SEM:** It is possible to generalize the algorithms and performance guarantees to nonlinear SEMs. While the overall RO-CRL framework and pipeline remain intact, such a generalization has two distinct implications: one for the CRL guarantees and another for the CB guarantees. Setting aside the technical details, here is a high-level description of how this generalization works. On the CRL side, the CRL subroutine can be replaced with methods designed for general SEMs. For example, [11] provides an algorithm along with finite-sample guarantees applicable to general SEMs. As a result, the performance guarantees stated in Theorem 2 would need to be adjusted accordingly. On the bandit side, the UCB rule and reward estimators can similarly be replaced with their counterparts developed for nonlinear causal bandits (for instance, those based on Gaussian processes or neural networks [20, 23]). Such generalization, however, is highly non-trivial and even in the simpler contexts (e.g., standalone CRL and CB), the linear and nonlinear settings are investigated separately, each with its exclusive technical challenges.
> - **Transformation:** There are existing results for CRL under nonlinear transformations [9, 10, 13]. However, finite-sample analysis of CRL under general transformations is – to our knowledge – still an open problem. Nevertheless, we note that any future finite-sample result can be integrated into our RO-CRL framework to extend it to nonlinear transformations: Given that the estimator provides a graph and variables recovery error bound similar to Theorem 2, these estimates and error bounds can be used by the robust causal bandit algorithm. Similar to SEM generalization, as long as the setting does not violate the CB assumptions, the overall pipeline would work.
> - **Utility:** Our utility‑function assumption is a direct extension of the latent SEM assumption. Specifically, the framework can be modularly generalized to accommodate non‑linear utilities when adopting the robust non-linear estimates in bandit settings. The key open question, therefore, is the same one that arises for latent‑SEM causal‑bandit methods: How robust are these algorithms to violations in the underlying variable?
>
> **Q2. Dimension of the latent space.** The latent dimension is assumed to be known a priori, and there is a mechanical reason why: Our action space is *defined* by the latent space. That is, the number of controllable variables equals the number of latent dimensions. In fact, in CRL, the recovered variables are generally identified with their corresponding interventions. This enables the recovery of variables with known names/semantics but unknown values from high-dimensional observations. We note that for this reason, knowing the latent dimension – even if not strictly realistic – is a very common assumption in causal representation learning literature, including all references [2-13].
>
> **Q3. Single-node interventions aren’t actually affecting many latent variables?** Thank you for the thoughtful question. Indeed, transitioning to multinode intervention is a natural and realistic extension. We have adopted the single-node intervention model to facilitate clarity in exposition and describing the key ideas. Since this is the first paper addressing reward-oriented CRL, we deliberately chose single-node interventions to place the focus on the key ideas. At the expense of more crowded notations, all the steps can be readily extended to multi-node interventions, and all the logics of all the key processes remain the same up to some minor adjustments. We note that even in the CRL literature (without the downstream tasks), single-node interventions have been the primary focus primarily for the same reasons of facilitating clarity in analysis. Nonetheless, relaxing this assumption using existing techniques is a promising direction for future work.
>
> **W1. Number of interventions.** We would like to clarify what appears to be a misunderstanding. CRL guarantees depend on two issues: (1) how many intervention environments per node are necessary/sufficient? (2) What is the recovery guarantee for a given number of samples in each intervention environment? The full identifiability results generally dispense with the second dimension, assuming an *infinite* number of samples per intervention environment is available, and focus on characterizing how many intervention environments per node are required to ensure identifiability. As the reviewer correctly points out, the state-of-the-art results establish that one or two intervention environments per node is sufficient for perfect identifiability, assuming that under each environment we get an infinite number of samples.
>
> In our framework, we also assume that we have exactly one intervention per node. Our key questions are
> 1. What happens if we back off from having an infinite number of samples and assume access to only a finite number of samples?
> 2. If we have a limited number of samples, how can we collect them sequentially and adaptively to the data so that we can form the best decision with the fewest samples?
>
> The sequential design of data collection is naturally posed as a bandit framework. We hope this clarifies the reviewer’s primary concern about the scope of the paper.
>
> **W2. Hard and soft interventions.** Thank you for the question. To address this concern, we note three points:
>
> 1. Hard interventions are special cases of soft interventions. When lacking information about what type of intervention has been used, we go with the default choice of assuming a soft intervention as it will accommodate both.
> 2. Stage 1a is common to both soft and hard interventions. This stage will be carried out regardless of whether we know we have soft or hard intervention. If we know we have soft or we do not know the type of intervention, the CRL part of the algorithm terminates after completing Stage 1a.
> 3. Stage 1b is exclusive to only hard intervention and will be carried out on top of Stage 1b only when we know that we have hard interventions. In such a case, carrying out Stage 1b, will refine the results and guarantees we get from performing only Stage 1a.
>
> We note that such a conditional post-processing depending on *a priori* knowledge of whether hard interventions are used is not uncommon in CRL literature [3, 5].
>
> **W3. Limited evaluation and real-world examples.** We have performed additional experiments to evaluate the performance of RO-CRL across variations in the latent dimension, observed dimension, and assumption violations. The results are given in the additional experiments with Tables R1-7 in Response to reviewer FrNT. These results corroborate our existing evaluations in showing the efficacy of our proposed algorithm.
>
> Regarding real-world applications, unfortunately, currently there are no available datasets that are amenable to applying our algorithm. We hope that with the fast growth of the CRL literature, there will be such datasets for real-world experiments. We provide two potential use cases which we will include in the revised paper.
> - **Robotics:** Consider a robot operating in an environment that is observed through high-dimensional sensory data such as images. Interventional CRL offers a way to recover interpretable, disentangled latent factors (e.g., joint angles, end-effector positions) that causally generate visual data. CRL is particularly suitable for robotics in vision-based contexts because *high-dimensional observations* (images) obscure the causal variables and *robots* have known or assumed causal structure (e.g., a joint​ causes an end-effector position​). In this context, an example downstream objective could be moving the robot arm to a specific configuration using only the utility values and learned latent factors as control input. However, image generation is a highly nonlinear process, which does not fit our modeling assumptions.
> - **Genomics:** In gene regulatory networks, we observe high-dimensional expression data, typically measured from bulk or single-cell RNA sequencing. These observations are assumed to arise from a set of lower-dimensional latent variables, which represent unobserved biological drivers such as transcription factor activities, regulatory protein states, or signaling pathway activations. CRL is a natural fit for gene regulatory analysis because gene expression is influenced by a structured, latent causal process involving transcriptional and post-transcriptional regulation. Node-level interventions in the CRL framework correspond to perturbing individual latent regulatory variables. In gene networks, such interventions are biologically realizable via *gene knockouts* or **CRISPR interference* to silence specific genes or transcription factors and *overexpression systems* to activate regulators, to name a few. In this context, an example downstream objective could be to optimize a certain type of biological response over the set of some possible genetic modifications. However, we are not aware of current genomics datasets that support arbitrary multi-node interventions, which limits a direct evaluation at this time.

---

> > ### Comment · Reviewer_dqb6 · 2025-08-01
> >
> > Thank you for your response. You have addressed many of my concerns and I am happy to increase my evaluation score.

---

### Official Review · Reviewer_Z2af · 2025-07-12

**Clarity:** 3
**Significance:** 3
**Originality:** 3
**Rating:** 5
**Confidence:** 3

**Summary:**

This paper introduces reward-oriented CRL, which shifts the focus from perfectly recovering latent causal representations to learning them only to the extent needed for optimizing a downstream reward function. The authors consider linear structural equation models (SEMs) with linear transformations mapping latent variables to observations, and linear utility functions. They develop the RO-CRL algorithm that adaptively explores interventions while maintaining confidence bounds, providing finite-sample identifiability guarantees and regret bounds of $\tilde{O}(d^{1/3}n^{1/3}u^{2/3}T^{2/3} + u\sqrt{T})$ for soft interventions, with nearly matching lower bounds.

**Questions:**

1. Could the precision matrix approach generalize using kernel methods or neural approximations for the nonlinear setting?

3. How does $u$ scale with graph properties in typical instances? Can you provide examples where the gap between $u_S$ and $p$ is large vs. small? This would help understand when the bounds are truly tight.

4. What happens when Assumption 1 is violated (i.e., interventions only affect noise)? Could the algorithm be modified to handle or detect such cases?

5. What is the per-round computational complexity of RO-CRL? The eigendecomposition in equation (11) and matrix inversions throughout seem expensive for large $d$?

Not a question but the work of Chen et al. NeurIPS'24. "Identifying General Mechanism Shifts in Linear Causal Representations" seems relevant as additional CRL literature. There, the authors indentify general interventions under a similar linear CRL setting.

**Ethical Concerns:**

["NO or VERY MINOR ethics concerns only"]

**Final Justification:**

The authors addressed my questions. I think it is a good paper.

**Limitations:**

I have only skimmed over the appendix so far. The authors could discuss further:
- Computational scalability concerns for high-dimensional observations
- Sensitivity to assumption violations (especially bounded noise and known parameter ranges)

**Quality:**

3

**Strengths And Weaknesses:**

## Strengths

**Quality:**
- The paper provides rigorous theoretical analysis with finite-sample PAC identifiability guarantees (Theorem 2) that extend beyond typical asymptotic results in CRL
- The regret bounds are nearly tight, with upper bound $\tilde{O}(d^{1/3}n^{1/3}u^{2/3}T^{2/3} + u\sqrt{T})$ and lower bound $\Omega(d^{1/3}n^{1/3}p^{2/3}T^{2/3} + p\sqrt{T})$ differing mainly in graph-dependent constants
- The algorithm design cleverly balances CRL objectives with bandit optimization through weighted ridge regression and UCB-based selection
- As a special case, the framework improves existing causal bandit bounds from $\tilde{O}(d^{3/2}L\sqrt{T})$ to $\tilde{O}(d_G^L\sqrt{T})$

**Clarity:**
- The paper is well-structured with clear motivation for task-oriented CRL
- The algorithmic components (forced exploration, adaptive exploration with CRL rules, UCB selection, and under-sampling) are logically presented
- Mathematical notation is consistent and appropriately defined

**Significance:**
- Addresses a fundamental gap between CRL's goal of perfect recovery and practical needs where coarse representations suffice
- The framework naturally connects CRL and causal bandits, two previously disparate areas
- The finite-sample analysis is particularly valuable for practical applications

**Originality:**
- The reward-oriented perspective on CRL is novel and well-motivated
- The technical approach of using precision matrix differences for simultaneous graph learning and intervention optimization is creative

## Weaknesses

**Quality:**
- The restriction to linear SEMs and linear transformations is certainly limiting
- The assumptions (particularly Assumptions 1-3) are somewhat strong: bounded noise, known ranges for parameters, and intervention effects on weight matrices.
- The dependence on the uncertainty parameter $u$ in the bounds is not well-characterized in terms of problem instances

**Clarity:**
- The relationship between $u_S$, $u_H$, and $p$ in the bounds could be explained more intuitively
- The weight matrix design in equations (24)-(26) appears ad-hoc and could benefit from clearer justification
- The transitive closure recovery under soft interventions vs. exact graph recovery under hard interventions needs better exposition

**Significance:**
- The improvement from $u$ to $p$ in bounds may not be substantial in many graphs where these quantities are similar

**Originality:**
- While the reward-oriented perspective is novel, the technical tools (precision matrices, ridge regression, UCB) are somewhat standard

---

> ### Author Rebuttal · Authors · 2025-07-31
>
> **Q1. Generalize precision matrix to nonlinear settings.** Yes, such generalizations are indeed possible. We kindly refer the reviewer to our reply to “Latent SEM” in **Question 1. Generalize to non-linear setting** in the response to reviewer dqb6 for a detailed discussion of this subject.
>
> **Q2. Scaling of $u\_H, u\_S, p$.** Thank you for raising this point. Under hard intervention, $u\_H$ is upper bounded by $d_{G}^L$. Under soft intervention, the terms $u_S$​ and $p$ both grow at the same rate, which is exponentially with the number of nodes $n$ in most graphs. As a simple example, consider a chain graph, where each node $i$ is a child of node $i-1$. Hence, the ancestor set of node $i$ will be $\\{1,2,\cdots, i-1\\}$. Since both $u$ and $p$ (with equivalent definition in (217)-(218)) involve summation over the ancestor set, they both scale as $2^n$.
>
> Beyond scaling similarly, we can also quantify the reward gap by comparing their growth rates. Since the difference in $u$ and $p$ lies in the uncertainty added at each layer ($\sqrt{|An(u)|}$ vs. 1), the gap does not exceed their exponential growth. Based on this, we can show that the ratio of $u\_S$ to $p$ converges to a graph-dependent constant $c$ for common graphs. For instance, we have $c \leq 1.21$ for the hierarchical graph and $c \approx 1.174$ for the chain graph. This indicates that the upper bound on regret is only a constant factor larger than the lower bound.
>
> **Q3. Violation of Assumption 1.** Thank you for the thoughtful question. It is shown in [3, Appendix B] that without this assumption, the causal graph is *not identifiable*. For instance, without this assumption, we cannot distinguish between 2-node graphs with and without an edge. Since performing any downstream task hinges on an identifiable underlying causal model, if this assumption is not satisfied, we can information-theoretically argue that no algorithm can have a provable guarantee. We will add a remark right after Assumption 1 to show its fundamental connection to identifiability guarantees in CRL.
>
> **Q4. Per-round computational complexity.** The computational cost can be broken down into two parts:
> 1. **CRL:** Complexity of eigendecomposition and matrix inversion are both $O(d^3)$ and thus expensive for large $d$. However, since the transformation is linear, detecting the supporting subspace is simple: given that the observation model $X = g(Z)$ is noise-free, it suffices to collect $n$ samples to learn this subspace, which can be achieved during the initial forced exploration phase. Projecting all observations $X$ to this subspace (of dimension $n$) before doing further processing significantly improves scalability.
> 2. **UCB-based selection:** The computational bottleneck is the intervention‑selection step. Solving the UCB rule is intractable on general causal graphs (see [25, §3.3]). In practice, we may adopt the following procedure.
>     - Non‑negative edge weights and noise (shifted system): We can add a shift to the system to make the system non-negative while not impacting the order of regret. Because all rewards are monotone in their parents’ variables, we can calculate the UCB by sequentially maximizing nodes in causal order and using the closed‑form in linear bandit. The resulting worst‑case complexity is $O(n\*d\_{\hat G\_t}^3)$, where $n$ is the number of latent dimensions and $\hat d\_{G\_t}$ is the maximum in-degree in graph $\hat G\_t$
>     - UCB upper bounds: [25] proposes techniques that efficiently compute an upper bound on the UCB. These could be incorporated into our approach. But the upper bound is loose in practice, and their robustness to imperfect latent recovery is unknown.
>
> **On Chen et al. NeurIPS'24.** Thank you, we will include the reference in the final version of the paper. This paper’s setting and corresponding finite-sample results may provide another way to derive reward-oriented CRL results similar to our paper.
>
> **Quality 1. Restriction to linear SEMs.** We kindly refer the reviewer to our reply to **Question 1. Generalize to non-linear setting** in the response to reviewer dqb6.
>
> **Quality 2. Assumptions.** Our assumptions are standard, and some are necessary for identifiability guarantees of CRL and finite‑sample guarantees. More specifically:
>
> - *Assumption 1*. As noted earlier, without this assumption, CRL becomes unidentifiable. We will add a remark to clarify the necessity for this assumption.
>
> - *Assumption 2.* The boundedness of the weight matrices is not a critical issue. In the CRL setting, latent variables can be recovered only up to a scaling factor, and this level of recovery is sufficient for downstream tasks (even for reconstructing X). This is because the system X=G⋅Z is equivalent to X=(aG)⋅(Z/a) for any nonzero scalar a. Therefore, if we recover Z/a instead of the ground-truth Z, we can compensate for the scaling by estimating the transformation as a instead of G. In this sense, recovery of the latent variables is exact up to scaling, that is, any element of the set {Z/a:a∈R} is equally valid. Consequently, imposing boundedness on Z does not compromise the results. This boundedness can, in fact, be equivalently modeled by requiring the weight matrices that govern the dependencies among the coordinates of Z to be bounded.
>
> - *Assumption 3.* The primary goal of these assumptions is to assume a bounded system. Even in the simpler linear bandit setting, where the reward node is a child of all other nodes and there are no additional causal relations, analyses assume bounded feature vectors and weights [R1, R2]. Our causal setting naturally generalizes from that to SEMs, where we require the parents of each node to be bounded, which is the result of bounded weights and noise terms. These assumptions are standard in the causal bandit literature (e.g., [21]).
>
> [R1] Abbasi-Yadkori et al. Improved Algorithms for Linear Stochastic Bandits. NeurIPS 2021
>
> [R2] Flynn et al. Improved Algorithms for Stochastic Linear Bandits Using Tail Bounds for Martingale Mixtures. NeurIPS 2023
>
> **Quality 3. Dependence of bounds on $u$ .** In equation (35), we specify the parameters $u\_H$ and $u\_S$ in terms of the graph topology. We acknowledge that using the notation $u\_m$ to refer to $u\_H$​ and $u\_S$​ may create confusion, as it could be interpreted as a separate parameter. We will revise the notation to avoid this ambiguity.
>
> To interpret (35), note that the parameter $u$ captures the cumulative uncertainty in the reward due to the estimation of latent SEM parameters. These uncertainties propagate along the causal paths from the root node(s) to the reward node. Accordingly, we define $u$ to reflect the accumulated uncertainties in estimating the weight matrices along these paths. This accumulation is expressed through the summation terms in (35). We will add a clarifying comment after the equation to make this interpretation explicit.
>
> **Clarity 1. The relationship between $u\_S$, $u\_H$, and $p$ in the bounds.** Please see the response to Q2.
>
> **Clarity 2. The weight matrix design in equations (24)-(26).** Thanks for asking us to clarify. We give lower weight to recovered latent variables that appear uncertain, based on the data collected so far. The weight matrix design is carefully crafted to keep our estimates tight and makes the confidence radius $\beta\_{i,t}$ in Eqs. (27)-(29) independent of the running estimation error, which feeds into the reward. It handles two kinds of uncertainty:
> 1. Overall estimation error  (captured by $\zeta\_t$): When this aggregate error is large, we down‑weight all samples, requiring more evidence before drawing overly optimistic conclusions.
> 2. Sample‑specific uncertainty: A weighted $\ell\_2$ norm acts as an exploration bonus, reducing the influence of individual samples that sit far from the population in linear regression and may be noisier than others because of imperfect recovery.
> By integrating these two factors, the weighting mechanism focuses learning on the most reliable information rather than relying on ad‑hoc choices.
>
> **Clarity 3. The transitive closure recovery under soft interventions vs. exact graph recovery under hard interventions.** Thank you for the suggestion. We will add explanations for why there is fundamentally such a distinction in observation between hard and soft interventions. For that, we will cite the known impossibility results that establish that such recoveries are the best that can be guaranteed under the specified interventions.
>
> **Significance. The improvement from $u$ to $p$ in bounds may not be substantial.** We would like to clarify that $u$ and $p$ being close is a *desirable* property. These terms do *not* represent improvements. Rather, $p$ appears as a constant in the lower bound and $u$ as a constant in the upper bound. Therefore, having these two constants be as close as possible is desirable, as it establishes the tightness of the bound.
>
> **Originality. While the reward-oriented perspective is novel, the technical tools (precision matrices, ridge regression, UCB) are somewhat standard.** We note that some of the results/analysis in CRL do not have precedence in the CRL literature and can be considered contributions to that literature too (e.g., finite-sample guarantees under hard interventions). We do recognize that some of the techniques used in the causal bandit aspects are not entirely novel (although some aspects are novel, e.g., graph-dependent analysis versus worst-case analysis in the literature).
>
> **Limitation 1. Scalability.** Addressed in response to Q1.
>
> **Limitation 2. Sensitivity to assumption violations.** We provide additional experiments for violations of all assumptions. We kindly refer the reviewer to our “Assumption Violations” part in **Additional experiments** in the response to reviewer FrNT.

---

> > ### Author Response · Authors · 2025-08-07
> >
> > We would like to kindly follow up and check if our comments have addressed the reviewer’s questions and concerns.

---

### Decision · Program_Chairs · 2025-09-17

**Decision:**

Accept (poster)

**Comment:**

Causal representation learning is the problem of inverting a generative model to recover latent representations from data along with their causal relationships. In recent years, this problem has seen numerous positive results, mainly focused on identifiability and robustness. This paper addresses an interesting twist on this problem: Instead of providing guarantees on the latent representations themselves, the authors consider proving regret bounds for specific tasks. The results are similar in spirit to bandit-style problems.

The main results are presented in the linear setting, and it is not clear how to extend beyond this case. Other limitations include requiring knowledge of the latent dimension and factors, as well as being able to intervene on every factor. The assumptions also seem to be sufficient to identify everything (as in classical CRL) anyway, so there may not be any advantages to this approach.

Nonetheless, reviewers appreciated the new perspective on CRL (several of whom are themselves experts in CRL), and supported accepting the paper.